



# Drought resistance increases from the individual to the ecosystem level in highly diverse neotropical rain forest: a meta-analysis of leaf, tree and ecosystem responses to drought

Thomas Janssen[1] †, Katrin Fleischer[2], Sebastiaan Luyssaert[3], Kim Naudts[1] & Han Dolman[1]

[1]Department of Earth Sciences, Cluster Earth and Climate, Vrije Universiteit Amsterdam, Amsterdam, the Netherlands
[2]Land Surface-Atmosphere Interactions, Technical University of Munich, Freising, Germany
[3]Department of Ecological Sciences, Vrije Universiteit Amsterdam, Amsterdam, the Netherlands

*Correspondence to*: Thomas A. J. Janssen (t.a.j.janssen@vu.nl)

**Abstract.** The effects of future warming and drying on tropical forest functioning remain largely unresolved. Here, we conduct a meta-analysis of observed drought responses in neotropical humid forests, focused on carbon and water exchange. Measures of leaf, tree and ecosystem scale performance were retrieved from 138 published studies conducted across 229 sites in neotropical forests. Differentiating between seasonal and episodic drought we find that; (1) during seasonal drought, the increase of atmospheric evaporative demand and a decrease of soil water potential results in a decline of leaf water potential, stomatal conductance, leaf photosynthesis and stem diameter growth while leaf litterfall and leaf flushing increase. (2) During episodic drought, we observe a further decline of stomatal conductance, photosynthesis, stem growth and, in contrast to seasonal drought, also a decline of transpiration. Responses of ecosystem scale processes, productivity and evapotranspiration, are of a smaller magnitude and often not significant. Furthermore, we find that the magnitude and direction of a drought-induced change in photosynthesis, stomatal conductance and transpiration reported in a study is correlated to study-averaged wood density. Therefore, wood density is a good proxy of hydraulic behaviour and can be used to predict leaf and tree scale responses to drought. We present new insights into the functioning of tropical forest in response to drought and offer a response-benchmark for land surface models.

## 1 Introduction

The neotropical rainforests of South and Central America, with the Amazon Basin at its centre, cover the largest tract of tropical forest on Earth. As such, these forests are a crucial component of the regional and global climate system as a source of convective heat and moisture, driving atmospheric moisture transport and precipitation patterns (Poveda and Salazar, 2004; Zemp et al., 2014). General circulation models project that South and Central America will warm by 2 °C to 5 °C in the coming decades under the business as usual emission scenario (Marengo et al., 2010). Furthermore, seasonal drought is expected to become more severe (Boisier et al., 2015; Malhi et al., 2009; Marengo et al., 2010). Undisturbed old growth forest in the Amazon Basin has increased in aboveground biomass since the 1980's, acting as a substantial sink of atmospheric carbon





(Feldpausch et al., 2016; Phillips et al., 2009). However, recent drought events appear to have at least temporarily reversed the Amazon carbon sink through reduced productivity (Gatti et al., 2014; Zhao and Running, 2010), elevated tree mortality (Feldpausch et al., 2016; Phillips et al., 2009) and increased emissions from fire (Aragão et al., 2018; Van Der Laan-Luijkx et al., 2015; Van Der Werf et al., 2008). Furthermore, the integrity of neotropical forests may be threatened by unforeseen

feedback mechanisms triggered by drought and deforestation. These vegetation-atmosphere feedbacks can reduce atmospheric moisture recycling and increase carbon emissions, which further amplifies forest loss and global climate change (Cox et al., 2000, 2004; Davidson et al., 2012; Erfanian et al., 2017; Exbrayat et al., 2017; Malhi et al., 2009; Zemp et al., 2017).

Despite the critical role of neotropical forests in driving future climate scenarios, there are large uncertainties surrounding the
sensitivity of these forests to drought. Uncertainties are partly the result of the biological diversity found in neotropical forests as the magnitude and direction of a response to drought is found to be strongly dependent on the species measured (Bonal et al., 2000a; Domingues et al., 2014). Also, uncertainties arise as droughts differ in length, periodicity and severity (Bonal et al., 2016; Marengo et al., 2011; Meir et al., 2018). Finally, ecophysiological responses to drought occur on a multitude of spatial and temporal scales. These responses range from the almost instant closure of the stomata on a single leaf, to large scale tree
mortality that has persistent effects on many ecosystem processes (Brando et al., 2008; Rowland et al., 2015b, 2015a). Currently, there is no quantitative overview or understanding of how neotropical forests respond to different intensities of drought, from the leaf level up to the entire ecosystem. Below we formulate three key issues that guide our meta-analysis.

## 1.1 What type of droughts occur in neotropical forests?

Here, we differentiate three types of drought that differ in periodicity and severity: seasonal drought, episodic drought and
multi-year drought. Seasonality in precipitation is widespread in neotropical forests. Tropical humid forests loose roughly 100 mm of water every month through evapotranspiration (da Rocha et al., 2004; Shuttleworth, 1988). Months receiving less than 100 mm of rainfall will thus result in a precipitation deficit, these months are generally referred to as dry season months (Aragão et al., 2007; Sombroek, 2001). Seasonal droughts are by definition periodic and trees are generally found to be adapted to such a seasonal decline in precipitation (Brando et al., 2010; Goulden et al., 2004; Hutyra et al., 2007). Episodic droughts,
on the other hand, are caused by anomalous climatic conditions, primarily those imposed by strong El Nino Southern Oscillations (ENSO) and tropical North Atlantic sea surface temperature anomalies (Marengo et al., 2011). Episodic droughts often coincide with record breaking air temperatures and high vapour pressure deficits (Jiménez-Muñoz et al., 2016; Lee et al., 2013; Panisset et al., 2017). Elevated temperature and evaporative demand can amplify drought conditions through increased evapotranspiration and heat stress, resulting in so-called "global-change type droughts" (Allen et al., 2015; Breshears
et al., 2013).

Multi-year droughts are defined as a more permanent reduction of precipitation spanning years to decades, as projected by some climate model simulations (Boisier et al., 2015; Malhi et al., 2009). Long term records of river discharge and oxygen



isotopes in tree rings indicate that neotropical forests experienced several multi-year droughts in the 20[th] century, notably in
the 1960s (Brienen et al., 2012; Marengo et al., 2011; Richey et al., 1989). To date, the effect of prolonged rainfall reduction
on leaf, tree and ecosystem functioning have only experimentally been assessed in two throughfall exclusion experiments at
Tapajós and Caxiuanã in the eastern Amazon (Fisher et al., 2006; Meir et al., 2009; Nepstad, 2002). The results from the
Tapajós and Caxiuanã experiments have been previously synthesised (e.g. Meir *et al.*, 2009, 2018; da Costa *et al.*, 2010a) and
much of our knowledge about leaf, tree and ecosystem scale responses to multi-year droughts in tropical forests originates
from these experiments. Therefore, and because of the low number of replicates (i.e., 2) of such experiments, this meta-analysis
will focus only on the effects of seasonal and episodic drought on leaf, tree and ecosystem functioning.

**1.2 How is drought impacting leaf, tree and ecosystem scale processes?**

On the leaf scale, seasonal and episodic drought are often found to result in a downregulation of stomatal conductance; the
ease by which $CO_2$ and water vapor can diffuse between the atmosphere and the leaf intercellular spaces through the stomates
(Hogan et al., 1995; Huc et al., 1994). The most recent evidence suggests that stomates close in response to a decline in leaf
water potential ($\psi_l$) (Buckley, 2019). Here, we focus specifically on how drought-induced changes in $\psi_l$, the water potential
gradient and the different conductance's along the hydraulic pathway are driving the observed drought-induced changes in
productivity and transpiration. During steady state transpiration, leaf transpiration is given by:

$$E = k_{sl}(\psi_s - \psi_l) \qquad (1)$$

where $E$ is the leaf transpiration rate, $k_{sl}$ the soil to leaf hydraulic conductance and $\psi_s$ is the soil water potential. Rewriting
equation 1 as:

$$\psi_l = \psi_s - \frac{E}{k_{sl}} \qquad (2)$$

shows that a drought-induced decline of $\psi_l$ can be a compound result of a decline in soil water potential, a reduction of soil to
leaf hydraulic conductance and increased leaf transpiration.

Stomatal downregulation does not only constrain potential transpiration from the leaf but also the diffusion of $CO_2$ into the
leaf, potentially limiting leaf photosynthesis. The decline of stomatal conductance in response to drought is often larger
compared to the decline in leaf photosynthesis, resulting in an increase of intrinsic water use efficiency (iWUE) (Bonal et al.,
2000a; Santos et al., 2018). It is unclear how leaf-scale processes respond to drought in neotropical humid forest, with some
studies reporting strong reductions in stomatal conductance, leaf transpiration and photosynthesis during seasonal and episodic
drought (e.g. Hogan et al., 1995a; Huc et al., 1994; Sendall et al., 2009; Wolfe et al., 2016) while others report no significant
change in stomatal conductance and photosynthesis and even an increase of leaf transpiration (e.g. Allen and Pearcy, 2000;
Domingues et al., 2014; Fisher et al., 2006).





Leaf scale responses to drought can propagate to the tree scale, with reduced growth of the stem and new leaves, increased leaf shedding and litter fall and reduced tree daily transpiration (Brum et al., 2018; Doughty et al., 2015a; Fontes et al., 2018; Hofhansl et al., 2014; Phillips et al., 2009). Furthermore, the combined drought response of all individual trees in the ecosystem contributes to the observed ecosystem scale response to drought. Reduced leaf photosynthesis and leaf and stem growth can

result in a decline of gross ecosystem productivity (GEP) and aboveground net primary productivity (ANPP) while reduced tree transpiration might result in a decline of ecosystem evapotranspiration. Moreover, increased leaf litterfall in response to drought can boost microbial respiration and result in an increase of ecosystem respiration ($R_{eco}$) (Sayer et al., 2007). Next to the vegetation response to drought, a drought response of the microbial community as a result of reductions in soil moisture, atmospheric moisture and increased temperatures can contribute to the ecosystem scale response to drought. For example, soil

respiration is limited by temperature and moisture in neotropical humid forests and is found to decline with a dry season decline in soil moisture (Chambers et al., 2004; Sotta et al., 2004; Zanchi et al., 2014). The integration and synthesis of the observed drought responses on these three key spatial scales has not been carried out but can act as a method to identify critical drought response mechanisms and highlight current knowledge gaps.

### 1.3 Can hydraulic behaviour explain differences in drought responses among species?

Different tree species show markedly different responses to drought, both on the leaf level (Bonal et al., 2000a; Domingues et al., 2014) and the individual tree level (Esquivel-Muelbert et al., 2017a, 2017b; Phillips et al., 2009). The magnitude and direction of observed drought-induced responses depend on the hydraulic behaviour of the particular species measured in that study (Bonal et al., 2000a; Fisher et al., 2006; Machado and Tyree, 1994). For example, species can adopt different drought avoiding and drought tolerating strategies (Volaire, 2018). Drought avoiding strategies aim to avoid a dangerous decline in $\psi_l$

that could initiate xylem embolism and thus damage the hydraulic pathway. Maintaining a stable high $\psi_l$ during drought can be achieved by strict stomatal control on leaf transpiration (Huc et al., 1994; Machado and Tyree, 1994), increasing deep soil water uptake (Bonal et al., 2000b; Brum et al., 2019), maintaining a high plant internal water storage and conductance (Tyree et al., 2003; Wolfe, 2017) and through leaf shedding (Wolfe et al., 2016). Conversely, the drought tolerant strategy implies that low leaf and xylem water potentials are tolerated without significant embolism-induced losses of hydraulic conductance

and leaf turgor (Maréchaux et al., 2015; Markesteijn et al., 2011a; Tyree et al., 2003). Related to the drought tolerant versus drought avoiding dichotomy is the concept of isohydric versus non-isohydric behavior, where isohydric species maintain a stable and high $\psi_l$ in response to soil drying (drought avoiding) while non-isohydric species lower $\psi_l$ in accordance with a decline in soil water potential (Martínez-Vilalta et al., 2014; Meinzer et al., 2016). It is unclear whether neotropical forest tree species are generally isohydric as suggested by remote sensing analysis (Konings and Gentine, 2017) or non-isohydric as has

been generally observed in situ (Domingues et al., 2014; Rundel and Becker, 1987; Tobin et al., 1999).

Tree hydraulic behavior is strongly dependent on the characteristics of the xylem sapwood (Janssen et al., 2019; Markesteijn et al., 2011b, 2011a; Meinzer et al., 2008b, 2008a; Wolfe, 2017). Drought tolerant species characterized by non-isohydric



behavior are generally found to have xylem that is highly resistant to embolism (Skelton et al., 2015; Vogt, 2001). Conversely,

drought avoiding species are able to buffer midday declines in xylem water potential by using water that is stored in the sapwood (i.e. capacitance) (Borchert, 1994; Machado and Tyree, 1994; Meinzer et al., 2008b). In this meta-analysis, we will use wood density as a proxy of hydraulic behavior and examine whether differences in study-averaged wood density explain the variability in observed leaf and tree scale responses to drought between different studies. In neotropical tree species, sapwood capacitance and conductivity decline while embolism resistance generally increases with increasing wood density

(De Guzman et al., 2017; Janssen et al., 2019; Meinzer et al., 2008b; Santiago et al., 2018). This suggests that low wood density species can be considered drought avoiders while high wood density tree species are characterized as drought tolerant. Wood density is often not functionally related to the specific hydraulic properties (conductivity, capacitance and embolism resistance) that are driving hydraulic behavior (Janssen et al., 2019; Lachenbruch and Mcculloh, 2014). Nonetheless, wood density is an easily interpretable and widely available plant trait and therefore a useful proxy to compare different studies in which more

specific hydraulic properties and traits were not measured.

Here, we present a meta-analysis of a new database of neotropical forest responses to seasonal and episodic drought on three key spatial scales: the leaf, tree and ecosystem scale. We focus specifically on impacts of drought on carbon and water interactions. The aim of this meta-analysis is to: 1) provide a benchmark of neotropical humid forests responses to seasonal

and episodic drought and identify inconsistencies when going from the leaf to the ecosystem scale. 2) identify differences and similarities between episodic and seasonal drought responses. And 3) explore the relationships between study-averaged wood density and the magnitude and direction of leaf and tree scale responses to seasonal and episodic drought.

## 2 Methods

### 2.1 Data collection

The data collection focussed on published observations from the lowland humid forest of the neotropics, roughly between 20° South to 20° North (Figure 1, a). We searched the literature present in the Web of Science between 1979 and 2019. This time frame matches the ERA5 reanalysis climate data time series that were used to obtain harmonized meta-data for the retrieved literature. Publications were archived in a database if it contained one of the following measures: stomatal conductance, leaf photosynthesis, leaf water potential, stem sap flux density, stem diameter increment, leaf flushing, leaf litterfall, ecosystem

evapotranspiration, gross ecosystem productivity, ecosystem respiration and net ecosystem productivity. For studies that reported at least one of these measures, the observed values were stored in a database containing the reported value, the location and the month and year in which the measurement took place. If possible, the leaf and tree scale measures of individual trees including genus and species name were stored in the database. Otherwise, site averages were used. Observations of ecosystem scale processes always consisted of site averages.





Using the site latitude and longitude, we created spatial points of all the site locations (Figure 1, a). These points could then be used to extract spatial data using the routine *extract* in the R package *raster* (Hijmans et al., 2019). The site biome was extracted from the *Terrestrial ecoregions of the world map* from the World Wildlife Fund (Olson et al., 2001). Sites that were not located in the "Tropical and subtropical moist broadleaf forest" biome were omitted from the meta-analysis. Furthermore, the site elevation was extracted from the *World digital elevation model ETOPO5* from the European Environmental Agency (2019).

All sites that were located at elevations higher than 1000 m a.s.l were regarded montane environments and were as such omitted from the meta-analysis. The final database used for the meta-analysis included observed drought responses from 138 published studies conducted across 229 sites in neotropical humid forests.

Monthly averaged values of soil water content, air temperature and dewpoint air temperature at 2 meter above the surface were

retrieved for all the sites from the ECMWF ERA5 reanalysis product from January 1979 to August 2019. Monthly averaged air temperature and dewpoint temperature at 12:00 p.m. were used to calculate midday vapor pressure deficit (VPD) following Buck (1981). Monthly integrated soil moisture over the entire soil profile was calculated as the weighted average of soil moisture content in the four soil layers (0 – 189 cm below the surface) provided in the ERA5 product. Integrated soil moisture was then used to estimate relative extractable soil water (REW) which is shown to capture both wet-dry season oscillations as

well as episodic droughts (Figure 1 d, see section 2.3).

**2.2 Data pre-processing and deriving additional measures**

From the collected leaf, tree and ecosystem performance measures we derived additional measures of transpiration, productivity and water use efficiency. On the leaf scale, midday leaf-area specific transpiration was calculated as the product of midday stomatal conductance and midday VPD derived from the ERA5 reanalysis data. Instantaneous intrinsic water use

efficiency (iWUE) and instantaneous actual water use efficiency (WUE) were calculated as the ratios between leaf-area specific photosynthetic rate and stomatal conductance or leaf transpiration at midday, respectively.

On the tree scale, in 13 out of 28 studies that reported sap flux density results, only the maximum midday sap flux density values were reported but not the integrated daily transpiration rates. For these studies, the integrated daily transpiration was

estimated from a significant linear relationship (RMSE = 3.25 kg dm$^{-2}$ day$^{-1}$) between daily maximum sap flux density and daily transpiration (Figure S1). Errors introduced by this approach were estimated to be less than 34% of the integrated daily transpiration rates. Furthermore, we calculated the instantaneous soil to leaf hydraulic conductance on a sapwood area basis following Love and Sperry (2018):

$$k_{sl} = \frac{J_{max}}{(\psi_{l\,pd} - \psi_{l\,md})} \qquad (3)$$



where $J_{max}$ is the daily maximum sap flux density, $\psi_{l\,md}$ is the midday leaf water potential and $\psi_{l\,pd}$ is the pre-dawn leaf water potential. Pre-dawn $\psi_l$ is measured before the onset of leaf transpiration and considered a proxy of $\psi_s$ in the rooting zone. The difference between midday $\psi_l$ and pre-dawn $\psi_l$ is the midday water potential gradient within the tree, from the root up to the canopy (see also Equation 1).


On the ecosystem scale, the above-ground net ecosystem productivity (ANPP) was calculated as the sum of biomass allocated to stem growth and biomass allocated to canopy growth (sensu Doughty et al., 2015a; Hofhansl et al., 2014a). Finally, the ecosystem water use efficiency was calculated as the ratio between gross ecosystem productivity (GEP) and ecosystem evapotranspiration (sensu Yang et al., 2016).

**2.3 Dry season and drought definition**

As the dry season progresses, soil moisture content, relative extractable soil water (REW) and soil water potential decline as daily evapotranspiration surpasses precipitation (see e.g. Wright *et al.*, 1992; Nepstad, 2002). The occurrence of rain during or at the end of the dry season generally results in a rapid increase of soil water potential and a relief of plant water stress (Fontes et al., 2018; Roberts et al., 1990; Tobin et al., 1999). Therefore, we define the dry season months as months in which
REW is reduced relative to the previous month (Figure 1 d). The REW is the amount of soil water available for plant uptake, which is often expressed as the volumetric soil moisture scaled between field capacity (REW = 1) and permanent wilting point (REW = 0). However, as there are insufficient measurements to construct reliable soil water retention curves across the study sites, we could not calculate REW. Instead, we estimated a pseudo REW as the normalized integrated soil moisture from ERA5, with 0 in the driest month and 1 in the wettest month of the entire timeseries (1979-2019) at that specific site.


Some months in the middle of the wet season with high REW showed small reductions in REW compared to the previous month. To exclude these months from the dry season, all months with REW higher than the 65% quantile of REW were labelled as wet season months despite a reduction in REW. Subsequently, dry season months where REW was lower than the 10% quantile of REW in all dry season months were labelled as episodic droughts (Figure 1 d). In three studies, additional months
that were not initially classified as episodic droughts but were considered exceptional dry months in that study were classified as episodic drought months. These months were September 2002 at Tapajós in Domingues et al. (2014), April 1977 at Barro Colorado Island in Fetcher (1979) and January, February and March 2009 in Hofhansl et al. (2014). In studies where the measurement period covered multiple months, we labelled time intervals that included at least three months of episodic drought as episodic drought and otherwise as dry season months for comparison (Figure 4). The subdivision resulted in 3326
observations in the wet season, 3006 in the dry season and 624 during episodic drought. Strong ENSO years (1996-1997, 2009-2010, 2015-2016) are clearly visible as years with many recorded episodic drought months across the 229 sites (Figure 1).





## 2.4 Meta-analysis

Quantitative drought responses of different plant physiological and ecosystem scale processes were synthesized using meta-
analytical statistics. The log response ratio was used as a metric of drought effect size and converted back to percentage change
for convenient interpretation. The log response ratio is the natural-log proportional difference between the means in a treatment
and a control group (Hedges et al., 1999; Lajeunesse, 2011). Measurements were often available in pairs or as repeated
measurements (wet season-dry season, dry season-episodic drought), so that the variance of the calculated response ratio has
to be adjusted for by the Pearson product correlation coefficient between the measurement pairs (Lajeunesse, 2011). For
individual tree measurements, which were available for stomatal conductance, photosynthesis, leaf water potential, tree
transpiration and sometimes leaf flushing, the average, standard deviation and correlation coefficient were calculated from the
pool of measured trees in each study. When site averages were used, which was the case for all the other measures, the average
and standard deviation calculated from the different measurement years were used. The log response ratio and sample variance
of the measures in individual studies and sites were calculated using the *escalc* routine and the mean effect sizes and 95%
confidence intervals in the *rma* routine, both available in the R package *metafor* (Viechtbauer, 2017).

The variability in magnitude and direction of leaf and tree scale responses to drought were related to the average wood density
of the species measured in the different studies. To calculate the average wood density for each study, we created a separate
dataset including for each study the genus and species names of the individual trees measured in the study. Preferably, the
species-specific wood density was retrieved from the original source. However, if this was not possible, we retrieved wood
density from a database of wood properties in neotropical tree taxa collated previously by us (Janssen et al., 2019) or from the
global wood density database (Chave et al., 2009b; Zanne et al., 2009). If species-specific wood density was not available, the
genus averaged wood density was used instead. Study averaged wood density was used in the *rma* routine from the R package
*metafor* (Viechtbauer, 2017) to test whether wood density was a significant moderator variable in the mixed-effect meta-
regression model.

## 3 Results

### 3.1 Responses to seasonal drought

The meta-analysis shows that across the measured neotropical forest sites, a dry season decline in relative extractable water
(REW) is associated with a decline of soil matric potential in the topsoil (Figure 2 a, c). Furthermore, dry season days are often
less cloudy resulting in higher net radiation, higher air temperature, lower relative humidity and therefore higher vapor pressure
deficit (VPD) compared to the wet season (Figure 1 d, Figure 2 d). As a result of a decline in water supply from the soil and
the increase of evaporative demand from the atmosphere, the meta-analysis indicates that across studies, pre-dawn and midday
$\psi_l$ both significantly decline from the wet to the dry season (Figure 2 a, Figure 3, a). Pre-dawn $\psi_l$ declines from an average -



0.22 ±0.1 MPa in the wet season to -0.34 ±0.21 MPa in the dry season among studies and sites. Midday $\psi_l$ declines from an
average -0.91 ±0.35 MPa to -1.32 ±0.41 MPa from the wet to the dry season (Figure 2 a). Therefore, the average midday water
potential gradient increases from 0.69 MPa in the wet season to 0.98 MPa in the dry season (Figure 3 b).

The dry season decline of $\psi_l$ triggers stomatal closure resulting in a decline of stomatal conductance and leaf photosynthesis
of 44% and 19% from the wet to the dry season, respectively. However, the data shows no significant decline in leaf
transpiration from the wet to the dry season (Random-effects model, p = 0.46, n = 24) as the average increase of VPD from
the wet to the dry season is of the same magnitude as the decline of stomatal conductance (Figure 2 b, d). As the decline in
stomatal conductance outweighs the decline in leaf photosynthesis from the wet to the dry season, intrinsic water use efficiency
(iWUE) increases by 31% from the wet to the dry season (Figure 3, a). Nonetheless, as leaf transpiration is sustained in the
dry season, actual water use efficiency (WUE) declines by 19% from the wet to the dry season (Figure 3, a). These results
suggest that on the leaf scale, increased leaf transpiration and a drop in $\psi_l$ in the dry season are largely prevented by
downregulating stomatal conductance, also likely contributing to a decline of leaf photosynthesis.

The meta-analysis shows that on the tree scale, the average water potential gradient is increased while soil to leaf hydraulic
conductance is reduced from the wet to the dry season (Figure 3 b). As the increased gradient and reduced conductance are
cancelling each other out, there is on average no significant change in daily tree transpiration from the wet to the dry season
(Figure 3, b). This is in agreement with the observed sustained dry season transpiration on the leaf scale (Figure 2 a). We
observe a distinct seasonality of alternating stem and canopy growth (Figure 3, b). The shedding of old and flushing of new
leaves during the dry season cumulates into an average 33% higher leaf litterfall and 32% higher leaf flushing in the dry season,
compared to the wet season. While canopy productivity increases, average stem diameter growth declines by roughly the same
magnitude (34%) from the wet to the dry season (Figure 3 b). These results suggest that generally, above-ground productivity
alternates between stem and canopy growth from the wet to the dry season.

Despite no observed changes in leaf or tree transpiration, we observed a significant 12% increase of ecosystem
evapotranspiration from the wet to the dry season (Figure 3 c). Furthermore, the meta-analysis showed a 9% decline in gross
ecosystem productivity (GEP) from the wet to the dry season (Figure 3 a). However, the decline of GEP is cancelled out by a
dry season decline of ecosystem respiration ($R_{eco}$) resulting in no significant change in net ecosystem productivity (NEP =
GEP – $R_{eco}$) from the wet season to the dry season. There was also no significant change in above-ground net primary
productivity (ANPP) from the wet to the dry season (Figure 3 c). As GEP is reduced and evapotranspiration increased,
ecosystem water-use efficiency is significantly reduced by 19% from the wet to the dry season (Figure 3 c), which is of a
similar magnitude as the dry season reduction of WUE on the leaf scale.



## 3.2 Responses to episodic drought

We found that on average, the number of months per year classified as episodic drought have been increasing since 1979
(Figure 1 b) driven by a multi-decade decline in dry season soil moisture across the sites. Several previously described El Niño
related drought events in 1983, 1987, 1997, 2010 and 2015 are clearly visible as years with relatively many episodic drought
months (Figure 1 b, d, e). Episodic droughts are associated with a higher VPD and a lower $\psi_s$ compared to a regular dry season
(Figure 1 d, Figure 2 a, d). Consequently, the pre-dawn $\psi_l$ is on average 0.39 MPa lower (-0.73 ±0.59 MPa) during episodic
drought compared to an average dry season (-0.34 ±0.21 MPa) (Figure 2 a, Figure 4 a). Midday $\psi_l$ declines from -1.32 ±0.41
MPa in the dry season to -1.95 ±0.55 MPa during episodic drought, increasing the average water potential gradient by 0.24
MPa across all measured trees. However, the meta-analysis indicates that this increase is not significant as there is a large
variability in the water potential gradient response (Figure 4 b).

The decline of midday $\psi_l$ in response to episodic drought is related to a 42% reduction of stomatal conductance and 25%
reduction in leaf photosynthesis compared to a regular dry season (Figure 4 a). Contrary to seasonal drought, leaf scale
transpiration is reduced by an average 30% during episodic drought compared to a regular dry season (Figure 4 a). This
suggests that stomatal downregulation generally outweighs the increased evaporative demand from the atmosphere during
episodic drought, effectively limiting leaf transpiration. Furthermore, as stomatal conductance shows a larger magnitude
decline in response to episodic drought compared to leaf photosynthesis, WUEi is significantly higher during episodic drought
compared to a regular dry season (Figure 4 a). However, leaf transpiration and leaf photosynthesis decline with a similar
magnitude in response to episodic drought, so there is no significant change in WUE observed (Figure 4 a).

On the tree scale, we observe a substantial decline of 52% in soil to leaf hydraulic conductance during episodic drought
compared to a regular dry season, associated with an average 21% reduction of daily transpiration (Figure 4 b). Furthermore,
stem diameter growth is reduced by an average 6% during episodic drought compared to a regular dry season while leaf
flushing and litter fall are not significantly different (Figure 4 d). On the ecosystem scale, we observe that despite the decline
in tree transpiration, the meta-analysis suggests that the observed 4% decline in evapotranspiration is not significantly different
to the evapotranspiration in the dry season (*Random-effects model, p = 0.47, n = 5*). Furthermore, there is a small and no
significant decline of $R_{eco}$ (-4%),  while NEP (+16%), ANPP (+6%) and GEP (6%) show small and no significant increases in
response to episodic drought (Figure 4 c). This results in a marginal significant increase of 13% in ecosystem water use
efficiency (Figure 4 c).



### 3.3 Relationships between study-averaged wood density and drought responses

The meta-analysis revealed a consistent downregulation of stomatal conductance and leaf photosynthesis as pre-dawn and midday $\psi_l$ decline during seasonal and episodic drought. However, the magnitude of these leaf-scale responses to seasonal and episodic drought varies substantially among different studies. We find that between-study variation in the stomatal conductance, leaf transpiration, leaf photosynthesis and midday $\psi_l$ response to seasonal and episodic drought correlates with differences in study-averaged wood density. Generally, studies that measured mainly low wood density tree species showed a

stronger response of stomatal conductance, leaf transpiration and leaf photosynthesis to seasonal and episodic drought compared to studies that measured mainly high wood density species (Figure 5). Furthermore, the effects of study averaged wood density on the response of midday $\psi_l$ were less clear but generally the magnitude of the response increased with wood density (Figure 5 d).

We find that the response of midday $\psi_l$ to a decline in pre-dawn $\psi_l$ is also strongly dependent on study-averaged wood density (Figure S2). Tree species from studies with a high average wood density (> 0.7 g cm$^{-3}$) showed a strong reduction in midday $\psi_l$ and increase the water potential gradient in response to a decline in pre-dawn $\psi_l$, which is in accordance with the definition of extreme non-isohydric behavior (sensu Martínez-Vilalta et al., 2014). On the other hand, tree species in studies with a low average wood density species (< 0.5 g cm$^{-3}$) are characterized by partly isohydric behavior as they show a decline in the water

potential gradient in response to a decline in pre-dawn $\psi_l$. Studies with intermediate average wood density (0.5-0.7 g cm$^{-3}$) show an intermediate response with midday $\psi_l$ declining parallel to a decline in pre-dawn $\psi_l$ (slope ~1) suggesting strictly non-isohydric behavior (Figure S2). Related to these results we found that the stomatal response to VPD also depends on study-averaged wood density, with low wood density species showing strong stomatal downregulation in response to increased VPD, while no stomatal downregulation is observed in high wood density species (Figure S3). These results imply that low

wood density species prevent a midday drop in $\psi_l$ during seasonal and episodic drought by downregulating stomatal conductance, leaf transpiration and photosynthesis in response to elevated VPD, while high wood density tree species keep a more variable $\psi_l$ and have no strong stomatal control on leaf transpiration.

The two tree-scale measures for which enough species-specific data was available, tree daily transpiration and leaf flushing,

also showed significant relationships with study-averaged wood density (Figure 6). The relationship between study-averaged wood density and the magnitude of the seasonal drought response of tree daily transpiration was similar as the relationship with leaf transpiration (Figure 5 b, Figure 6 a). Roughly half of the studies that measured mainly low wood density species showed a dry season decline in tree transpiration. The other half of the studies that measured mainly high wood density species showed a dry season increase of tree transpiration (Figure 6 a). Similarly, dry season leaf flushing is found to be more

pronounced in high wood density species compared to low wood density species that actually show on average a decline of leaf flushing in the dry season (Figure 6 b).



## 4 Discussion

### 4.1 How do leaf, tree and ecosystem scale processes respond to seasonal drought?

Stomatal behaviour and changes in soil to leaf hydraulic conductance determine the hydrological response to seasonal drought
in neotropical trees, driving tree transpiration and ecosystem evapotranspiration (Figure 2 & 3). The downregulation of
stomatal conductance and soil to leaf hydraulic conductance in the dry season is a widely observed hydrological response to a
decline in leaf and xylem water potential (Fisher et al., 2006; Machado and Tyree, 1994; Williams et al., 1998). Stomatal
closure is cancelled out by a higher midday VPD in the dry season resulting in no observed change in leaf transpiration from
the wet to the dry season (Figure 3 a). Similarly, the dry season decline in hydraulic conductance is cancelled out by an increase
of the water potential gradient (midday $\psi_l$ - pre-dawn $\psi_l$) within the tree, resulting in no average change in tree daily
transpiration from the wet to the dry season (Figure 3). The decline of soil to leaf hydraulic conductance in the dry season is
the result of embolism formation in the xylem vessels that reduces xylem hydraulic conductance (Bonal et al., 2000a; Fontes
et al., 2018; Machado and Tyree, 1994; Meinzer et al., 2008b). Our data did not allow to disentangle whether dry season
transpiration is mainly constrained by a decline in stomatal conductance or a decline in soil to leaf hydraulic conductance.
However, the decline of hydraulic conductance and stomatal conductance with decreasing xylem water potential are strikingly
similar (Brodribb et al., 2003) suggesting that xylem hydraulic vulnerability and stomatal sensitivity are strongly coordinated
(Fontes et al., 2018; Maréchaux et al., 2018; Meinzer et al., 2008b).

The meta-analysis suggests that the dry season downregulation of stomatal conductance is accompanied by a smaller but
significant decline in leaf photosynthesis (Figure 3 a). Therefore, the leaf-scale intrinsic water use efficiency (WUEi) increases
on average from the wet to the dry season (Figure 3 a). This increase of WUEi in the dry season was also found in earlier site-
specific studies (Bonal et al., 2000a; Hogan et al., 1995; Santos et al., 2018). We also find that sustained leaf transpiration in
the dry season is resulting in a decline of actual leaf water use efficiency (WUE) from the wet to the dry season (Figure 3 a).
Furthermore, as gross ecosystem productivity (GEP) declines and evapotranspiration increases, we also observe a decline of
ecosystem water use efficiency from the wet to the dry season (Figure 3 c). Therefore, our results suggests that despite a decline
in stomatal conductance, neotropical forests become less water efficient in the dry season. This is in agreement with a global
synthesis of eddy-covariance measurements that showed that humid tropical forests show a decline of ecosystem water use
efficiency in response to drought (Yang et al., 2018).

Our results show that across neotropical forests, above-ground net primary productivity (ANPP) does not change from the wet
to the dry season (Figure 3 c). This suggests that growth is shifted from the stem in the wet season to the canopy in the dry
season (Figure 3 b) without changes in overall above-ground growth. Furthermore, our meta-analysis suggests that on average
GEP is reduced in the dry season compared to the wet season (Figure 3 c) indicating that above-ground carbon use efficiency
(ANPP / GPP) is increased in the dry season. The increase of carbon use efficiency might be explained by a relative dry season





decline of autotrophic respiration (Doughty et al., 2015b; Rowland et al., 2014) that is driven by a decline in stem respiration (del Aguila-Pasquel et al., 2014; Nepstad, 2002). Alternatively, a dry season decline of carbon allocation towards the roots might explain why a decline in GEP does not result in an apparent decline of above-ground growth (Girardin et al., 2016).

Another explanation is that in the dry season, stored non-structural carbohydrates (NSC's) are utilized for canopy growth.
Previous work suggests that NSC concentrations in neotropical trees are not reduced in response to seasonal and episodic drought (Dickman et al., 2019; Würth et al., 2005). This implies that temporary reductions in photosynthesis and gross productivity are not sufficient to limit actual tree growth in the dry season (Würth et al., 2005). In summary, the observed seasonal drought responses related to stomatal and hydraulic conductance, transpiration and photosynthesis occur on relatively short time-scales of hours and days and are therefore adequately captured by our approach. However, tree and ecosystem scale
responses related to productivity, carbohydrate status and growth allocation operate on seasonal to multi-annual timescales which do not always correspond with the observations on the leaf scale and might not be fully captured by our meta-analysis.

**4.2 How do leaf, tree and ecosystem scale processes respond to episodic drought?**

Episodic droughts seem to have become more common in South and Central America recently. Previously classified as once in a century episodic droughts are now occurring roughly every five years (Aragão et al., 2007; Coelho et al., 2012; Erfian
et al., 2017; Marengo et al., 2008, 2011; Panisset et al., 2017). Following our definition of episodic drought, we observe a significant increase of episodic drought occurrence across the 229 neotropical forest sites since 1979 (Figure 1 b). This result is in agreement with the analysis of alternative datasets indicating that dry seasons in Amazonia have been becoming dryer since 1979 (Fu et al., 2013). The mechanisms driving this dry season drying are uncertain but have been attributed to changes in global atmospheric circulation (Fu et al., 2013) and more regionally to deforestation (Costa and Pires, 2010; Debortoli et
al., 2017). Furthermore, ENSO swings that are clearly linked to major droughts in neotropical forests (Figure 1) have been intensifying in the 20[th] and 21[st] century (Grothe et al., 2019). In this meta-analysis, we were able to use leaf, tree and ecosystem scale data from five major episodic drought years, namely from 1987, 1997, 2005, 2010 and 2015 (Figure 1).

Episodic droughts reduce the supply of water from the soil and increase the evaporative demand of the atmosphere beyond
values that are observed in a regular dry season (Figure 2) (see also Jiménez-Muñoz et al., 2016; Lee et al., 2013; Panisset et al., 2017). We find that both stomatal conductance, leaf transpiration and photosynthesis are reduced and WUEi is increased during episodic drought compared to a regular dry season (Figure 4 a). This suggests that the physiological responses to episodic drought on the leaf level are, in terms of direction and magnitude, a continuation of the seasonal drought responses observed. Stomatal limitations may explain the observed decline of leaf photosynthesis, as changes in nutrient or chlorophyll
concentrations were not reported for the 2015 drought in the central Amazon (Santos et al., 2018). Alternatively, reductions in carboxylation capacity and mesophyll conductance in response to leaf desiccation or high leaf temperatures could cause a more permanent reduction of photosynthesis during episodic drought (Dewar et al., 2018; Doughty, 2011; Felsemburgh, 2009; Lloyd





and Farquhar, 2008; Zhou et al., 2013). The average midday $\psi_l$ observed during episodic drought (-1.95 MPa) induces leaf turgor loss in many tropical rainforest trees (Maréchaux et al., 2015). The importance of tissue desiccation and heat-induced

damage to the photosynthetic machinery is presently not known but could become increasingly important in the tropical carbon cycle in a warmer climate.

We observe reductions of stem growth while leaf litter fall and leaf flushing do not show a consistent positive or negative change in response to episodic drought (Figure 3 b). The decline of stem growth during episodic drought is widely observed

across tropical humid forests and has been linked to a temporary decline in tropical forest carbon sink (Brienen et al., 2015; Clark et al., 2003, 2018; Feldpausch et al., 2016; Rifai et al., 2018). However, declines in stem growth are not always obvious (Doughty et al., 2014, 2015a; Phillips et al., 2009) and are at some sites compensated for by an increase in canopy growth (Doughty et al., 2015a; Hofhansl et al., 2014) resulting in no observed net change in ANPP during episodic drought (Figure 3 c). The observed reduction of stem growth is not likely related to reductions in carbohydrate availability (carbon starvation)

(Mcdowell et al., 2008; Sala et al., 2012) as recent evidence from neotropical humid forests suggests that leaf and wood tissue concentrations of NSCs are kept relatively constant during severe episodic drought (Dickman et al., 2019). Alternatively, stem growth is limited by cell turgor loss in the vascular cambium as a result of tissue desiccation, which limits cell formation and thus the formation of new tissue in the stem (Körner and Basel, 2013; Krepkowski et al., 2011; Muller et al., 2011). It is essential to understand which mechanisms, turgor mediated, carbon mediated, or a combination of both, are driving drought-

induced declines in stem growth, as they can operate on different time scales and can have different sensitivities to drought.

### 4.3 What are the differences between seasonal and episodic drought?

We find that the responses of stomatal conductance, leaf photosynthesis, midday and pre-dawn $\psi_l$ to episodic drought are basically a continuation of the same leaf physiological responses observed during seasonal drought (Figure 3 & 4). However, unlike seasonal drought, the decline in stomatal conductance outweighs the increase of VPD during episodic drought,

effectively reducing leaf transpiration. Similarly, the additional embolism-induced reduction of soil to leaf hydraulic conductance outweighs the increase of the water potential gradient within the tree, causing a reduction of tree daily transpiration during episodic drought compared to a regular dry season (Figure 4 b). Our results are in agreement with site-specific observations that tree daily transpiration is reduced through a combination of stomatal downregulation and a loss of soil to leaf hydraulic conductance, both in response to episodic drought (Fontes et al., 2018) and multi-year drought (Fisher et

al., 2006). Unlike the rapid recovery of stomatal conductance, soil to leaf hydraulic conductance has been observed not to recover fully after episodic drought (Fontes et al., 2018) imposing a legacy effect on transpiration in the first months following episodic drought. Furthermore, the loss of hydraulic conductance might be considered an early warning signal for embolism-induced drought mortality (Rowland et al., 2015b) following episodic drought (Feldpausch et al., 2016; Phillips et al., 2009).





Contrary to seasonal drought, we observe no increase in leaf flushing and litterfall and no significant declines in $R_{eco}$ and GEP during episodic drought. One explanation for this apparent discrepancy is that leaf flushing, litterfall and GEP operate on seasonal timescales and are strongly dependent on tree phenology. Most neotropical tree species shed old and flush new leaves during the dry season as their leaf phenology is synchronized to maximum daily insolation (Borchert et al., 2015; Bradley et al., 2011; Brando et al., 2010; Graham et al., 2003; Wagner et al., 2016; Wright and van Schaik, 1994). This results in an initial

decline followed by a progressive increase of photosynthetic capacity on the ecosystem scale in the late dry season as leaves mature (Albert et al., 2018; Doughty and Goulden, 2009; Wu et al., 2016). Leaf flush and maturation, and with it the increase of leaf photosynthetic capacity, drive a progressive increase of GEP during the dry season in humid neotropical forests (Albert et al., 2018; Araújo et al., 2016; Doughty and Goulden, 2009; Hutyra et al., 2007; Restrepo-Coupe et al., 2013). Episodic droughts by our definition always occur at the end of the dry season, when REW is lowest (Figure 1). Therefore, the peaks in

litter fall and leaf flush that generally occur in the first half of the dry season, have already occurred before the episodic drought starts and therefore GEP is relatively high. We hypothesize that the seasonal timescales of tree phenology and ecosystem productivity could be counteracting the potential negative effects of short episodic droughts on GEP, which were therefore not observed in the meta-analysis.

### 4.4 What are observed inconsistencies in leaf, tree and ecosystem scale responses?

Our meta-analysis indicates a general tendency of seasonal and episodic drought responses becoming smaller and not significant when going from the leaf and tree scale to the ecosystem scale. Regarding transpiration, we observed sustained leaf and tree scale transpiration in the dry season (Figure 3) and a decline of leaf and tree scale transpiration in response to episodic drought (Figure 4). In contrast, ecosystem evapotranspiration increases in the dry season (Figure 3 c) and does not significantly change during episodic drought (Figure 4 c). This discrepancy is not logically explained by an increased contribution of

evaporation from the soil and canopy to evapotranspiration, as both soil and canopy evaporation are expected to be lower in the dry season and during episodic drought compared to the wet season (Shuttleworth, 1988).

A more likely explanation is that the leaf and tree-scale data used in our meta-analysis are biased towards fast-growing pioneer tree species with low wood density that are growing in upper canopy positions (e.g. Dünisch and Morais, 2002; Huc et al.,

1994; Kunert et al., 2010; Machado and Tyree, 1994). Stomatal control on transpiration is stronger in low wood density compared to high wood density tree species (Figure 5 & 6, Figure S2). Furthermore, sun-exposed trees in upper canopy positions experience a higher evaporative demand from the atmosphere, resulting in a more pronounced downregulation of stomatal conductance and photosynthesis in response to seasonal and episodic drought compared to understory trees (Domingues et al., 2014; Fisher et al., 2006; Santos et al., 2018). This sample bias in the meta-analysis might also explain why

ecosystem scale responses of water and carbon exchange to episodic drought seem to contradict the observations on the leaf and tree scale. Leaf photosynthesis and stem growth are observed to significantly decline while GEP and ANPP show a small and not significant increase in response to episodic drought (Figure 4). This meta-analysis result is confirmed by unexpected





results from site-specific studies that found that GEP and ANPP are not reduced during episodic drought (Bonal et al., 2008) despite significant declines of leaf photosynthesis (Doughty et al., 2014).


Another explanations for the apparent contradiction between leaf, tree and ecosystem scale responses to episodic drought is the limited timescale on which we analysed ecosystem drought responses. The temporal scale of some tree and ecosystem scale responses to episodic drought might extend far beyond the actual drought (e.g. Gonçalves et al., 2020; Hofhansl et al., 2014). For example, episodic drought events have been found to elevate tree mortality rates across neotropical forests (Condit

et al., 1995; Feldpausch et al., 2016; Phillips et al., 2009; Williamson et al., 2000). Tree mortality can significantly impact ecosystem productivity and transpiration, carbon storage and canopy structure, impacting the understory light environment and microclimate for many years (da Costa et al., 2018; Leitold et al., 2018; Rice et al., 2004, 2008; Rowland et al., 2018; Saatchi et al., 2013; Yang et al., 2018). Furthermore, extensive leaf flushing in the first months after an episodic drought have been reported (Doughty et al., 2014, 2015a; Gonçalves et al., 2020; Hofhansl et al., 2014) contributing to ANPP exceeding

pre-drought values in the years directly following episodic drought (Doughty et al., 2014, 2015a; Hofhansl et al., 2014). These legacy effects of drought are not captured by or meta-analysis, which is a limitation of the method used. Therefore, we were unable to grasp the complete, or final extent of the tree and ecosystem scale responses to episodic drought.

### 4.5 How is wood density related to leaf and tree scale responses to drought?

The meta-analysis shows that the magnitude and direction of the stomatal conductance, leaf and tree-scale transpiration and
leaf flushing response to seasonal and episodic drought is strongly related to the wood density of trees measured in a particular study (Figure 5 & 6). Generally, we find that studies that measured tree species with a relatively low wood density showed a more isohydric and drought avoiding response, including strong stomatal control on transpiration and no dry season leaf flushing (Figure 5 & 6). Conversely, studies that measured tree species with a relatively high wood density showed no stomatal downregulation, increased leaf and tree-scale transpiration and increased leaf flushing in the dry season (Figure 5 & 6). As a
result, high wood density trees show a stronger desiccation of the leaves and stem during drought and a lower midday leaf and xylem water potential (Figure 5 c, d) (Borchert, 1994; De Guzman et al., 2017; Meinzer et al., 2008b; Sterck et al., 2014). Wood density appears a good proxy of hydraulic behaviour and could well be used to predict responses of stomatal conductance, transpiration and leaf flushing to seasonal and episodic drought (see e.g. Christoffersen et al., 2016).

Differences in wood density among tree species have been widely studied and are linked to differences in plant hydraulic properties such as hydraulic conductance, sapwood capacitance and embolism resistance (Baas et al., 2004; Chave et al., 2009a; Janssen et al., 2019; Poorter et al., 2010). The use of wood density as a proxy of more fundamental hydraulic properties has been criticized as it often lacks a functional basis (Lachenbruch and Mcculloh, 2014; Patiño et al., 2012). Sapwood capacitance, the amount of water released from the xylem under a certain pressure, is arguably the only hydraulic property that is
functionally related to wood density, as the amount of space available for water storage in the wood scales inversely with wood





density (Janssen et al., 2019; Meinzer et al., 2008b; Pratt and Jacobsen, 2017; Ziemińska et al., 2019). Sapwood capacitance is positively related to maximum stomatal conductance, transpiration, soil to leaf hydraulic conductance and midday $\psi_l$ (Meinzer et al., 2003; Oliva Carrasco et al., 2015). We show that these relationships hold when relating not species but study-averaged wood density, as a proxy of sapwood capacitance, to study-averaged stomatal conductance, daily transpiration, midday $\psi_l$ and soil to leaf hydraulic conductance (Figure S4). Our results suggest that wood density, via sapwood capacitance, is largely driving the magnitude of the stomatal and transpiration response to seasonal and episodic drought in neotropical trees.

The difference in hydraulic behaviour between low and high wood density tree species is confirmed by the observation that the decline of stomatal conductance with VPD and the slope of the relationship between midday $\psi_l$ and pre-dawn $\psi_l$ are strongly dependent on wood density (Figure S2 & S3). We find that low wood density trees with high sapwood capacitance show a relatively high soil to leaf hydraulic conductance as stored water is used for transpiration (Figure S4) while stomatal conductance is downregulated with increasing VPD to avoid dehydration (Figure S3) (Goldstein et al., 1998; Meinzer et al., 2004, 2008b). Conversely, in high wood density trees, transpiration is primarily constrained by the relatively low soil to leaf hydraulic conductance all year around and stomatal downregulation plays a minor role. High wood density trees maintain stomatal conductance (0.07 – 0.14 mol m$^{-2}$ s$^{-1}$) even during severe episodic drought (Alexandre, 1991; Bonal et al., 2000a; Roberts et al., 1990; Santos et al., 2018; Stahl et al., 2013b). This implies that transpiration has to increase during seasonal and episodic drought in high wood density trees, resulting in a significant decline of midday $\psi_l$ (Figure S2) (Alexandre, 1991; Bonal et al., 2000a; Brum et al., 2019; Domingues et al., 2014).

These results present a contradiction to remote sensing data that suggests that neotropical humid forests are strictly isohydric (Konings and Gentine, 2017). At least at the leaf level, most neotropical trees in this meta-analysis and especially high wood density trees show non-isohydric behaviour (Figure S2). Following the definition of Martinez-Vilalta et al. (2014): the water potential gradient generally increases with a decline in pre-dawn $\psi_l$ in response to seasonal and episodic drought (Figure 3 & 4, Figure S2). The observed insensitivity of stomatal conductance to VPD in high wood density trees has been reported previously for lowland rainforest species (Bonal et al., 2000a; Domingues et al., 2014; Granier et al., 1992; Huc et al., 1994) and for tree species of tropical montane cloud forest (Rada et al., 2009). Stomatal insensitivity to VPD is a possible adaptation to surviving in a humid and deeply shaded understory, as the $CO_2$ concentration inside the leaf is kept high to maximize photosynthesis during brief moments of high irradiance, known as sun flecks (Domingues et al., 2014; Pons et al., 2005; Tinoco-Ojanguren and Pearcy, 1992).

The capability to maintain stomatal conductance and transpiration during short episodic droughts has been explained by the uptake of deep soil moisture using tap roots (Bonal et al., 2000a; Brum et al., 2019; Meinzer et al., 1999; Nepstad et al., 1994; Stahl et al., 2013a, 2013b). Soil water at a depth of up to 18 meters was found to be accessible for trees at Tapajós in the eastern





Amazon (Davidson et al., 2011), enabling trees to maintain a favourable water status during short dry periods. This also becomes clear from the relatively high average pre-dawn $\psi_l$ during episodic drought (-0.73 MPa), compared to tree species of tropical dry forest where pre-dawn $\psi_l$ can approach -2.5 MPa in a regular dry season, inducing leaf wilting and high mortality rates in tree seedlings (Sobrado, 1986; Veenendaal et al., 1996). Soil depth, root functioning and differences in root architecture are believed to be crucial regulators during drought (Brum et al., 2019; Meinzer et al., 1999; Stahl et al., 2013a), but lack of

data in neotropical forests prevented us from including these traits in our meta-analysis.

Deep soil moisture uptake is not always sufficient to maintain a favourable water status within the tree as drought-induced tree mortality events have been widely observed across the neotropics (Condit et al., 1995; Feldpausch et al., 2016; Phillips et al., 2009; Williamson et al., 2000), likely resulting from hydraulic failure (Rowland et al., 2015b). The effect of an increased

evaporative demand during drought should not be overlooked, as a high VPD can trigger xylem embolism in trees even when soil water is still easily accessed (Fontes et al., 2018; Phillips et al., 2001). Moreover, our results point to the lack of drought avoidance in high wood density tree species as stomatal conductance and transpiration are sustained under high evaporative demand, resulting in a strong decline of xylem and leaf water potential during drought (Figure 5 & 6, Figure S2). However, many high wood density tree species in humid neotropical forests have evolved in permanently wet environments are not

always tolerant against xylem embolism (Janssen et al., 2019; Powell et al., 2017; Santiago et al., 2018). The combination of relatively low sapwood capacitance, limited stomatal control on transpiration and limited embolism resistance can amount to high drought-induced mortality rates in some of these high wood density tree taxa (Janssen et al., 2019). This highlights the fact that a lack of properties contributing to drought avoidance in a particular individual or species are not always compensated for by a high drought tolerance, making this individual or species highly vulnerable to drought-induced mortality.

**5 Conclusions**

In this study, we performed a meta-analysis that provides a quantitative overview of leaf, tree and ecosystem responses to seasonal and episodic drought in neotropical humid forest. We find that the observed leaf-scale responses to episodic drought are a continuation of the responses observed during seasonal drought: reductions in leaf water potential, stomatal conductance and photosynthesis. The observed dry season decline in stem growth and increases of leaf flushing and litter fall seem to be

unrelated to water stress, as is assumed in most land surface models (LSMs). Rather, the seasonal oscillation of growth allocation between stem and canopy seems to be driven by tree phenology which is synchronised to maximum incoming solar radiation in the dry season. The analysis confirms that the variability and magnitude of drought responses decline when going from the individual leaf to the ecosystem level in highly diverse tropical forests. Biodiversity driven dynamics at the community level, such as niche partitioning, likely contribute to ecosystem resistance and resilience in response to episodic

drought. Finally, we found that wood density, via its direct relationship with sapwood capacitance, acts as a good proxy of



hydraulic behaviour and largely explains the magnitude of stomatal and transpiration responses to seasonal and episodic drought.  The results presented in this study can act as a response-benchmark for LSM simulations

**Data availability**

The data compiled for this study and used in the meta-analysis is available at https://hdl.handle.net/10411/41KALW

**Author contribution**

T.J., S.L. and H.D. designed the research, T.J., K.F., S.L., K.N. and H.D. coordinated the writing and contributed ideas, T.J. compiled the database and analysed the data, K.F., S.L., K.N. and H.D. assisted with writing the final manuscript

**Competing interests**

The authors declare that they have no conflict of interest

**Acknowledgements**

The analyses presented in this study would not have been possible without the effort of all the individual researches that enabled this study by providing freely available datasets in addition to their published work. Especially, we would like to thank Dr Tomas Ferreira Domingues, Dr Celso von Randow and Dr Alessandro Carioca de Araújo for sharing their data with us. H.D. and T.J. were funded by the Netherlands Earth System Science Centre (NESSC), financially supported by the Ministry 595 of Education, Culture and Science (OCW) (grant 024.002.001). K.F. is funded by the DFG grant No. RA 2060/5-1. S.L. was funded through an Amsterdam Academic Alliance (AAA) fellowship.

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



**Figure 1: Summary of the database. Site locations (a), average number of episodic drought months recorded per site per year (b) and number of monthly observations in the database per year (c). Below, a monthly time-series of relative extractable water and vapour pressure deficit for the K34 site in the central Amazon (d) and the multivariate ENSO index (e). The map shows the locations of the 229 neotropical forest sites from which data was used in this meta-analysis. In green, the distribution of tropical and subtropical moist broadleaf forest from the Terrestrial ecoregions of the world map (Olson et al. 2001).**








**Figure 2: Plant hydraulic status, plant hydraulic conductance, stomatal conductance and environmental drivers in the wet season, dry season and during episodic drought. Soil water potential, pre-dawn leaf water potential, midday leaf water potential (a), soil to leaf hydraulic conductance and stomatal conductance (b) are derived from published data. Relative extractable water (c) and vapor pressure deficit (d) are derived from monthly ECMWF ERA5 reanalysis data extracted to 229 neotropical forest sites in South and Central America (1979-2019). Capital letters indicate a significant (p < 0.05, Tukey HSD) difference between the wet season, dry season and episodic drought values.**



**a) Leaf**

**b) Tree**

**c) Ecosystem**

Mean seasonal drought effect (%)


**Figure 3: Meta-analysis results of leaf, tree and ecosystem scale responses to seasonal drought. The values are averages and 95% confidence intervals of percentage change in leaf, tree and ecosystem scale performance from the wet to the dry season. Repeated measurements were used, therefore the variance of the response ratio is adjusted for by the correlation coefficient between the repeated measurements (Lajeunesse, 2011). The number of consulted studies or sites is provided in brackets. The significance symbols depict the p-value derived from a Random-effects model (\*\*\* p < 0.001, \*\* p < 0.01, \* p < 0.05) testing whether the effect size differs significantly from 0.**








**Figure 4. Meta-analysis results of leaf, tree and ecosystem scale responses to episodic drought. The values are averages and 95% confidence intervals of percentage change in leaf, tree and ecosystem scale performance during episodic drought, relative to an average dry season. Repeated measurements were used, therefore the variance of the response ratio is adjusted for by the correlation coefficient between the repeated measurements (Lajeunesse, 2011). The number of consulted studies or sites is provided in brackets. The significance symbols depict the p-value derived from a Random-effects model (\*\*\* p < 0.001, \*\* p < 0.01, \* p < 0.05) testing whether the effect size differs significantly**





**Figure 5: Effect size of leaf-scale responses to seasonal drought (black) and episodic drought (red) for different studies against the study averaged wood density. The point size is the inverse of the sample standard error of the effect size in the study. The test statistics are retrieved from a Mixed-effect model testing the significance of wood density as a moderator in the drought response. The solid line is the model prediction and the dashed lines are the 95% confidence intervals. Regression lines were only drawn if the relationship was significant (p < 0.05).**





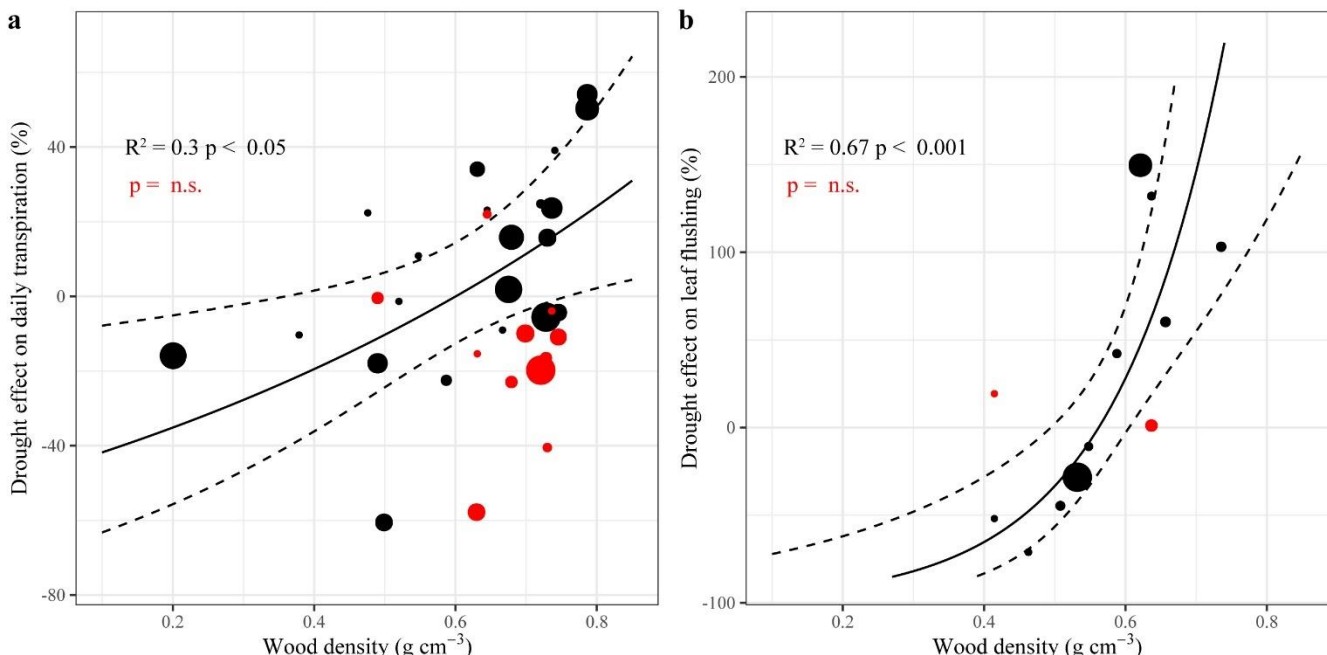

**Figure 6 Effect size of daily transpiration (a) and leaf flushing (b) to seasonal drought (black) and episodic drought (red) for different studies against the study averaged wood density. The point size is the inverse of the sample standard error of the effect size in the study. The test statistics are retrieved from a Mixed-effect model testing the significance of wood density as a moderator in the drought response. The solid line is the model prediction and the dashed lines are the 95% confidence intervals. Regression lines were only drawn if the relationship was significant (p < 0.05).**
