# Peer review of "Drought resistance increases from the individual to the ecosystem level in highly diverse neotropical rain forest: a meta-analysis of leaf, tree and ecosystem responses to drought"

_Biogeosciences, 2019_

## Referee Comment (RC1) · Anonymous Referee #1 · 16 Jan 2020

Janssen et al. examine whether a meta-analysis of leaf-, tree- and ecosystem-level data can help understand, and predict, neotropical rainforest responses to drought. They ask two questions: (i) how does drought impact the vegetation from the leaf to the ecosystem scale?, and (ii) can different hydraulic behaviours at different locations or among species explain differences in the responses to drought? They find that episodic drought effects compound on dry season effects at both the leaf and tree scales. However, vegetation responses are buffered at the ecosystem scale and, notably, are often not significant during episodic drought. Finally, independently

compiled wood density data are used to explain some of the variability observed at the leaf and tree scales during the dry season (and to a lesser degree during episodic droughts).

I commend the authors for this undertaking (138 studies!) and for the quality of their writing. The study will make an important contribution by explaining the eco-physiological impacts of drought on a key region's rain forests, at a range of scales. However, I have several major methodological concerns that should be addressed in revision.

———————————————————- Main comments ———————————————————-

My fist observation is that, according to the number of measurements / estimates compiled by the authors, episodic droughts data (624) represent 9% of the total amount of data (6956) and to 17% of the dry season data (3006). This feels like a very high number of episodic drought observations compared to the rest of the observations. Looking at Figure 1b and c, the number of observations per months appears biased towards the more recent years. Does this bias explain the frequency increase in the average of episodic drought months per year in the more recent years? Looking at Fig. 1d, I am also questioning the definitions used for the wet season, dry season, and episodic droughts. For example, in 2000, the K34 site starts of by being in the wet season for 5 months, then in the dry season for 1 month, then in the wet season again for 1 month, then in the dry season for 2 months, wet season for 2 months, dry season for 1 month, wet season for 2 months, dry season for 1 month.... This pattern of oscillating wet and dry season is seen repeated within the following years, but how likely is it to represent the "real" wet and dry season? And so, how can dry season effects on the vegetation be captured on time scales that make sense? Again, if we look at the dry season and episodic drought between 2015 and 2016, we

see a transition from episodic drought to wet season although the relative extractable soil water is very close to 0. I understand from the authors' definition of the wet season that this is because the soil moisture has started to be replenished. Realistically, if the vegetation had just gone through an episodic drought, then would the next month's measurements of stomatal conductance, photosynthetic rate, etc. be representative of a wet season month?

Therefore, owing to potential hydraulic function damage sustained during the drought, the authors might want to rethink their definitions of the wet and dry seasons, as well as of the episodic droughts, in terms of what makes sense when considering potential multi-weeks (but not multi-years) legacy impacts on the vegetation. One solution would be to classify some of the data as being within "recovery months" (i.e. from a drought or from the dry season to the wet season) and to analyse them separately. The authors should also consider testing the sensitivity of their results to different quantile threshold definitions for what consists in the wet and dry season, as well as in an episodic drought.

My second concern relates to the method used to calculate the percentage changes shown in Figs. 3 and 4. It is very clear from Figs. S2, S3, and S4 that wood density is a good proxy for leaf- and tree-level hydraulic behaviour. So why not cluster the analysis of the rates of change by types of wood density (e.g. low vs high), to ensure that opposite types of leaf- and tree- level behaviours are not compensating and cancelling each other out when looking at the rates of change?

I understand that this is what Figs. 5 and 6 attempt to do, but I do think the broader narrative would be more successful had the meta-analysis differentiated between isohydric and anisohydric behaviours from the start. Clustering by behaviour might also help reconciliate and explain the current inconsistencies in the findings from the leaf-level up to the ecosystem scale.

My third point has to do with the VPD values used to estimate changes in leaflevel transpiration. The leaf-level transpiration is estimated using the relationship $E = gs \times D$ where $D$ is VPD.

Here, the authors use monthly averaged atmospheric midday VPD derived from the ERA5 reanalysis data. I am surprised because the VPD values present in the database are very low, with a maximum of 2.35 kPa across all 6956 data points and the 95th percentile < 1 kPa. Given that > 50% of the total data is classified as corresponding to either the dry season or to an episodic drought, I would at least expect the 95th percentile value of the average monthly midday VPD to be > 1 kPa! It is unclear to me whether these low values are due to using the Buck method to calculate VPD, or to the ERA5 data themselves.

Additionally, using atmospheric VPD rather than leaf-to-air VPD (which the relation $E = gs \times D$ is designed for) ignores feedback effects from the leaf to the atmosphere above. When plants transpire during a drought (or a heatwave), they also cool the air immediately above them, leading to lower leaf-to-air VPD than atmospheric VPD.

One finding of this paper is that "the data shows no significant decline in leaf transpiration from the wet to the dry season [...] as the average increase of VPD from the wet to the dry season is of the same magnitude as the decline of stomatal conductance". Instead, higher estimates of midday VPD (e.g. from a different reanalysis product) could lead Janssen et al. to predict an increase in transpiration during the dry season. Or, conversely, using leaf-to-air VPD might lead to a smaller magnitude increase in leaf-to-air VPD than the decline in stomatal conductance, thus leading to predicting a reduction in leaf-level transpiration in the dry season!

It is very hard to tell what the implications of the VPD estimates are, but they currently make it hard to trust the leaf-level estimates of transpiration, Potential ways forward are:

1. to use a different method than the Buck method and to quantify the uncertainty;

2. to compare the current VPD estimates with different reanalysis products (e.g. ERA-Interim which has been evaluated more) or other products, such as the

CRU data, and to quantify the uncertainty;

3. to calculate a proxy of leaf-to-air VPD using atmospheric VPD and leaf water potential to account for a degree of leaf-atmosphere feedbacks.

———————————————- Minor comments ——————————————-

L. 22: it's hard to see how the results could be used as a benchmark for LSMs, given e.g. the unexplained differences in transpiration responses from the leaf- and tree- scales to the ecosystem scale. Instead, do the authors mean that the relationships they find between the different variables and wood density could help guide LSM parameterisation efforts in neotropical forests?

L.32: maybe consider citing Yang et al. 2018 (https://doi.org/10.1038/s41467-018-05668-6), which uses LiDAR and allometric relationships, in place of Zhao and Running? The Zhao and Running paper has temperature dependencies which are problematic and have been discussed in several technical comments....

L.54: I suggest starting a new paragraph at "Episodic droughts"

L.55-56: do tropical North Atlantic SST anomalies affect all the neotropics? Or do they primarily affect the easternmost region?

L. 75: "stomates progressively close" is more exact than "stomates close"

L.76: also:

1. Martin St-Paul et al. 2017 (http://doi.wiley.com/10.1111/ele.12851),

2. Drake et al. 2017 (https://doi.org/10.1016/j.agrformet.2017.08.026),

3. Choat et al. 2018 (https://doi.org/10.1038/s41586-018-0240-x)

L. 84-85: $E$ can either stay the same, increase, or decrease during a drought, all of which could result on a decline in $\Psi_l$.... Also, $k_{sl}$ declines as a result of a decline in $\Psi_s$

L. 87-88: stomatal closure (described above) and stomatal downregulation are not the same, so the link isn't clear from the current phrasing. Also, using the words "potential" and "potentially" could lead to misinterpretation

L. 88-90: Is this meant as a global statement? Or is it still in the context of neotropical forests? Generally, this is quite variable depending on species, ecosystem, and timing... with different responses being observed at different stages of a drought

L. 104-106: I think the paragraph would be clearer if this sentence came right after the reference to Sayer et al 2007, L. 102

L. 107: here, maybe repeat what the three spatial scales are

L. 114: change "drought avoiding and drought tolerating strategies" to "drought avoidance or tolerance strategies"?

L.114-115: xylem embolism doesn't always substantially damage the hydraulic pathway, maybe consider rephrasing as "Drought avoidance strategies aim to avoid dangerous declines in $\Psi_l$ that could lead to significant xylem embolism and thus damage..."?

L. 118-120: consider rewriting as: "Conversely, drought tolerance strategies imply [. . .] without significant and/or irreversible embolism-induced losses of hydraulic function"?

L. 120-123: the isohydric vs anisohydric (why use "non-isohydric" rather than anisohydric?) need a bit more explanation, e.g. isohydric species maintain a constant midday $\Psi_l$ but also down-regulate their stomatal conductance. It would also be worth mentioning that the spectrum of isohydric and anisohydric behaviours is quite large, with some species having the capacity to oscillate between more-or-less isohydric or anisohydric behaviours depending on the environmental conditions...

L.131-133: this is a very nice hypothesis! To introduce it, the authors could refer to the work of Rosas et al. 2019 (https://doi.org/10.1111/nph.15684)

L. 136-138: I think that moving this sentence to line 133 before "In neotropical..." would make the text flow better

L. 138-140: this is very useful contextualisation, maybe it could make it into the abstract?

L. 153: typo: "it" to "they"

L. 156: what about measurement techniques and errors? Were those also included in the database? I imagine there would be different margins of error depending on the measurement technique. Also, were there quality checks or did all the above described data make it into the database?

L. 157: was the time of day not reported? This would highly impact measurements of stomatal conductance and leaf photosynthesis...

L. 157-158: how many different species, genus, and/or different site averages?

L. 160-161: the information on how the spatial data were extracted is probably not needed

L. 171: shouldn't "midday vapor pressure deficit" be "monthly averaged midday vapor pressure deficit"?

L. 173-175: the authors need to mention that this assumption largely ignores variations in root distributions

L. 177-178: please clarify what "ecosystem performance measures" means

L. 179: were all the stomatal conductance measurements made at midday? Also, it is worth mentioning that this relationship assumes a perfect coupling between the stomates and the atmosphere above, i.e. it assumes that the boundary layer conductance to water vapour, $g_b$, is much larger than $g_s$. But in forests with large leaves and dense canopies, decoupling is often observed because $g_b$ is relatively small, such that when $g_s$ and $g_b$ are of similar magnitudes $E \approx 0.5 \times gs \times D$. In the context of this study, it is impossible to estimate what the coupling/decoupling factor is at a given location and/or at a given time, but the authors should mention this (in the context of leaf shedding and flushing?), given their findings

L. 184-186: from the text alone, it is very unclear how transpiration was estimated. How does the RMSE represent the linear relationship?
Looking at Fig S1, I presume the authors have compiled tree scale measurements of E so, in the analysis, why not just use those measurements (instead of the estimates described by the linear relationship)?

L. 190: my interpretation of equation 3 is that it should only be valid at steady-state. How did the authors ensure steady-state conditions? Were the data filtered

depending on VPD?

L. 192: typo: "rooting zone" to "root-zone"

L. 193-194: strictly speaking, difference between $\Psi_l$ at midday and $\Psi_{pre-dawn}$ is a proxy of the water gradient within the tree, from the root up to the canopy. For it to equate soil-canopy gradient, further information on tree height would be needed to account for gravitational effects and relate $\Psi_{pre-dawn}$ to $\Psi_s$...

L. 225-227: I realise it's common to use log response ratios when comparing large amounts of data, but why not directly use the percentage change to quantify drought effect size?

L. 242-243: given the large variability in hydraulic behaviour observed within a genus, is it reasonable to use the genus average as a proxy here? And how many of the location points are affected by this assumption?

L. 254: the reference to Figure 2a is needed here too

L. 260: I find hard to believe that this is an actual result and not simply a product of the methods used to calculate the leaf-level transpiration....

L. 265: but a drop in $\Psi_l$ is observed!

L. 275-276: the authors could mention that this is in line with the findings of Rosas et al. 2019 along a mesic-xeric gradient (although their study is not on neotropical species)

L. 304 (and later): the "WUEi" notation is inconsistent with the "iWUE" notation

used in the introduction

L. 311: "we observe that" is not needed

L. 315: typo: "marginal" should be "marginally"

L. 322: give the ranges of variation?

L. 320-328: the findings would benefit from being broken down in terms of the dry season (significant) vs episodic drought (mainly not significant)

L. 334: the authors need to state that the relationship is not significant...

L. 336: "intermediate response" is very vague, please reformulate

L. 337-342: these findings are very useful!

L. 344-346: How is it "similar"? Fig. 6a seems to show far less significance and way more scatter than Fig. 5b

L. 356: why "hydrological"? Do the authors mean hydraulic?

L. 358: please replace "cancelled out" by "offset"

L. 392-394: I don't follow this sentence.... these effects can be consistently observed for weeks, and even months? Do the authors mean that leaf effects are typically observed on shorter time scales due to the "life expectancy" of a leaf compared to a tree, or to an ecosystem?

[Figure]

L.405-406: the mention of these "ENSO swings" would be a better fit L. 400, right after the list of references. But what is an ENSO swing? This is never defined...

L. 398-407: I'm not entirely clear why the increase in the frequency of episodic droughts is not first mentioned in the results section?

L. 538-540: this should come earlier, after L. 120-123

L. 546: but can also be explained by plant capacitance

L. 564: typo: missing "and" after "environments"

Fig. 1a: the K34 site should be indicated on the map given Fig. 1d and 1e
Fig. 1b: this is averaged across sites, right? I wonder whether it would make more sense to actually average the episodic drought months across the whole area of neotropical forests shown on the map. This would potentially reduce sampling biases in concluding that episodic droughts have been increasing in neotropical forests. Alternatively, the authors could consider weighting this by the number of monthly observations per year.
Fig. 1e: visually, it would be very nice if the ENSO index was coloured to match the wet and dry season and the episodic droughts

Fig. 2a: where do the top soil $\Psi_s$ data come from? The caption says published data, but I didn't find it in the methods?
Fig. 2a and 2b: yes to the mention of capital letters in the legend, but what does it mean when letters are coupled (e.g. AB in the dry season in Fig. 2a) or when the letter A or B appear during episodic droughts?
Fig.2: I imagine the horizontal lines in the box plots show the median, the boxes themselved interquartile ranges, the vertical lines the 5th-95th percentiles and the

points are outliers? This needs to be mentioned in the legend

Figs. 3  4: what do the horizontal lines represent? Ranges?
Additonally, it would be useful:

1. to also mention the number of data points (or average number of data points per study/site) in brackets;

2. to visually separate the variables that were directly retrieved from the literature from those that necessitated further calculations.

Figs.  5  6: so bigger points mean smaller errors?  Does that also play a role in the weighting of the solid and dashed lines?

---

## Referee Comment (RC2) · Anonymous Referee #2 · 28 Jan 2020

In this study, Janssen et al perform a meta-analysis to look to drought impacts on carbon and water exchange across scales ranging from leaf, to plant, to ecosystem in neotropical rainforests. In particular, the authors contrast physiological responses from seasonal water stress versus 'episodic' drought events. In doing this, the authors also look at wood density as a proxy for plant physiological responses. Its clear that the authors put in a significant amount of work through compiling 138 studies across 229 sites and this study will clearly contribute to the physiological literature on forest drought responses. I do have some major revision suggestions before publication which are

summarized here and, in some cases, elaborated in the line specific comments below.

1) I think that the manuscript would benefit from reframing of the dry season 'drought' as a routine period of decreased water availability. When I think of droughts, I think of a prolonged period of abnormally low rainfall. Given that dry seasons occur every year, I don't see them meeting this definition. This reframing would provide a nice platform to discuss physiological responses to routine (seasonal) stress, such as phenology, versus episodic stress and can help get at important physiological mechanisms. This would involve some substantial reworking of the text, but I think it would really help the story line. 2) Some of the methods are confusing and I think elaborating a bit more, providing and providing a table might help. See line specific comments below 3) The Figure legends need to be more descriptive of all the features in the figures 4) Overall the manuscript does an impressive job discussing a range of processes, but the reader might be more attentive if it were a little shorter. Where possible, I would suggest the authors minimize extraneous discussion. In particular, I think talking about isohydricity requires significant motivation for a general audience (which is not provided), so I would cut this text.

Line-specific comments: L21-22 There is nothing to back up this statement on LSMs. I suggest the authors remove it

L35 Khanna et al 2017 Regional dry-season climate changes due to three decades of Amazonian deforestation

L50 after going through the MS, I am confused about where the multi-year drought is presented in the authors analysis

L53-54 This is exactly why I would argue that 'seasonal drought' is a misnomer

121 The concept of isohydricity/anisohydricity will likely not be familiar to a broad audience at Biogeosciences. I would encourage the authors to eliminate the jargon and focus on the mechanisms of interest (see Martinez-Vilalta 2016 "Water potential regulation, stomatal behaviour and hydraulic transport under drought: deconstructing the iso/anisohydric concept") or else devote more space to describing isohydric behavior

L131-132 But the authors actually try and demonstrate that wood density is usable as a proxy first, so isn't this more of a hypothesis?

L145 where does the multiyear drought aspect come in that the authors mentioned in the intro?

Intro overall: The introduction is on the longer side (as is the manuscript) and the three separate sections do not have a smooth transition. I would try to combine 1.2 and 1.3 and streamline the text. Given that sections 1.2-1.3 are more about the mechanisms, I would put thes sections first and then say that different types of physiological stress also impact C and W fluxes including seasonal, routine decreases in water availability versus droughts

L152 ERA5 citation?

L153-155 It would be good for the authors to say how many studies were associated with each of these diagnostics and how many studies had multiple observations, maybe a table would be useful?

L157-159 I am confused about how observations were calculated/recorded across scales? Could the authors elaborate? For example, were ecosystem-level measurements independent from leaf level, or were different levels calculated using observations from different scales by the authors? Both more details and a table detailing number of measurements for each diagnostic and number of studies that cover multiple diagnostics and scales would help

L161-162, this doesn't need to be included

L166-167 This is a huge amount of work, I commend the authors

L170 it would be good to say the spatial resolution and include a citation for on ERA5

(30km?)

L171 monthly average midday VPD?

L187 how was this error estimated?

L219 Figures should be renumbered so Fig 2 comes after Fig 1

L208-209 It would be good to include a possible caveat about uncertainty associates with ERA5 soil moisture (which I presume is modeled)

L212-213 why the 65% and 10% quantiles? Did the authors test the sensitivity of their results to this assumption?

L220 Could a study be both a dry season and episodic drought? I am a little confused about the partitioning. Also, where does multi-year drought come in?

L225 did the authors check to see if it was necessary to log the response?

Where did the ENSO data come from?

L254 how can predawn wp be positive 0.22?! Please check for a typo

L281 Could the authors include Reco in some of the figures, they refer a change in Reco several times in the text but no visuals are provided

L363 typo include 'us'

L370 denoted iWUE previously

L401 it is also true that there are more observations post 2000. The authors should discuss how this might impact their results

L419 it would be nice to put these numbers in a physiological context

L451-458 I think there is a really nice opportunity to contrast tree physiological strategies when exposed to routine stress (the dry season) versus drought that the current narrative doesn't allow for when both are classified as drought

L461 is it the short timescale, or that fact that the plants are used to this type of stress and use phenology to deal with it?

L464 Or maybe title: how do we scale from the leaf to the ecosystem? This is a universal problem in ecology that researchers across many subdisciplines will sympathize with

L502 see previous suggestion about isohydricity

L536 This isn't a great comparison. The authors of this manuscript analyze site-specific data whereas the spatial scale of the Konings study is o(100km). For me, this paragraph does not contribute much to the study and in general I think the isohydricity framework is not useful here (and otherwise)

L575 LSMs are brought up in only in the abstract and conclusion. It doesn't add to the discussion and I would remove this

L582 How should they be used to benchmark LSMs? If the authors insist on including this, please the need to spell out the methodology rather than throwing it in as a concluding sentence

Other relevant citation: Detto 2018 "Resource acquisition and reproductive strategies of tropical forest in response to the El Niño-Southern Oscillation"

Figures: Fig. 1 panel a make lat/lon bigger b) its really hard for me to wrap my head around what the authors mean by this metric, can they elaborate? d) What do the dots mean? Please describe this in the legend and also detail what dark and light gray correspond to legend (e) it would be good to remind the reader what the positive/negative ENSO index means L1131 "terrestrial" isn't capitalized

Fig. 2 can the authors denote the sample size above each category in the figure? For example, does n=3 for Episodic drought soil-leaf hydraulic conductance in panel b? Make sure to describe the figure fully (detail quantile boxes, median line, error bars, and outliers) in the legend

Fig. 3-4 I would combine these two figures into 1 2-column 3-row figure. I generally really like this format and found it very effective in the Ainsworth review. Great job describing all aspects of the figure in the legend

Fig 5-6 I would combine these figures into a 2-c 3-r figure The point size is the inverse of the sample standard error of the effect size in the study.–> so studies with a smaller SE have a larger dot? Moderator= independent variable?

Fig 5 legend, make sure to walk the reader though each panel

---

## Author Comment (AC1) · 18 Feb 2020

Author response to Referee #1

Janssen et al. examine whether a meta-analysis of leaf-, tree- and ecosystem-level data can help understand, and predict, neotropical rainforest responses to drought. They ask two questions: (i) how does drought impact the vegetation from the leaf to the ecosystem scale?, and (ii) can different hydraulic behaviours at different locations or among species explain differences in the responses to drought? They find that

episodic drought effects compound on dry season effects at both the leaf and tree scales. However, vegetation responses are buffered at the ecosystem scale and, notably, are often not significant during episodic drought. Finally, independently compiled wood density data are used to explain some of the variability observed at the leaf and tree scales during the dry season (and to a lesser degree during episodic droughts). I commend the authors for this undertaking (138 studies!) and for the quality of their writing. The study will make an important contribution by explaining the ecophysiological impacts of drought on a key region's rain forests, at a range of scales. However, I have several major methodological concerns that should be addressed in revision.

Response: thank you very much for your extensive report. We appreciate the thorough review of our manuscript and your acknowledgement of the relevance of our work. Below we reply to the comments raised in the referee report.

Main comments My fist observation is that, according to the number of measurements / estimates compiled by the authors, episodic droughts data (624) represent 9% of the total amount of data (6956) and to 17% of the dry season data (3006). This feels like a very high number of episodic drought observations compared to the rest of the observations. Looking at Figure 1b and c, the number of observations per months appears biased towards the more recent years. Does this bias explain the frequency increase in the average of episodic drought months per year in the more recent years?

Response: yes, there is a sampling bias in our meta-analysis towards the more recent years and this does result in more months being classified as episodic drought months than would be expected based on the 10% cut-off point. The sampling bias however does not explain the trend of increased episodic droughts (Figure 1 b) because the entire time series of ERA5 soil moisture was used to count the episodic droughts, whether we have field data in those months or not (see also specific comment on Figure 1). This will be made more clear in the revised version of the manuscript.

For the purpose of this meta-analysis we defined "episodic drought" as a month where
the relative extractable soil water (REW) was lower than the 10% quantile of REW in the entire ERA5 soil moisture time-series (1979-2019). If the data retrieved from the literature was evenly distributed over the entire timespan of the ERA5 data we would expect that around 10%, so 363 of the 3630 initially classified dry season months would be classified as episodic drought. As pointed out by you, we classified almost double that amount, 624 months, as episodic drought. This is indeed explained by the bias of the retrieved data towards recent years (2009-2019) that were also drier compared to the previous decades (Figure 1b and 1c). Elaborating on this, we find that before 2009 ∼14% (346 out of 2503) of initially classified dry season months were classified as episodic drought while in the decade 2009-2019 ∼25% (280 out of 1125) of the dry season months were classified as episodic drought. *Note that the total amount of data slightly changed since the previous version because of the use of a new digital elevation model*

Looking at Fig. 1d, I am also questioning the definitions used for the wet season, dry season, and episodic droughts. For example, in 2000, the K34 site starts of by being in the wet season for 5 months, then in the dry season for 1 month, then in the wet season again for 1 month, then in the dry season for 2 months, wet season for 2 months, dry season for 1 month, wet season for 2 months, dry season for 1 month.... This pattern of oscillating wet and dry season is seen repeated within the following years, but how likely is it to represent the "real" wet and dry season? And so, how can dry season effects on the vegetation be captured on time scales that make sense?

Response: the separation of months into wet and dry season was indeed based on the depletion or replenishment of ERA5 soil moisture for a specific site. We will more elaborately discuss our choice of wet season and dry season definition when introducing the classification procedure in the Methods of the revised version of the manuscript. The rationale behind our definition is that when evapotranspiration exceeds precipitation, there is a precipitation deficit resulting in the depletion of soil moisture. This definition is often used to distinguish wet and dry season in neotropical forests and also used

to calculate drought metrics such as the cumulated water deficit (Aragão et al., 2007). Although our definition is simple and might not always capture the complete ecophysiological responses to the dry season at every site in every year, we think that it does capture the general wet season and dry season states across the different sites. For example, the vapor pressure deficit is clearly higher during the dry season compared to a wet season (Figure 1d and 2d) indicating both a dry soil and a dry atmosphere during the dry season. For some years with an exceptional weak dry season (as in the year 2000 at the K34 site) the classification results in a rapid sequence of wet and dry season months. We partly tried to correct for these small changes in soil moisture by classifying all months with REW higher than 65% of REW as wet season months despite a reduction in REW (Methods L211-L213) but this did not entirely prevent that in some exceptional years there is a rapid sequence of wet and dry seasons in a single year. Nonetheless, the wet-dry season classification method resulted in that months classified as dry season are actually dry and are not just classified as dry season months because it is a particular month of the year.

Again, if we look at the dry season and episodic drought between 2015 and 2016, we see a transition from episodic drought to wet season although the relative extractable soil water is very close to 0. I understand from the authors' definition of the wet season that this is because the soil moisture has started to be replenished. Realistically, if the vegetation had just gone through an episodic drought, then would the next month's measurements of stomatal conductance, photosynthetic rate, etc. be representative of a wet season month? Therefore, owing to potential hydraulic function damage sustained during the drought, the authors might want to rethink their definitions of the wet and dry seasons, as well as of the episodic droughts, in terms of what makes sense when considering potential multi-weeks (but not multi-years) legacy impacts on the vegetation. One solution would be to classify some of the data as being within "recovery months" (i.e. from a drought or from the dry season to the wet season) and to analyse them separately.

Response: on the leaf scale, the available literature reported no clear legacy effects of episodic drought on stomatal conductance, leaf water potential and photosynthesis (Alexandre, 1991; Santos et al., 2018). However, on the tree scale there are legacy effects reported, for example reduced hydraulic conductance and transpiration directly following episodic drought (Fontes et al., 2018) and changes in stem growth and leaf flushing (Doughty et al., 2014, 2015; Hofhansl et al., 2014). We discuss observed legacy effects reported in the literature in the Discussion (L486-L496) and acknowledge that the method used by us is only able to capture instantaneous responses and not the legacy effects, which is a limitation. When revising this manuscript we will more elaborately discuss the presence of legacy effects in our meta-analysis.

The authors should also consider testing the sensitivity of their results to different quantile threshold definitions for what consists in the wet and dry season, as well as in an episodic drought.

Response: the lack of a "sensitivity analysis" was also noted by referee #2 L212-213. In the revised version of the manuscript we will more elaborately discuss our choice for a 10% threshold. We opted for a threshold that provided a reasonably strict episodic drought definition while still yielding a large enough sample size for the statistical analysis to differentiate between episodic drought and a regular dry season. We have now also tested a wide episodic drought threshold of 15% and a narrow episodic drought threshold of 5%. The wider episodic drought definition resulted in a decline of the sample size for the wet season – dry season comparison as more dry season months were classified as episodic drought. Furthermore, the p-values of 14 out of 23 variables declined but none of the previous significant variables became not significant in the wide (15%) threshold wet season – dry season comparison. wide episodic drought definition resulted in the increase of the sample size in the dry season - episodic drought comparison but also a decline in the p-values of 15 out of 23 variables, while none of the previously significant variables became not significant in the dry season – episodic drought comparison. The narrow episodic drought definition (5%) resulted in larger

sample size for the wet season – dry season comparison compared to the baseline (10%) definition because more episodic drought months are now classified as dry season, and the increase of p-values in about half (12 out of 23) of the variables, while one variable (soil-leaf hydraulic conductance) became not significant in the wet season – dry season comparison. Furthermore, the narrow definition resulted in a decline in sample size for almost all variables (19 out of 23) and a decline of the p-values in 15 out of 23 variables in the dry season – episodic drought comparison with 4 previously significant variables now showing no significant change (soil-leaf hydraulic conductance, leaf transpiration, leaf photosynthesis and ecosystem water use efficiency). These results confirm that our analysis of seasonal drought is quite robust, with no major changes in the magnitude and direction of change of any variables in response to seasonal drought with different threshold values for episodic drought. Also the responses to episodic drought show no major changes in direction or magnitude but we observe a decline in significance levels in some variables, mainly because of a reduction in sample size. We will add these results to the supplementary material and discuss the implication of the threshold in the discussion.

My second concern relates to the method used to calculate the percentage changes shown in Figs. 3 and 4. It is very clear from Figs. S2, S3, and S4 that wood density is a good proxy for leaf- and tree-level hydraulic behaviour. So why not cluster the analysis of the rates of change by types of wood density (e.g. low vs high), to ensure that opposite types of leaf- and tree- level behaviours are not compensating and cancelling each other out when looking at the rates of change? I understand that this is what Figs. 5 and 6 attempt to do, but I do think the broader narrative would be more successful had the meta-analysis differentiated between isohydric and anisohydric behaviours from the start. Clustering by behaviour might also help reconciliate and explain the current inconsistencies in the findings from the leaf-level up to the ecosystem scale.

Response: we agree that merging the drought responses of all the species and functional groups present in the database results in the loss of the variability in responses

observed. In the case of transpiration this merging indeed results in that we observe on average no significant changes in transpiration from the wet to the dry season (Figure 4) while studies that measured mainly high wood density species or low wood density species did show a significant increase or decrease in transpiration, respectively (Figure 5 and 6). As you mention, showing this variability is the purpose of Figure 5 and 6 while for Figure 3 and 4 the aim is to show the average response. We did consider splitting the data shown in Figure 3 and 4 in studies measuring mainly isohydric and non-isohydric species, however, this distinction is not easily made. As can be seen in Figure 5 and 6, there are not really two clusters of hydraulic behaviour but it is rather a continuum, related to the continuum from strictly isohydric to extreme non-isohydric behaviour that is observed in plants globally (Klein, 2014; Martínez-Vilalta et al., 2014; Meinzer et al., 2017). We think that arbitrary splitting the data in isohydric and non-isohydric studies will not help to reconcile the inconsistencies in drought responses observed from the leaf to the ecosystem scale. However, the relationships found and shown in Figure 5, 6, S2 S3 and S4, could inform vegetation models that could then simulate the effect of a variability in plant hydraulic behaviour on ecosystem scale productivity and transpiration.

My third point has to do with the VPD values used to estimate changes in leaf level transpiration. The leaf-level transpiration is estimated using the relationship E = gs × D where D is VPD. Here, the authors use monthly averaged atmospheric midday VPD derived from the ERA5 reanalysis data. I am surprised because the VPD values present in the database are very low, with a maximum of 2.35 kPa across all 6956 data points and the 95th percentile < 1 kPa. Given that > 50% of the total data is classified as corresponding to either the dry season or to an episodic drought, I would at least expect the 95th percentile value of the average monthly midday VPD to be > 1 kPa! It is unclear to me whether these low values are due to using the Buck method to calculate VPD, or to the ERA5 data themselves. Additionally, using atmospheric VPD rather than leaf-to-air VPD (which the relation E = gs × D is designed for) ignores feedback effects from the leaf to the atmosphere above. When plants transpire during a drought (or

a heatwave), they also cool the air immediately above them, leading to lower leaf-to-air VPD than atmospheric VPD. One finding of this paper is that "the data shows no significant decline in leaf transpiration from the wet to the dry season [. . .] as the average increase of VPD from the wet to the dry season is of the same magnitude as the decline of stomatal conductance". Instead, higher estimates of midday VPD (e.g. from a different reanalysis product) could lead Janssen et al. to predict an increase in transpiration during the dry season. Or, conversely, using leaf-to-air VPD might lead to a smaller magnitude increase in leaf-to-air VPD than the decline in stomatal conductance, thus leading to predicting a reduction in leaf-level transpiration in the dry season! It is very hard to tell what the implications of the VPD estimates are, but they currently make it hard to trust the leaf-level estimates of transpiration, Potential ways forward are: 1. to use a different method than the Buck method and to quantify the uncertainty; 2. to compare the current VPD estimates with different reanalysis products (e.g. ERA-Interim which has been evaluated more) or other products, such as the CRU data, and to quantify the uncertainty; 3. to calculate a proxy of leaf-to-air VPD using atmospheric VPD and leaf water potential to account for a degree of leaf-atmosphere feedbacks.

Response: thank you for pointing out the very low VPD that we used to calculate leaf transpiration in our meta-analysis! After reviewing our pre-processing steps that we used to obtain the ERA5 VPD at every site, we found the mistake that resulted in these low VPD estimates: instead of using local time (12:00) temperature and dewpoint temperature to calculate midday VPD, we erroneously used 12:00 UTC when downloading the ERA5 temperature data.

For the analyses in the revised version of the manuscript we will now use temperature and dewpoint temperature at four different times (15:00 – 18:00 UTC) that correspond with local 12:00 in four time zones covering our study area. The ERA5 results seem to correspond reasonably well with VPD observations from flux towers in our study area (Figure 1). Using the actual midday VPD logically resulted in changes in the range of
VPD found in the meta-analysis (Figure 2). However, we observe no major changes in the direction or magnitude of the leaf transpiration response to seasonal (Figure 3) and episodic drought. The response of leaf transpiration to seasonal drought remains not significantly different from 0 and significantly declines in response to episodic drought. This can be explained because the new and correct midday VPD is higher compared to the previous VPD estimate in the wet season, in the dry season and during episodic drought which results in marginal changes in the relative response of leaf transpiration.

We recognise that atmospheric VPD and leaf-to-air VPD can be very different depending on the cooling feedback resulting from leaf transpiration. However, leaf-to-air VPD, leaf temperature or leaf water potential were not consistently provided in the original source papers that provided the stomatal conductance data, preventing us from calculating actual leaf transpiration. We agree that we have to be more careful with the results from this analysis. In the revised version of the manuscript we will discuss the implications of using atmospheric VPD instead of leaf-to-air VPD in calculating leaf transpiration.

Minor comments L. 22: it's hard to see how the results could be used as a benchmark for LSMs, given e.g. the unexplained differences in transpiration responses from the leaf and tree- scales to the ecosystem scale. Instead, do the authors mean that the relationships they find between the different variables and wood density could help guide LSM parameterisation efforts in neotropical forests?

Response: yes, we agree with your suggested change of formulation: "We present new insights into the functioning of tropical forest in response to drought and present novel relationships between wood density and drought responses that can help guide the parametrization of land surface models."

L.32: maybe consider citing Yang et al. 2018 (https://doi.org/10.1038/s41467-018-05668-6), which uses LiDAR and allometric relationships, in place of Zhao and Running? The Zhao and Running paper has temperature dependencies which are problematic and have been discussed in several technical comments....

Response: thank you for your suggested change in referenced literature, we will change this in the revision.

L.54: I suggest starting a new paragraph at "Episodic droughts"

Response: this will be changed

L.55-56: do tropical North Atlantic SST anomalies affect all the neotropics? Or do they primarily affect the easternmost region?

Response: according to Marengo et al. 2011, the North Atlantic SST affected the position of the ITCZ, forcing the ITCZ anomalously northward during 2010, resulting in an episodic drought in the southern Amazon Basin.

L. 75: "stomates progressively close" is more exact than "stomates close"

Response: agree, better formulation. This will be changed.

L.76: also: 1. Martin St-Paul et al. 2017 (http://doi.wiley.com/10.1111/ele.12851), 2. Drake et al. 2017 (https://doi.org/10.1016/j.agrformet.2017.08.026) 3. Choat et al. 2018 (https://doi.org/10.1038/s41586-018-0240-x )

Response: agree, these references are a good addition to Buckley et al. 2019

L. 84-85: E can either stay the same, increase, or decrease during a drought, all of which could result on a decline in $\Psi l$.... Also, $k_{sl}$ declines as a result of a decline in $\Psi s$

Response: this will be reformulated.

L. 87-88: stomatal closure (described above) and stomatal downregulation are not the same, so the link isn't clear from the current phrasing. Also, using the words "potential" and "potentially" could lead to misinterpretation

Response: this will be reformulated.

L. 88-90: Is this meant as a global statement? Or is it still in the context of neotropical forests? Generally, this is quite variable depending on species, ecosystem, and timing... with different responses being observed at different stages of a drought

Response: yes we agree this is confusing, this will be reformulated.

L. 104-106: I think the paragraph would be clearer if this sentence came right after the reference to Sayer et al 2007, L. 102

Response: this will be reformulated.

L. 107: here, maybe repeat what the three spatial scales are

Response: this will be reformulated.

L. 114: change "drought avoiding and drought tolerating strategies" to "drought avoidance or tolerance strategies"?

Response: this will be reformulated.

L.114-115: xylem embolism doesn't always substantially damage the hydraulic pathway, maybe consider rephrasing as "Drought avoidance strategies aim to avoid dangerous declines in $\Psi l$ that could lead to significant xylem embolism and thus damage..."?

Response: yes we agree, this will be reformulated.

L. 118-120: consider rewriting as: "Conversely, drought tolerance strategies imply [. . .] without significant and/or irreversible embolism-induced losses of hydraulic function"?

Response: this will be reformulated.

L. 120-123: the isohydric vs anisohydric (why use "non-isohydric" rather than anisohydric?) need a bit more explanation, e.g. isohydric species maintain a constant midday $\Psi l$ but also down-regulate their stomatal conductance. It would also be worth mentioning that the spectrum of isohydric and anisohydric behaviours is quite large, with some species having the capacity to oscillate between more-or-less isohydric or anisohydric

behaviours depending on the environmental conditions...

Response: agree, we will elaborate on this

L.131-133: this is a very nice hypothesis! To introduce it, the authors could refer to the work of Rosas et al. 2019 (https://doi.org/10.1111/nph.15684)

Response: thank you, we will read Rosas et al. 2019 and consider including a reference to this paper.

L. 136-138: I think that moving this sentence to line 133 before "In neotropical..." would make the text flow better

Response: agree, this paragraph will be restructured.

L. 138-140: this is very useful contextualisation, maybe it could make it into the abstract?

Response: we will consider mentioning this contextualization in the abstract

L. 153: typo: "it" to "they"

Response: this will be changed

L. 156: what about measurement techniques and errors? Were those also included in the database? I imagine there would be different margins of error depending on the measurement technique. Also, were there quality checks or did all the above described data make it into the database?

Response: no, we did not differentiate between measurement techniques in the database and this could indeed result in differences in uncertainties. However, since the meta-analysis deals with relative changes in a variable of interest or "effect sizes" the absolute values are less important. Furthermore, since our meta-analysis deals with values that are averaged for the different studies, which each included multiple tree species and individuals, the variability due to differences between species and in-

**BGD**

dividuals is much larger compared to measurement uncertainties (see e.g. Santos et al. 2018). It would indeed be very interesting to see how different measurement techniques have an effect on measured drought responses but this is beyond the scope of this study.

L. 157: was the time of day not reported? This would highly impact measurements of stomatal conductance and leaf photosynthesis...

Response: very often the day and time of day were not reported for tree and ecosystem scale responses. For the leaf scale responses (stomatal conductance, photosynthesis and leaf water potential) we always used the value at maximum photosynthesis (at midday), except for pre-dawn leaf water potential, naturally. We will include this information in the methods of the revised version.

L. 157-158: how many different species, genus, and/or different site averages?

Response: this will be included in the revised version.

L. 160-161: the information on how the spatial data were extracted is probably not needed

Response: agree, this will be omitted in the revised version. See also specific comment by referee #2.

L. 171: shouldn't "midday vapor pressure deficit" be "monthly averaged midday vapor pressure deficit"?

Response: yes it should, this will be changed.

L. 173-175: the authors need to mention that this assumption largely ignores variations in root distributions

Response: we will consider including this suggestion.

L. 177-178: please clarify what "ecosystem performance measures" means

Response: we will clarify this in the revised manuscript

L. 179: were all the stomatal conductance measurements made at midday? Also, it is worth mentioning that this relationship assumes a perfect coupling between the stomates and the atmosphere above, i.e. it assumes that the boundary layer conductance to water vapour, gb, is much larger than gs. But in forests with large leaves and dense canopies, decoupling is often observed because gb is relatively small, such that when gs and gb are of similar magnitudes $E \approx 0.5 \times gs \times D$. In the context of this study, it is impossible to estimate what the coupling/decoupling factor is at a given location and/or at a given time, but the authors should mention this (in the context of leaf shedding and flushing?), given their findings

Response: yes, all stomatal conductance measurement were made at midday. Thank you for your insights. We will elaborate on the role of boundary layer conductance in tropical forest canopies in the revised version and also discuss the possible limitations of our crude estimate of leaf transpiration.

L. 184-186: from the text alone, it is very unclear how transpiration was estimated. How does the RMSE represent the linear relationship? Looking at Fig S1, I presume the authors have compiled tree scale measurements of E so, in the analysis, why not just use those measurements (instead of the estimates described by the linear relationship)?

Response: multiple studies that were included in the database reported either maximum sapflux density (at midday) or daily tree transpiration, but not both. We used the linear relationship to calculate daily tree transpiration for the studies that reported only maximum sapflux density. We will elaborate on this in the revised version.

L. 190: my interpretation of equation 3 is that it should only be valid at steadystate. How did the authors ensure steady-state conditions? Were the data filtered depending on VPD?

Response: yes, equation 3 assumes that the system is at steady state. Steady state conditions were not ensured and we doubt whether we could account for this in a meta-analysis. The "instantaneous soil to leaf hydraulic conductance" calculated here should be regarded a measure of whole-tree hydraulic conductance at midday (Love and Sperry, 2018).

L. 192: typo: "rooting zone" to "root-zone"

Response: this will be changed in the revised manuscript

L. 193-194: strictly speaking, difference between $\Psi l$ at midday and $\Psi pre-dawn$ is a proxy of the water gradient within the tree, from the root up to the canopy. For it to equate soil-canopy gradient, further information on tree height would be needed to account for gravitational effects and relate $\Psi pre-dawn$ to $\Psi s$...

Response: this will be changed in the revised manuscript

L. 225-227: I realise it's common to use log response ratios when comparing large amounts of data, but why not directly use the percentage change to quantify drought effect size?

Response: the log response ratio is used to derive the test statistics following Lajeunesse (2011) and then back converted to percentage change.

L. 242-243: given the large variability in hydraulic behaviour observed within a genus, is it reasonable to use the genus average as a proxy here? And how many of the location points are affected by this assumption?

Response: yes, we agree that there can be large within genus variability in wood density and hydraulic behaviour. However, across neotropical tree species about 74% of the variation in wood density can be explained by genus level variability (Chave et al., 2006), so genus level wood density could be regarded a useful proxy. Genus averaged wood density was used in 127 cases out of a total of 786 individuals measured. As the wood density was averaged per study, we believe that the effect of using genus averaged wood density instead of species averaged wood density is small. The alternative would be to not provide a wood density value to this individual, which would probably cause more bias in the study averaged wood density than providing the genus average.

L. 254: the reference to Figure 2a is needed here too

Response: this will be included

L. 260: I find hard to believe that this is an actual result and not simply a product of the methods used to calculate the leaf-level transpiration....

Response: the result that on average leaf transpiration does not change from the wet to the dry season (see also final major comment and figures), follows from the averaging of study level responses of 25 studies from which 11 studies showed a (marginal) increase in leaf-level transpiration and 14 studies a (marginal) decrease in leaf-level transpiration from the wet to the dry season (Figure 5 b). Furthermore, the same result is found when looking at tree scale transpiration which is independent from our calculation of VPD and leaf transpiration. The bias towards studies measuring low wood density trees in sun-exposes canopy positions likely contributes to the overestimation of the dry season decline in stomatal conductance and therefore leaf transpiration (this bias is discussed in the Discussion).

L. 265: but a drop in $\Psi l$ is observed!

Response: yes, this is confusing and will be reformulated.

L. 275-276: the authors could mention that this is in line with the findings of Rosas et al. 2019 along a mesic-xeric gradient (although their study is not on neotropical species)

Response: we will read Rosas et al. 2019 and consider including a reference to this paper.

L. 304 (and later): the "WUEi" notation is inconsistent with the "iWUE" notation used in the introduction

Response: This will be changed to iWUE in the next version of the manuscript.

L. 311: "we observe that" is not needed

Response: agree, this will be omitted in the next version.

L. 315: typo: "marginal" should be "marginally"

Response: this will be corrected.

L. 322: give the ranges of variation?

Response: this sentence is confusing and will be omitted or reformulated in the revised version.

L. 320-328: the findings would benefit from being broken down in terms of the dry season (significant) vs episodic drought (mainly not significant)

Response: agree, only for stomatal conductance and leaf transpiration is the relationship with wood density significant ($p < 0.05$) during episodic drought and shows a similar relationship as during seasonal drought. We will highlight that this is not the case for leaf photosynthesis and midday leaf water potential.

L. 334: the authors need to state that the relationship is not significant...

Response: yes, we will add that this relationship is not significant

L. 336: "intermediate response" is very vague, please reformulate

Response: "intermediate response" will be omitted, "midday ðĺIJŞðĺŚŹ declining parallel to a decline in pre-dawn ðĺIJŞðĺŚŹ"should be a sufficient description of the midday leaf water potential response to declining pre-dawn leaf water potential in the intermediate wood density group.

L. 337-342: these findings are very useful!

Response: thank you.

L. 344-346: How is it "similar"? Fig. 6a seems to show far less significance and way more scatter than Fig. 5b

Response: we agree that this sentence is vaguely formulated. By "similar" we refer to the similarity in the relationships between wood density and the direction and magnitude of leaf and tree scale transpiration. Both show an increase of transpiration from the wet to the dry season in studies that measured high wood density species and a decline of transpiration in studies that measured low wood density species. This will be clarified in the revised version.

L. 356: why "hydrological"? Do the authors mean hydraulic?

Response: yes, this is a vague term and will be omitted in the revised version.

L. 358: please replace "cancelled out" by "offset"

Response: this will be replaced.

L. 392-394: I don't follow this sentence.... these effects can be consistently observed for weeks, and even months? Do the authors mean that leaf effects are typically observed on shorter time scales due to the "life expectancy" of a leaf compared to a tree, or to an ecosystem?

Response: yes, this is partly what is referred to here but not explicitly mentioned. Leaf shedding and flushing can be a mechanism that results in leaf scale responses being visible on shorter time scales but on even shorter timescales also the opening and closure of the stomates. The purpose of these two sentences is to highlight the presence of buffering in the system, in this case because of non-structural carbohydrates, that could result in the observed inconsistencies in drought responses going from the leaf to the ecosystem. This paragraph will be reformulated.

L.405-406: the mention of these "ENSO swings" would be a better fit L. 400, right after the list of references. But what is an ENSO swing? This is never defined...

Response: this will be changed.

L. 398-407: I'm not entirely clear why the increase in the frequency of episodic droughts is not first mentioned in the results section?

Response: this is mentioned in results section 3.2, L289-L290. This will be highlighted in the next version.

L. 538-540: this should come earlier, after L. 120-123

Response: this will be moved.

L. 546: but can also be explained by plant capacitance

Response: this will be changed.

L. 564: typo: missing "and" after "environments"

Response: "and" will be included.

Fig. 1a: the K34 site should be indicated on the map given Fig. 1d and 1e Response: the location of K34 will be indicated on the map in the revised version.

Fig. 1b: this is averaged across sites, right? I wonder whether it would make more sense to actually average the episodic drought months across the whole area of neotropical forests shown on the map. This would potentially reduce sampling biases in concluding that episodic droughts have been increasing in neotropical forests. Alternatively, the authors could consider weighting this by the number of monthly observations per year.

Response: yes, this is averaged across sites. The number of episodic drought months counted in Fig. 1b are independent of the monthly observations retrieved from the literature as all months classified as episodic drought in the time series (1979-2019) at that each site are included. We will consider calculating also the number of episodic drought months across the neotropics in a rectangular grid and compare this to the

counted number of episodic droughts in Fig. 1b to check for sampling bias.

Fig. 1e: visually, it would be very nice if the ENSO index was coloured to match the wet and dry season and the episodic droughts

Response: thank you for the suggestion, we will look into this and see whether we can make the ENSO index more visually interesting.

Fig. 2a: where do the top soil $\Psi$s data come from? The caption says published data, but I didn't find it in the methods?

Response: the references are in the supplementary material (main database excel file) but this is presently not clear. References to the top soil $\Psi$s will be included in the Methods in the revised version.

Fig. 2a and 2b: yes to the mention of capital letters in the legend, but what does it mean when letters are coupled (e.g. AB in the dry season in Fig. 2a) or when the letter A or B appear during episodic droughts?

Response: for Fig 2a this indicates that there is a significant difference in topsoil water potential between the wet season (A) and episodic drought (B) but not between the dry season and either the wet season or episodic drought (AB). This will be elaborated on in the caption of the revised version of this figure.

Fig.2: I imagine the horizontal lines in the box plots show the median, the boxes themselved interquartile ranges, the vertical lines the 5th-95th percentiles and the points are outliers? This needs to be mentioned in the legend

Response: yes exactly. We will explain the ranges of the boxplots in the caption of the revised version of this figure.

Figs. 3 4: what do the horizontal lines represent? Ranges?

Response: the horizontal lines are the 95% confidence interval range, this will be more clearly described in the caption of the revised version.

Additionally, it would be useful: 1. to also mention the number of data points (or average number of data points per study/site) in brackets; 2. to visually separate the variables that were directly retrieved from the literature from those that necessitated further calculations.

Response: we will consider making these changes in the revised versions of these figures.

Figs. 5 6: so bigger points mean smaller errors? Does that also play a role in the weighting of the solid and dashed lines?

Response: yes, the size of the points is determined based on the inverse of the sampling variance of that particular study (i.e. precision) and yes, the model is also constructed using inverse-variance weights. These details will be added to the Methods and in the caption in the revised version of the manuscript.

References

Alexandre, D. Y.: Comportement hydrique au cours de la saison seche et place dans la succession de trois arbres guyanais: Trema micrantha, Goupia glabra et Eperua grandiflora, Ann. des Sci. For., 48(1), 101–112, 1991.

Aragão, L. E. O. C., Malhi, Y., Roman-Cuesta, R. M., Saatchi, S., Anderson, L. O. and Shimabukuro, Y. E.: Spatial patterns and fire response of recent Amazonian droughts, Geophys. Res. Lett., 34(7), L07701, doi:10.1029/2006GL028946, 2007.

Chave, J., Muller-Landau, H. C., Baker, T. R., Easdale, T. A., Hans Steege, T. E. R. and Webb, C. O.: Regional and phylogenetic variation of wood density across 2456 neotropical tree species, Ecol. Appl., 16(6), 2356–2367, doi:10.1890/1051-0761(2006)016[2356:RAPVOW]2.0.CO;2, 2006.

Doughty, C. E., Malhi, Y., Araujo-murakami, A., Metcalfe, D. B., Silva-Espejo, J. E., Arroyo, L., Heredia, J. P., Pardo-Toledo, E., Mendizabal, L. M., Rojas-Landivar, V. D., Vega-Martinez, M., Flores-Valencia, M., Sibler-Rivero, R., Moreno-Vare, L., Jessica Viscarra, L., Chuviru-Castro, T., Osinaga-Becerra, M., Ledezma, R., Javier, E., Arroyo, L., Heredia, J. P., Pardo-Toledo, E., Mendizabal, L. M. and Victor, D.: Allocation trade-offs dominate the response of tropical forest growth to seasonal and interannual drought, Ecology, 95(8), 1–6, doi:10.1890/13-1507.1, 2014. Doughty, C. E., Metcalfe, D. B., Girardin, C. A. J., Amézquita, F. F., Cabrera, D. G., Huasco, W. H., Silva-Espejo, J. E., Araujo-Murakami, A., da Costa, M. C., Rocha, W., Feldpausch, T. R., Mendoza, A. L. M., da Costa, A. C. L., Meir, P., Phillips, O. L. and Malhi, Y.: Drought impact on forest carbon dynamics and fluxes in Amazonia, Nature, 519(7541), 78–82, doi:10.1038/nature14213, 2015.

Fontes, C. G., Dawson, T. E., Jardine, K., McDowell, N., Gimenez, B. O., Anderegg, L., Negrón-Juárez, R., Higuchi, N., Fine, P. V. A., Araújo, A. C. and Chambers, J. Q.: Dry and hot: the hydraulic consequences of a climate change-type drought for Amazonian trees, Philos. Trans. R. Soc. Lond. B. Biol. Sci., 373(1760), doi:10.1098/rstb.2018.0209, 2018.

Hofhansl, F., Kobler, J., Drage, S., Pölz, E. M., Wanek, W., Ofner, J., Drage, S., Pölz, E. M. and Wanek, W.: Sensitivity of tropical lowland net primary production to climate anomalies, Global Biogeochem. Cycles, 28(12), 10585, doi:10.1002/2014GB004934.Received, 2014.

Klein, T.: The variability of stomatal sensitivity to leaf water potential across tree species indicates a continuum between isohydric and anisohydric behaviours, Funct. Ecol., 28(6), 1313–1320, doi:10.1111/1365-2435.12289, 2014.

Lajeunesse, M. J.: On the meta-analysis of response ratios for studies with correlated and multi-group designs, Ecology, 92(11), 2049–2055, doi:10.1890/11-0423.1, 2011. Love, D. M. and Sperry, J. S.: In situ embolism induction reveals vessel refilling in a natural aspen stand, Tree Physiol., 38(7), 1006–1015, doi:10.1093/treephys/tpy007, 2018.

Martínez-Vilalta, J., Poyatos, R., Aguadé, D., Retana, J. and Mencuccini, M.: A

new look at water transport regulation in plants, New Phytol., 204(1), 105–115, doi:10.1111/nph.12912, 2014. Meinzer, F. C., Smith, D. D., Woodruff, D. R., Marias, D. E., McCulloh, K. A., Howard, A. R. and Magedman, A. L.: Stomatal kinetics and photosynthetic gas exchange along a continuum of isohydric to anisohydric regulation of plant water status, Plant Cell Environ., 40(8), 1618–1628, doi:10.1111/pce.12970, 2017.

Santos, V. A. H. F. dos, Ferreira, M. J., Rodrigues, J. V. F. C., Garcia, M. N., Ceron, J. V. B., Nelson, B. W. and Saleska, S. R.: Causes of reduced leaf-level photosynthesis during strong El Niño drought in a Central Amazon forest., 2018.

**Fig. 1.** The new ERA5 vapor pressure deficit at midday (12:00) compared with vapor pressure deficit measured at 12:00 at 9 flux tower sites in the study area. The dashed line represents the 1:1 line and the so

**Fig. 2.** Updated Figure 2 with new vapor pressure deficit estimates.

[Figure]

**Fig. 3.** Updated Figure 3 with new vapor pressure deficit estimates.

---

## Author Comment (AC2) · 18 Feb 2020

Response to Referee #2

In this study, Janssen et al perform a meta-analysis to look to drought impacts on carbon and water exchange across scales ranging from leaf, to plant, to ecosystem in neotropical rainforests. In particular, the authors contrast physiological responses from seasonal water stress versus 'episodic' drought events. In doing this, the authors also look at wood density as a proxy for plant physiological responses. Its clear that the

authors put in a significant amount of work through compiling 138 studies across 229 sites and this study will clearly contribute to the physiological literature on forest drought responses. I do have some major revision suggestions before publication which are summarized here and, in some cases, elaborated in the line specific comments below.

Response: thank you for your elaborate review of our manuscript and acknowledging the work that was put into creating the database behind the meta-analysis. We believe that your comments and suggestions will greatly enhance the quality of the manuscript. Below we will respond to the major and minor comments raised by you in your report.

1) I think that the manuscript would benefit from reframing of the dry season 'drought' as a routine period of decreased water availability. When I think of droughts, I think of a prolonged period of abnormally low rainfall. Given that dry seasons occur every year, I don't see them meeting this definition. This reframing would provide a nice platform to discuss physiological responses to routine (seasonal) stress, such as phenology, versus episodic stress and can help get at important physiological mechanisms. This would involve some substantial reworking of the text, but I think it would really help the story line.

Response: we agree that seasonal drought would not meet the criterium of being a prolonged period of abnormally low rainfall and we highlight this distinction with episodic drought and multi-year drought in the Introduction (L50-L54). However, the term "seasonal drought" is widely used in the literature (Esquivel-Muelbert et al., 2017; Rowland et al., 2013; Stahl et al., 2013) and leaf, tree and ecosystem processes similar to episodic drought are operating during seasonal drought. We discuss the differences between seasonal and episodic drought in the Discussion section 4.3 and mention how for example phenology is likely driving the observed seasonal responses in leaf flushing, shedding and stem growth. In the revised version of the manuscript we will highlight this distinction between seasonal and episodic drought from the start and we will, where needed, adapt the text to improve the story line.
2) Some of the methods are confusing and I think elaborating a bit more, providing and providing a table might help. See line specific comments below

Response: referee #1 also provided suggestions to improve the readability of the Methods, we will adapt this in the revised version. Concerning the table, see specific comments below.

3) The Figure legends need to be more descriptive of all the features in the figures

Response: we will improve the figure legends and captions in the revised version, see also specific comments about figures and comments on figure captures by referee #1.

4) Overall the manuscript does an impressive job discussing a range of processes, but the reader might be more attentive if it were a little shorter. Where possible, I would suggest the authors minimize extraneous discussion. In particular, I think talking about isohydricity requires significant motivation for a general audience (which is not provided), so I would cut this text.

Response: thank you. We agree that the readability of the paper would improve if it were shorter. While preparing the revised version we will critically examine every paragraph to reduce the length of the manuscript. In the revised version of the manuscript we will no longer use the concept of isohydric and non-isohydric behaviour because, as you mention, it is not suitable for the broad audience of Biogeoscieces (specific comment L121). We will instead focus on the mechanisms behind the observed drought responses because this is more interesting to the audience and does not constrain us to refer to two strategies so we can focus on the continuum of hydraulic behaviour.

Line-specific comments:

L21-22 There is nothing to back up this statement on LSMs. I suggest the authors remove it

Response: See also specific comment of referee #1. The statement will be rephrased to: ""We present new insights into the functioning of tropical forest in response to

drought and present novel relationships between wood density and drought responses that can help guide the parametrization of land surface models."

L35 Khanna et al 2017 Regional dry-season climate changes due to three decades of Amazonian deforestation

Response: thank you for the literature suggestion, Khanna et al. 2017 will be added in the revised version of the manuscript.

L50 after going through the MS, I am confused about where the multi-year drought is presented in the authors analysis

Response: no it is not presented, see for the justification of omitting multi-year droughts L70-71

L53-54 This is exactly why I would argue that 'seasonal drought' is a misnomer

Response: see response to major comment #1.

L121 The concept of isohydricity/anisohydricity will likely not be familiar to a broad audience at Biogeosciences. I would encourage the authors to eliminate the jargon and focus on the mechanisms of interest (see Martinez-Vilalta 2016 "Water potential regulation, stomatal behaviour and hydraulic transport under drought: deconstructing the iso/anisohydric concept") or else devote more space to describing isohydric behaviour

Response: See response to major comment #4.

L131-132 But the authors actually try and demonstrate that wood density is usable as a proxy first, so isn't this more of a hypothesis?

Response: we agree this is a confusing sentence. We will reformulate this sentence.

L145 where does the multiyear drought aspect come in that the authors mentioned in the intro? Intro overall: The introduction is on the longer side (as is the manuscript) and the three separate sections do not have a smooth transition. I would try to com-

bine 1.2 and 1.3 and streamline the text. Given that sections 1.2-1.3 are more about the mechanisms, I would put thes sections first and then say that different types of physiological stress also impact C and W fluxes including seasonal, routine decreases in water availability versus droughts

Response: thank you for your suggestions on restructuring the Introduction. We will shorten and restructure the Introduction in the revised version of the manuscript.

L152 ERA5 citation?

Response: a citation to the ECMWF website will be provided in the revised version.

L153-155 It would be good for the authors to say how many studies were associated with each of these diagnostics and how many studies had multiple observations, maybe a table would be useful?

Response: we will add a table with a summary of the database in the Supplement of the revised version of the manuscript.

L157-159 I am confused about how observations were calculated/recorded across scales? Could the authors elaborate? For example, were ecosystem-level measurements independent from leaf level, or were different levels calculated using observations from different scales by the authors? Both more details and a table detailing number of measurements for each diagnostic and number of studies that cover multiple diagnostics and scales would help

Response: the scales refer to the native "resolution" on which measurements were provided in the source papers. Many studies that measured leaf an tree scale responses provided data on the individual tree level, giving also a species and genus name. For these studies we could link these individual observations to a wood density value. However, many studies reporting ecosystem level responses provided data on the ecosystem level, e.g. 1 ha of forest. This will be elaborated on in the revised version of the manuscript.

L161-162, this doesn't need to be included

Response: agree, this can be omitted. See also specific comment by referee #1.

L166-167 This is a huge amount of work, I commend the authors

Response: thank you.

L170 it would be good to say the spatial resolution and include a citation for on ERA5 (30km?)

Response: yes indeed, 0.25 degrees is 27.75 km at the equator. This information will be added in the revised version.

L171 monthly average midday VPD?

Response: yes, this is the monthly averaged VPD at midday (12:00), see comments by referee #1 on VPD.

L187 how was this error estimated?

Response: this was estimated as the RMSE (3.25 kg dm-2 d-1) divided by the mean daily transpiration rate (9.56 kg dm-2 d-1) = 0.34 = 34%. This will be elaborated on in the revised version.

L219 Figures should be renumbered so Fig 2 comes after Fig 1

Response: this reference to Figure 4 is not necessary and will be omitted in the revised version of the manuscript.

L208-209 It would be good to include a possible caveat about uncertainty associates with ERA5 soil moisture (which I presume is modeled)

Response: this will be included in the next version of the manuscript.

L212-213 why the 65% and 10% quantiles? Did the authors test the sensitivity of their results to this assumption?

Response: we opted for a threshold that provided a reasonably strict episodic drought definition while still yielding a large enough sample size for the statistical analysis to differentiate between episodic drought and a regular dry season. We also tested a wide episodic drought threshold of 15% and a narrow episodic drought threshold of 5%. The wider episodic drought definition resulted in a decline of the sample size for the wet season – dry season comparison as more dry season months were classified as episodic drought. Furthermore, the p-values of 14 out of 23 variables declined but none of the previous significant variables became not significant in the wide (15%) threshold wet season – dry season comparison. The wide episodic drought definition resulted in the increase of the sample size in the dry season - episodic drought comparison but also a decline in the p-values of 15 out of 23 variables, while none of the previously significant variables became not significant in the dry season – episodic drought comparison. The narrow episodic drought definition (5%) resulted in larger sample size for the wet season – dry season comparison compared to the baseline (10%) definition because more episodic drought months are now classified as dry season, and the increase of p-values in about half (12 out of 23) of the variables, while one variable (soil-leaf hydraulic conductance) became not significant in the wet season – dry season comparison. Furthermore, the narrow definition resulted in a decline in sample size for almost all variables (19 out of 23) and a decline of the p-values in 15 out of 23 variables in the dry season – episodic drought comparison with 4 previously significant variables now showing no significant change (soil-leaf hydraulic conductance, leaf transpiration, leaf photosynthesis and ecosystem water use efficiency). These results confirm that our analysis of seasonal drought is quite robust, with no major changes in the magnitude and direction of change of any variables in response to seasonal drought with different threshold values for episodic drought. Also the responses to episodic drought show no major changes in direction or magnitude but we observe a decline in significance levels in some variables, mainly because of a reduction in sample size. We will add these results to the supplementary material and discuss the implication of the threshold in the discussion.

L220 Could a study be both a dry season and episodic drought? I am a little confused about the partitioning. Also, where does multi-year drought come in?

Response: the partitioning is mentioned in L214, 10% of the driest dry season months were classified as episodic drought (and not anymore as dry season). We will elaborate on this in the next version of the manuscript.

L225 did the authors check to see if it was necessary to log the response? Where did the ENSO data come from?

Response: the log-response ratio was calculated because this is the standard in the method used: the log transformed ratio of means (Lajeunesse, 2011). The ENSO data was retrieved from NOAA, a reference to this dataset will be included in the revised version.

L254 how can predawn wp be positive 0.22?! Please check for a typo

Response: the minus sign unfortunately ended up on the previous page.

L281 Could the authors include Reco in some of the figures, they refer a change in Reco several times in the text but no visuals are provided

Response: Reco is the same as Ecosystem respiration in Figure 3 and 4. We will consider removing the abbreviation altogether.

L363 typo include 'us'

Response: yes, this is a typo. This will be corrected.

L370 denoted iWUE previously

Response: yes, this will be corrected.

L401 it is also true that there are more observations post 2000. The authors should discuss how this might impact their results

Response: this is discussed in response to the first main comment of referee #1. We

will elaborate on this sampling bias in the revised version of the manuscript.

L419 it would be nice to put these numbers in a physiological context

Response: it is not entirely clear what is meant by this comment. Could you clarify?

L451-458 I think there is a really nice opportunity to contrast tree physiological strategies when exposed to routine stress (the dry season) versus drought that the current narrative doesn't allow for when both are classified as drought

Response: We agree that this comparison is important and, although we prefer sticking to the term seasonal drought (see further), comparing seasonal drought (routine stress) to episodic drought is one of the main objectives of the meta-analysis. There are basically two reasons why we use the term seasonal drought: 1) the term "seasonal drought" is commonly used in the literature and 2) the difference in drought conditions between the wet season and dry season is comparable to the difference between a regular dry season and episodic drought (Figure 2).

L461 is it the short timescale, or that fact that the plants are used to this type of stress and use phenology to deal with it?

Response: it is not clear what is driving these tree scale responses to drought and on which time scale these responses operate (Doughty et al., 2015; Hofhansl et al., 2014). The purpose of this sentence was to give a potential explanation for the lack of leaf litterfall and leaf flushing responses during episodic drought. We argue that this could be the result of the phenology time-scales and the amplitude of leaf exchange overwriting the more subtle responses to drought, if there are any. This is quite speculative and will be reformulated in the revised version of the manuscript.

L464 Or maybe title: how do we scale from the leaf to the ecosystem? This is a universal problem in ecology that researchers across many subdisciplines will sympathize with

Response: thank you for the nice suggestion. We will adapt this subtitle in the revised

version.

L502 see previous suggestion about isohydricity

Response: noted.

L536 This isn't a great comparison. The authors of this manuscript analyze site-specific data whereas the spatial scale of the Konings study is o(100km). For me, this paragraph does not contribute much to the study and in general I think the isohydricity framework is not useful here (and otherwise)

Response: we agree this is not a great comparison, it will be omitted in the revised version.

L575 LSMs are brought up in only in the abstract and conclusion. It doesn't add to the discussion and I would remove this

Response: noted, this will be removed.

L582 How should they be used to benchmark LSMs? If the authors insist on including this, please the need to spell out the methodology rather than throwing it in as a concluding sentence Other relevant citation: Detto 2018 "Resource acquisition and reproductive strategies of tropical forest in response to the El Niño-Southern Oscillation"

Response: the reference to LSMs will be removed.

Figures: Fig. 1 panel a make lat/lon bigger b) its really hard for me to wrap my head around what the authors mean by this metric, can they elaborate? d) What do the dots mean? Please describe this in the legend and also detail what dark and light gray correspond to legend (e) it would be good to remind the reader what the positive/negative ENSO index means

Response: we will include these suggestions in the revised version of the Figure caption.

[Figure]

L1131 "terrestrial" isn't capitalized

Response: noted, this will be changed.

Fig. 2 can the authors denote the sample size above each category in the figure? For example, does n=3 for Episodic drought soil-leaf hydraulic conductance in panel b? Make sure to describe the figure fully (detail quantile boxes, median line, error bars, and outliers) in the legend

Response: the suggested changes will be made in the next version. See also comments on this figure by referee #1.

Fig. 3-4 I would combine these two figures into 1 2-column 3-row figure. I generally really like this format and found it very effective in the Ainsworth review. Great job describing all aspects of the figure in the legend

Response: we will look into merging Figure 3 and 4 in the revised version of the manuscript.

Fig 5-6 I would combine these figures into a 2-c 3-r figure

Response: we will look into merging Figure 5 and 6 in the revised version of the manuscript.

The point size is the inverse of the sample standard error of the effect size in the study.—> so studies with a smaller SE have a larger dot? Moderator= independent variable?

Response: this will be changed and explained in the text, see also the response to the final comment of referee #1.

Fig 5 legend, make sure to walk the reader though each panel Response: we will reformulate the Figure 5 legend in the next version.   References

Doughty, C. E., Metcalfe, D. B., Girardin, C. A. J., Amézquita, F. F., Cabrera, D. G., Huasco, W. H., Silva-Espejo, J. E., Araujo-Murakami, A., da Costa, M. C., Rocha, W.,

Feldpausch, T. R., Mendoza, A. L. M., da Costa, A. C. L., Meir, P., Phillips, O. L. and Malhi, Y.: Drought impact on forest carbon dynamics and fluxes in Amazonia, Nature, 519(7541), 78–82, doi:10.1038/nature14213, 2015.

Esquivel-Muelbert, A., Baker, T. R., Dexter, K. G., Lewis, S. L., ter Steege, H., Lopez-Gonzalez, G., Monteagudo Mendoza, A., Brienen, R., Feldpausch, T. R., Pitman, N., Alonso, A., van der Heijden, G., Peña-Claros, M., Ahuite, M., Alexiaides, M., Álvarez Dávila, E., Murakami, A. A., Arroyo, L., Aulestia, M., Balslev, H., Barroso, J., Boot, R., Cano, A., Chama Moscoso, V., Comiskey, J. A., Cornejo, F., Dallmeier, F., Daly, D. C., Dávila, N., Duivenvoorden, J. F., Duque Montoya, A. J., Erwin, T., Di Fiore, A., Fredericksen, T., Fuentes, A., García-Villacorta, R., Gonzales, T., Guevara Andino, J. E., Honorio Coronado, E. N., Huamantupa-Chuquimaco, I., Killeen, T. J., Malhi, Y., Mendoza, C., Mogollón, H., Jørgensen, P. M., Montero, J. C., Mostacedo, B., Nauray, W., Neill, D., Vargas, P. N., Palacios, S., Palacios Cuenca, W., Pallqui Camacho, N. C., Peacock, J., Phillips, J. F., Pickavance, G., Quesada, C. A., Ramírez-Angulo, H., Restrepo, Z., Reynel Rodriguez, C., Paredes, M. R., Sierra, R., Silveira, M., Stevenson, P., Stropp, J., Terborgh, J., Tirado, M., Toledo, M., Torres-Lezama, A., Umaña, M. N., Urrego, L. E., Vasquez Martinez, R., Gamarra, L. V., Vela, C. I. A., Vilanova Torre, E., Vos, V., von Hildebrand, P., Vriesendorp, C., Wang, O., Young, K. R., Zartman, C. E. and Phillips, O. L.: Seasonal drought limits tree species across the Neotropics, Ecography (Cop.)., 40(5), 618–629, doi:10.1111/ecog.01904, 2017.

Hofhansl, F., Kobler, J., Drage, S., Pölz, E. M., Wanek, W., Ofner, J., Drage, S., Pölz, E. M. and Wanek, W.: Sensitivity of tropical lowland net primary production to climate anomalies, Global Biogeochem. Cycles, 28(12), 10585, doi:10.1002/2014GB004934.Received, 2014.

Lajeunesse, M. J.: On the meta-analysis of response ratios for studies with correlated and multi-group designs, Ecology, 92(11), 2049–2055, doi:10.1890/11-0423.1, 2011.

Rowland, L., Stahl, C., Bonal, D., Siebicke, L., Williams, M. and Meir, P.: The Response of Tropical Rainforest Dead Wood Respiration to Seasonal Drought, Ecosystems, 16(7), 1294–1309, doi:10.1007/s10021-013-9684-x, 2013.

Stahl, C., Hérault, B., Rossi, V., Burban, B., Bréchet, C. and Bonal, D.: Depth of soil water uptake by tropical rainforest trees during dry periods: Does tree dimension matter?, Oecologia, 173(4), 1191–1201, doi:10.1007/s00442-013-2724-6, 2013.

---

## Author Response (AR1)

**Response to the editor**

I have now received two very constructive reviews, both liked components of your manuscript but both raised a number of important issues that need to be carefully revised before this manuscript can be considered for publication.

In particular, the highlighted the sampling bias issue, which will need to be discussed very clearly in the revised text. I think adding text discussing a sensitivity test to the assumed drought thresholds is warranted. R1 highlighted the problematic nature of the author's drought threshold which leads to non-physical oscillating drought/non-droughts. This also likely leads to a gross over-estimate of drought occurrence too? Would it make sense to compare to something like SPI, which can be used to better integrate month-to-month behaviour. R1 highlighted your VPD range, which you've now fixed, I think it would be good to show that the new estimated values are consistent with any flux records you have in a supplementary figure to satisfy a reader. Both reviewers commented on a general lack of clarity in your methods, please carefully revise this.

Response: thank you for allowing us to revise our manuscript and for your careful review of the referee comments and our manuscript. Below we will reply point by point (in red) and in detail to the comments and suggestions raised by you and by the two referees. At the end of this document is the revised manuscript with the changes made visible using the "track changes" option in MS Word.

We have made a few major revisions in the analysis and the manuscript following the (shared) comments and suggestions raised by you and by the two referees:

- the sampling bias issue was indeed raised by both referees and is largely based on a misunderstanding on how the episodic drought months are counted. We have now included a second analysis where we counted the number of episodic drought months per year in a regular 1° grid covering our study area, it shows a similar trend (new Figure S4). By providing more context about the counting of the episodic drought months per year in the Methods, Results and in the Figure captions, we hope that his issue is now resolved.
- 2) we have now resolved the problem of the timeseries oscillating between wet and dry seasons highlighted by referee #1 by omitting single wet season months when they are in between two dry season months, and vice versa. This has resulted in extended periods of wet season and dry season that better capture the "actual" dry season at which referee #1 was referring to in the report (new Figure 1). This new smoothening method has also made the 65% quantile threshold, which was confusing to both referees, obsolete.
- 3) we performed a sensitivity analysis by rerunning the meta-analysis using two additional episodic drought thresholds (see Methods). The results of this sensitivity analysis are shown as two additional Figures in the Supplement (Figure S8 and S9).
- 4) The issue with the VPD has now been fixed, the methodology about retrieving midday VPD has been updated and the comparison of monthly averaged midday VPD from ERA5 with the monthly averaged midday VPD from the flux towers has been included in the Supplement (Figure S1).

I also have a number of key methodological issues that I really am not clear on, in particular the assumptions you make when comparing fluxes, in particular how you're controlling for LAI differences between species? Similarly, you're comparing fluxes that vary diurnally / seasonally / annually, so, how are you control for this to ensure your comparisons are meaningful? Would a maximum stomatal conductance make more sense than a direct measurement of stomatal conductance? If this information was explained, then I apologise - I note you say they are midday values but I don't recall reading this in your paper. I am similarly uncomfortable with the idea that you generate a number of derived metrics (e.g. WUE) from fluxes that are not measured (e.g. transpiration). In your response to R1 you note the use of a linear reln, where was this shown? Your assumption of perfect coupling to get transpiration is likely to be a gross error (see citations below).

I have listed a number of points that I also observed whilst reading through your paper below.

\* Can I ask you to check this statement: "Multi-year droughts are defined as a more permanent reduction of precipitation spanning years to decades, as projected by some climate model simulations (Boisier et al., 2015; Malhi et al., 2009)." I'm happy to be proven wrong, but my understanding was that coupled-climate models fail to project multi-year droughts.

Response: we cannot find any references that suggest that multi-year droughts are projected to become more frequent in coupled-climate models. The references to Boisier et al. 2015 and Malhi et al. 2009 are more about dry season lengthening, which we agree is not the same. We have omitted this statement from the revised version.

\* Line 100: "aboveground net primary productivity" - why only aboveground? This seems a curious distinction to make, why wouldn't drought also impact below-ground growth?

Response: we chose to use above-ground NPP because monthly estimates of root productivity are very scarce in the neotropics. In the new version of the manuscript we have also included total NPP in addition to ANPP in the meta-analysis.

\* Line 124: "as suggested by remote sensing analysis" - it would be good to include a word like "derived" in there, as the estimate of isohydricity from VOD is not a direct measurement. Further, the text at the end of that paragraph sets up a binary: isohydric vs anisohydric, in reality, there is more of a spectrum of behaviour, it would be nice if the text better reflected this.

Response: this sentence is now omitted from the manuscript. We followed the suggestion of referee #2 to omit mentioning the concept of isohydric vs non-isohydric behaviour and focus on the mechanisms instead.

\* In your meta-analysis search terms you have "gross ecosystem productivity", does that mean you ignored studies that report gross primary productivity, or Asat?

Response: no, we also used reported values of gross primary productivity, which are used interchangeably in the literature. We now use GPP in the manuscript because it makes more sense comparing GPP and NPP and not GEP and NPP.

\* Line 196: technically this is not ANPP, as your estimate is biomass change - turnover of tissues; ANPP is simply a summed growth term. It might be better to describe this as something more aligned with biomass? I realise that the approach you've taken is extremely common in the literature. Response: this might be a misunderstanding, as also in our calculation ANPP is a summed growth term (stem growth + canopy growth). We have adapted this sentence to also include total NPP and highlight that ANPP and NPP are just a sum of these growth terms.

\* It would be good to report in a table how many of these metrics, collected and derived you have for each of the 229 sites. I found myself wanting to see this while reading the methods. It is very clear from looking at your actual database that the collected variables vary considerably across sites. I would like to see the addition of a summary table of some kind with a revised manuscript.

**Response: this table has now been added to the Supplement (Table S2).**

\* I'd prefer you wrote the derived equations you have calculated for your meta-analysis out (with units) in the methods, as it currently isn't clear, e.g. iWUE, WUE, etc. Also, what are these stomatal conductance measurements given that this isn't a static property - how can you compare this across sites if they are measured at different times of day? Is it the maximum stomatal conductance (Gsmax)? If so, please say so. I'm also uncomfortable with you assuming transpiration = gs X D, for an ecosystem known to have leaves that are not well coupled to the boundary layer (see Jarvis and McNaughton 1986; De Kauwe et al. 2017 Biogeosci.). Why not simply constrain your analysis to quantities that you have direct estimates of, as opposed to introducing new assumptions/errors?

Response: we have now added five more equations to the Methods (section 2.2). For all leaf scale measurement (except pre-dawn leaf water potential) we used midday values (~12:00 local time), this was previously not clear but is now highlighted in the Methods (section 2.1, first paragraph). We also more explicitly mention that our estimation of "potential" leaf transpiration is not to be confused with actual leaf transpiration because of the decoupling of atmosphere and leaf boundary layer. Nonetheless, we think that including these derived measures (some of which are simply a ratio of two measures directly retrieved from the literature) still ads value to the meta-analysis of leaf, tree and ecosystem scale responses to drought.

\* On a related point, does varying LAI not confound direct comparison between species? How are you controlling for this?

Response: we do not take any measure of leaf area, crown cover or LAI into account into the metaanalysis and we do not directly compare species or individuals. To avoid these sources of uncertainty we have compared study averages of measures, giving the study averaged response to drought. We did this to answer: 1) what is the average response observed across all studies? and 2) can we explain the variability in the responses among studies? All the biases in large or small trees, high LAI or low LAI ecosystems is present in these studies but we did not aim to calculate the absolute response of these measures, just the study averaged response and the average percentage change across all studies. We have included and reformulated a few sentences in the revised version of our manuscript to highlight this difference.

\* In the meta-analysis text, you need to be a lot clearer how uncertainty is being estimated across sites. You also need to more clearly describe what you mean "treatment" to be? In some instances, this would be simply wet-season vs dry-season? But what about the experiments that have imposed a manipulation. Are all of these things being compared at once, I didn't find the text clear on these key points.

Response: we have only made two comparisons of natural drought conditions: wet season – dry season and dry season – episodic drought. Therefore the "treatment" in the meta-analysis is either the dry season (and wet season the control) or episodic drought (when dry season is the control).

We have now included two sentences in the Methods (section 2.4, 1st paragraph) in which we explain in more detail what exactly we are comparing.

\* "Furthermore, dry season days are often less cloudy resulting in higher net radiation," - where exactly is this statement shown/supported? This may be true, but Fig 2 does not show this.

Response: agree, this is not shown in Figure 2, this statement is now omitted from the Results.

\* Fig 2 needs the number of samples/studies above the box-plots.

Response: the number if samples has been included in the new Figure 2.

**Response to Referee #1**

Janssen et al. examine whether a meta-analysis of leaf-, tree- and ecosystem-level data can help understand, and predict, neotropical rainforest responses to drought. They ask two questions: (i) how does drought impact the vegetation from the leaf to the ecosystem scale?, and (ii) can different hydraulic behaviours at different locations or among species explain differences in the responses to drought? They find that episodic drought effects compound on dry season effects at both the leaf and tree scales. However, vegetation responses are buffered at the ecosystem scale and, notably, are often not significant during episodic drought. Finally, independently compiled wood density data are used to explain some of the variability observed at the leaf and tree scales during the dry season (and to a lesser degree during episodic droughts). I commend the authors for this undertaking (138 studies!) and for the quality of their writing. The study will make an important contribution by explaining the ecophysiological impacts of drought on a key region's rain forests, at a range of scales. However, I have several major methodological concerns that should be addressed in revision.

Response: thank you very much for your extensive report. We appreciate the thorough review of our manuscript and your acknowledgement of the relevance of our work. Below we reply to the comments raised in the referee report and highlight where the changes in the manuscript have been made following your comments and suggestions.

—————————- Main comments ———————-

My fist observation is that, according to the number of measurements / estimates compiled by the authors, episodic droughts data (624) represent 9% of the total amount of data (6956) and to 17% of the dry season data (3006). This feels like a very high number of episodic drought observations compared to the rest of the observations. Looking at Figure 1b and c, the number of observations per months appears biased towards the more recent years. Does this bias explain the frequency increase in the average of episodic drought months per year in the more recent years?

Response: yes, there is a sampling bias in our meta-analysis towards the more recent years and this does result in more months being classified as episodic drought months than would be expected based on the 10% cut-off point. Furthermore, in this summary we erroneously included multi-monthly data from forest plots that were classified as either episodic drought or dry season months for comparison, these numbers have now been updated. The sampling bias however does not explain the trend of increased episodic drought months per year (Figure 1 b) because the entire time series of ERA5 soil moisture was used to count the episodic droughts at each site, whether we have field data in those months or not (see also specific comment on Figure 1). We have now also included an analysis on the number of episodic droughts per year across the lowland moist neotropical forest biome (new Figure S3 and Table S3) to remove the spatial sampling bias (because

of clusters of sites in some places) and find again a significant increase of episodic drought months over the timeseries, together with an increase of midday air temperature and VPD. We hope that in this way we removed all possible confusion for the reader about whether the increase of episodic drought months per year is driven by the increase of months per year with data from the literature (which is not the case).

Looking at Fig. 1d, I am also questioning the definitions used for the wet season, dry season, and episodic droughts. For example, in 2000, the K34 site starts of by being in the wet season for 5 months, then in the dry season for 1 month, then in the wet season again for 1 month, then in the dry season for 2 months, wet season for 2 months, dry season for 1 month, wet season for 2 months, dry season for 1 month.... This pattern of oscillating wet and dry season is seen repeated within the following years, but how likely is it to represent the "real" wet and dry season? And so, how can dry season effects on the vegetation be captured on time scales that make sense?

Response: the separation of months into wet and dry season was indeed based on the depletion or replenishment of ERA5 soil moisture for a specific site. The rationale behind our definition is that when evapotranspiration exceeds precipitation, there is a precipitation deficit resulting in the depletion of soil moisture. This definition is often used to distinguish wet and dry season in neotropical forests and also used to calculate drought metrics such as the cumulated water deficit (Aragão et al., 2007). We have now included a new "smoothening" operation on wet season, dry season classification by labelling a single dry season month in between two wet season months as a wet season month, and vice versa (see Methods 2.3, end of 1st paragraph). This procedure has resulted in longer sequences of months falling in either the wet season or dry season (new Figure 1 d) that make more sense ecologically and it has also made the 65% quantile threshold obsolete.

Again, if we look at the dry season and episodic drought between 2015 and 2016, we see a transition from episodic drought to wet season although the relative extractable soil water is very close to 0. I understand from the authors' definition of the wet season that this is because the soil moisture has started to be replenished. Realistically, if the vegetation had just gone through an episodic drought, then would the next month's measurements of stomatal conductance, photosynthetic rate, etc. be representative of a wet season month? Therefore, owing to potential hydraulic function damage sustained during the drought, the authors might want to rethink their definitions of the wet and dry seasons, as well as of the episodic droughts, in terms of what makes sense when considering potential multi-weeks (but not multi-years) legacy impacts on the vegetation. One solution would be to classify some of the data as being within "recovery months" (i.e. from a drought or from the dry season to the wet season) and to analyse them separately.

Response: on the leaf scale, the available literature reported no clear legacy effects of episodic drought on stomatal conductance, leaf water potential and photosynthesis (Alexandre, 1991; Santos et al., 2018). However, on the tree scale there are legacy effects reported, for example reduced hydraulic conductance and transpiration directly following episodic drought (Fontes et al., 2018) and changes in stem growth and leaf flushing (Doughty et al., 2014, 2015; Hofhansl et al., 2014). We discuss observed legacy effects reported in the literature in the Discussion (L486-L496) and acknowledge that the method used by us is only able to capture instantaneous responses and not the legacy effects, which is a limitation.

The authors should also consider testing the sensitivity of their results to different quantile threshold definitions for what consists in the wet and dry season, as well as in an episodic drought.

Response: the lack of a "sensitivity analysis" was also noted by referee #2 L212-213. In the revised version of the manuscript we have now included as sensitivity analysis (see Methods) and we included the results of this sensitivity analysis in the results (referring to new Figure S8 and S9), and discuss its implications in the Discussion.

My second concern relates to the method used to calculate the percentage changes shown in Figs. 3 and 4. It is very clear from Figs. S2, S3, and S4 that wood density is a good proxy for leaf- and treelevel hydraulic behaviour. So why not cluster the analysis of the rates of change by types of wood density (e.g. low vs high), to ensure that opposite types of leaf- and tree- level behaviours are not compensating and cancelling each other out when looking at the rates of change? I understand that this is what Figs. 5 and 6 attempt to do, but I do think the broader narrative would be more successful had the meta-analysis differentiated between isohydric and anisohydric behaviours from the start. Clustering by behaviour might also help reconciliate and explain the current inconsistencies in the findings from the leaf-level up to the ecosystem scale.

Response: we agree that merging the drought responses of all the species and functional groups present in the database results in the loss of the variability in responses observed. In the case of transpiration this merging indeed results in that we observe on average no significant changes in transpiration from the wet to the dry season (Figure 4) while studies that measured mainly high wood density species or low wood density species did show a significant increase or decrease in transpiration, respectively (Figure 5 and 6). As you mention, showing this variability is the purpose of Figure 5 and 6 while for Figure 3 and 4 the aim is to show the average response. We did consider splitting the data shown in Figure 3 and 4 in studies measuring mainly isohydric and non-isohydric species, however, this distinction is not easily made. As can be seen in Figure 5 and 6, there are not really two clusters of hydraulic behaviour but it is rather a continuum, related to the continuum from strictly isohydric to extreme non-isohydric behaviour that is observed in plants globally (Klein, 2014; Martínez-Vilalta et al., 2014; Meinzer et al., 2017). We think that arbitrary splitting the data in isohydric studies will not help to reconcile the inconsistencies in drought responses observed from the leaf to the ecosystem scale.

My third point has to do with the VPD values used to estimate changes in leaf level transpiration. The leaf-level transpiration is estimated using the relationship  $E = gs \times D$  where D is VPD. Here, the authors use monthly averaged atmospheric midday VPD derived from the ERA5 reanalysis data. I am surprised because the VPD values present in the database are very low, with a maximum of 2.35 kPa across all 6956 data points and the 95th percentile < 1 kPa. Given that > 50% of the total data is classified as corresponding to either the dry season or to an episodic drought, I would at least expect the 95th percentile value of the average monthly midday VPD to be > 1 kPa! It is unclear to me whether these low values are due to using the Buck method to calculate VPD, or to the ERA5 data themselves. Additionally, using atmospheric VPD rather than leaf-to-air VPD (which the relation E =gs × D is designed for) ignores feedback effects from the leaf to the atmosphere above. When plants transpire during a drought (or a heatwave), they also cool the air immediately above them, leading to lower leaf-to-air VPD than atmospheric VPD. One finding of this paper is that "the data shows no significant decline in leaf transpiration from the wet to the dry season [...] as the average increase of VPD from the wet to the dry season is of the same magnitude as the decline of stomatal conductance". Instead, higher estimates of midday VPD (e.g. from a different reanalysis product) could lead Janssen et al. to predict an increase in transpiration during the dry season. Or, conversely, using leaf-to-air VPD might lead to a smaller magnitude increase in leaf-to-air VPD than the decline in stomatal conductance, thus leading to predicting a reduction in leaf-level transpiration in the dry

season! It is very hard to tell what the implications of the VPD estimates are, but they currently make it hard to trust the leaf-level estimates of transpiration, Potential ways forward are:

1. to use a different method than the Buck method and to quantify the uncertainty;

2. to compare the current VPD estimates with different reanalysis products (e.g. ERA-Interim which has been evaluated more) or other products, such as the CRU data, and to quantify the uncertainty;

*3. to calculate a proxy of leaf-to-air VPD using atmospheric VPD and leaf water potential to account for a degree of leaf-atmosphere feedbacks.*

Response: thank you for pointing out the very low VPD that we used to calculate leaf transpiration in our meta-analysis! After reviewing our pre-processing steps that we used to obtain the ERA5 VPD at every site, we found the mistake that resulted in these low VPD estimates: instead of using local time (12:00) temperature and dewpoint temperature to calculate midday VPD, we erroneously used 12:00 UTC when downloading the ERA5 temperature data. This means that we used early morning VPD instead of midday VPD.

For the analyses in the revised version of the manuscript we now use temperature and dewpoint temperature at four different times (15:00 – 18:00 UTC) that correspond with local 12:00 in four time zones covering our study area (see Methods). The ERA5 results seem to correspond reasonably well with VPD observations from flux towers in our study area (Figure 1). Using the actual midday VPD logically resulted in changes in the range of VPD found in the meta-analysis (Figure 2). However, we observe no major changes in the direction or magnitude of the leaf transpiration response to seasonal and episodic drought (Figure 3). The response of leaf transpiration to seasonal drought remains not significantly different from 0 and significantly declines in response to episodic drought. This can be explained because the new and correct midday VPD is higher compared to the previous VPD estimate in the wet season, in the dry season and during episodic drought which results in marginal changes in the relative response of leaf transpiration.

Figure 1 The new ERA5 vapor pressure deficit at midday (12:00) compared with vapor pressure deficit measured at 12:00 at 9 flux tower sites in the study area. The dashed line represents the 1:1 line and the solid line a linear regression fit.

We recognise that atmospheric VPD and leaf-to-air VPD can be very different depending on the cooling feedback resulting from leaf transpiration. However, leaf-to-air VPD, leaf temperature or leaf water potential were not consistently provided in the original source papers that provided the stomatal conductance data, preventing us from calculating actual leaf transpiration. We agree that we have to be more careful with the results from this analysis. In the revised version of the manuscript we now discuss the implications of using atmospheric VPD instead of leaf-to-air VPD in calculating leaf transpiration in the Methods (2.2 1st paragraph).

L. 22: it's hard to see how the results could be used as a benchmark for LSMs, given e.g. the unexplained differences in transpiration responses from the leaf and tree- scales to the ecosystem scale. Instead, do the authors mean that the relationships they find between the different variables and wood density could help guide LSM parameterisation efforts in neotropical forests?

Response: yes, we agree with your suggested change of formulation. The new formulation is: "We present new insights into the functioning of tropical forest in response to drought and present novel relationships between wood density and drought responses that can help guide the parametrization of land surface models."

L.32: maybe consider citing Yang et al. 2018 (https://doi.org/10.1038/s41467-018-05668-6), which uses LiDAR and allometric relationships, in place of Zhao and Running? The Zhao and Running paper has temperature dependencies which are problematic and have been discussed in several technical comments....

Response: thank you for your suggested change in referenced literature, Yang et al. 2018 has now replaced Zhao and Running in this references.

L.54: I suggest starting a new paragraph at "Episodic droughts"

Response: this has been changed.

*L.55-56: do tropical North Atlantic SST anomalies affect all the neotropics? Or do they primarily affect the easternmost region?*

Response: according to Marengo et al. 2011, the North Atlantic SST affected the position of the ITCZ, forcing the ITCZ anomalously northward during 2010, resulting in an episodic drought in the southern Amazon Basin.

L. 75: "stomates progressively close" is more exact than "stomates close"

Response: agree, better formulation. This has been changed.

L.76: also: 1. Martin St-Paul et al. 2017 (http://doi.wiley.com/10.1111/ele.12851), 2. Drake et al. 2017 (https://doi.org/10.1016/j.agrformet.2017.08.026) 3. Choat et al. 2018 (https://doi.org/10.1038/s41586-018-0240-x )

Response: agree, these references are a good addition to Buckley et al. 2019 and have been added.

*L.* 84-85: *E* can either stay the same, increase, or decrease during a drought, all of which could result on a decline in  $\Psi$ I.... Also, ksl declines as a result of a decline in  $\Psi$ s

Response: thank you for this nice suggestion, the suggestion has now been incorporated into the text and equation 2 has been removed.

L. 87-88: stomatal closure (described above) and stomatal downregulation are not the same, so the link isn't clear from the current phrasing. Also, using the words "potential" and "potentially" could lead to misinterpretation

Response: this sentence has been reformulated.

L. 88-90: Is this meant as a global statement? Or is it still in the context of neotropical forests? Generally, this is quite variable depending on species, ecosystem, and timing... with different responses being observed at different stages of a drought

Response: yes we agree this is confusing, this sentence has been reformulated.

*L.* 104-106: I think the paragraph would be clearer if this sentence came right after the reference to Sayer et al 2007, L. 102

Response: this has been reformulated.

L. 107: here, maybe repeat what the three spatial scales are

Response: we now repeat the tree spatial scales here.

*L.* 114: change "drought avoiding and drought tolerating strategies" to "drought avoidance or tolerance strategies"?

Response: this has been reformulated.

L.114-115: xylem embolism doesn't always substantially damage the hydraulic pathway, maybe consider rephrasing as "Drought avoidance strategies aim to avoid dangerous declines in  $\Psi$ I that could lead to significant xylem embolism and thus damage..."?

Response: yes we agree, this has been reformulated.

L. 118-120: consider rewriting as: "Conversely, drought tolerance strategies imply [. . .] without significant and/or irreversible embolism-induced losses of hydraulic function"?

Response: thank you for the suggestion, this sentence has been adapted.

L. 120-123: the isohydric vs anisohydric (why use "non-isohydric" rather than anisohydric?) need a bit more explanation, e.g. isohydric species maintain a constant midday  $\Psi$ I but also down-regulate their stomatal conductance. It would also be worth mentioning that the spectrum of isohydric and anisohydric behaviours is quite large, with some species having the capacity to oscillate between more-or-less isohydric or anisohydric behaviours depending on the environmental conditions...

Response: we have omitted all mentions of the isohydric / non-isohydric behaviour following a suggestion by referee #2.

L.131-133: this is a very nice hypothesis! To introduce it, the authors could refer to the work of Rosas et al. 2019 (https://doi.org/10.1111/nph.15684)

Response: thank you, we have restructured this paragraph following a suggestion of referee #2 so that the hypothesis is properly introduced. We use multiple references from neotropical forests to introduce our hypothesis so we see no need to reference the work of Rosas et al. 2019 here.

L. 136-138: I think that moving this sentence to line 133 before "In neotropical..." would make the text flow better

Response: agree, this paragraph has been restructured.

L. 138-140: this is very useful contextualisation, maybe it could make it into the abstract?

Response: we have now included a contextualization in the abstract, mentioning that wood density is not functionally related to plant hydraulic properties but appears to be a good proxy of hydraulic behaviour.

L. 153: typo: "it" to "they"

Response: this has been changed

L. 156: what about measurement techniques and errors? Were those also included in the database? I imagine there would be different margins of error depending on the measurement technique. Also, were there quality checks or did all the above described data make it into the database?

Response: no, we did not differentiate between measurement techniques in the database and this could indeed result in differences in uncertainties. However, since the meta-analysis deals with relative changes in a variable of interest or "effect sizes" the absolute values are less important. Furthermore, since our meta-analysis deals with values that are averaged for the different studies, which each included multiple tree species and individuals, the variability due to differences between species and individuals is much larger compared to measurement uncertainties (see e.g. Santos et al. 2018). It would indeed be very interesting to see how different measurement techniques have an effect on measured drought responses but this is beyond the scope of this study.

*L.* 157: was the time of day not reported? This would highly impact measurements of stomatal conductance and leaf photosynthesis...

Response: very often the day and time of day were not reported for tree and ecosystem scale responses. For the leaf scale responses (stomatal conductance, photosynthesis and leaf water potential) we always used the value at maximum photosynthesis (at midday), except for pre-dawn leaf water potential, naturally. We have now included this information in the Methods (2.1 1st paragraph) of the revised version.

L. 157-158: how many different species, genus, and/or different site averages?

Response: we refer to the database statistics in the next paragraph and included two new tables (Table S1 and S2) in the supplement to provide the reader with this information.

L. 160-161: the information on how the spatial data were extracted is probably not needed

Response: agree, this has been omitted in the revised version. See also specific comment by referee #2.

L. 171: shouldn't "midday vapor pressure deficit" be "monthly averaged midday vapor pressure deficit"?

Response: yes it should, this has been changed.

*L.* 173-175: the authors need to mention that this assumption largely ignores variations in root distributions

Response: we have included a sentence about rooting depth in this paragraph.

L. 177-178: please clarify what "ecosystem performance measures" means

Response: the term "performance" was confusing and has been omitted in the revised version of the manuscript.

L. 179: were all the stomatal conductance measurements made at midday? Also, it is worth mentioning that this relationship assumes a perfect coupling between the stomates and the atmosphere above, i.e. it assumes that the boundary layer conductance to water vapour, gb, is much larger than gs. But in forests with large leaves and dense canopies, decoupling is often observed because gb is relatively small, such that when gs and gb are of similar magnitudes  $E \approx 0.5 \times gs \times D$ . In the context of this study, it is impossible to estimate what the coupling/decoupling factor is at a given location and/or at a given time, but the authors should mention this (in the context of leaf shedding and flushing?), given their findings

Response: we have included two new sentences in this paragraph highlighting the difference between atmospheric VPD and the leaf-boundary layer VPD and how this likely leads to the overestimation of leaf transpiration in our meta-analysis.

L. 184-186: from the text alone, it is very unclear how transpiration was estimated. How does the RMSE represent the linear relationship? Looking at Fig S1, I presume the authors have compiled tree scale measurements of E so, in the analysis, why not just use those measurements (instead of the estimates described by the linear relationship)?

Response: multiple studies that were included in the database reported either maximum sapflux density (at midday) or daily tree transpiration, but not both. We used the linear relationship to

calculate daily tree transpiration for the studies that reported only maximum sapflux density. We have now included a new equation and rephrased the sentence to clarify how we derived total daily transpiration from maximum daily sapflux density values reported in the literature.

*L.* 190: my interpretation of equation 3 is that it should only be valid at steadystate. How did the authors ensure steady-state conditions? Were the data filtered depending on VPD?

Response: yes, equation 3 assumes that the system is at steady state. Steady state conditions were not ensured and we doubt whether we could account for this in a meta-analysis. The "instantaneous soil to leaf hydraulic conductance" calculated here should be regarded a measure of whole-tree hydraulic conductance at midday (Love and Sperry, 2018).

L. 192: typo: "rooting zone" to "root-zone"

Response: this has been corrected.

L. 193-194: strictly speaking, difference between  $\Psi$ I at midday and  $\Psi$ pre-dawn is a proxy of the water gradient within the tree, from the root up to the canopy. For it to equate soil-canopy gradient, further information on tree height would be needed to account for gravitational effects and relate  $\Psi$ pre-dawn to  $\Psi$ s...

Response: yes, this contextualisation has now been added to the sentence.

*L.* 225-227: I realise it's common to use log response ratios when comparing large amounts of data, but why not directly use the percentage change to quantify drought effect size?

Response: the log response ratio is used to derive the test statistics following Lajeunesse (2011) and then back converted to percentage change.

L. 242-243: given the large variability in hydraulic behaviour observed within a genus, is it reasonable to use the genus average as a proxy here? And how many of the location points are affected by this assumption?

Response: yes, we agree that there can be large within genus variability in wood density and hydraulic behaviour. However, across neotropical tree species about 74% of the variation in wood density can be explained by genus level variability (Chave et al., 2006), so genus level wood density could be regarded a useful proxy. Genus averaged wood density was used in 127 cases out of a total of 834 individuals measured. As the wood density was averaged per study, we believe that the effect of using genus averaged wood density instead of species averaged wood density is small. The alternative would be to not provide a wood density value to this individual, which would probably cause more bias in the study averaged wood density than providing the genus average. We have now included these considerations and the number of species where we used the genus average in the revised paragraph.

L. 254: the reference to Figure 2a is needed here too

Response: the reference has been included in the revised version.

*L.* 260: I find hard to believe that this is an actual result and not simply a product of the methods used to calculate the leaf-level transpiration....

Response: the result that on average leaf transpiration does not change from the wet to the dry season (see also final major comment and figures), follows from the averaging of study level responses of 25 studies from which 11 studies showed a (marginal) increase in leaf-level

transpiration and 14 studies a (marginal) decrease in leaf-level transpiration from the wet to the dry season (Figure 5 b). Furthermore, the same result is found when looking at tree scale transpiration which is independent from our calculation of VPD and leaf transpiration. The bias towards studies measuring low wood density trees in sun-exposes canopy positions likely contributes to the overestimation of the dry season decline in stomatal conductance and therefore leaf transpiration (this bias is discussed in the Discussion).

L. 265: but a drop in  $\Psi$  is observed!

Response: yes, this is confusing and has now been reformulated.

L. 275-276: the authors could mention that this is in line with the findings of Rosas et al. 2019 along a mesic-xeric gradient (although their study is not on neotropical species)

Response: we don't see how this sentence about alternating stem and canopy growth relates to the work of Rosas et al. 2019, maybe we have the incorrect paper? We read: "Adjustments and coordination of hydraulic, leaf and stem traits along a water availability gradient" in New Phytologist.

L. 304 (and later): the "WUEi" notation is inconsistent with the "iWUE" notation used in the introduction

Response: this has been changed to iWUE in the revised version of the manuscript.

L. 311: "we observe that" is not needed

Response: agree, this has been omitted.

L. 315: typo: "marginal" should be "marginally"

Response: this has been corrected.

L. 322: give the ranges of variation?

Response: this sentence is confusing and has been omitted in the revised version.

*L.* 320-328: the findings would benefit from being broken down in terms of the dry season (significant) vs episodic drought (mainly not significant)

Response: agree, only for stomatal conductance and leaf transpiration is the relationship with wood density significant (p < 0.05) during episodic drought and shows a similar relationship as during seasonal drought. We have restructured and reformulated this paragraph.

L. 334: the authors need to state that the relationship is not significant...

Response: yes, we have added this in the revised version of the manuscript.

L. 336: "intermediate response" is very vague, please reformulate

Response: "intermediate response" will be omitted, "midday  $\psi l$  declining parallel to a decline in predawn  $\psi l$ " should be a sufficient description of the midday leaf water potential response to declining pre-dawn leaf water potential in the intermediate wood density group. This has been reformulated.

L. 337-342: these findings are very useful!

Response: thank you.

L. 344-346: How is it "similar"? Fig. 6a seems to show far less significance and way more scatter than Fig. 5b

Response: we agree that this sentence is vaguely formulated. By "similar" we refer to the similarity in the relationships between wood density and the direction and magnitude of leaf and tree scale transpiration. Both show an increase of transpiration from the wet to the dry season in studies that measured high wood density species and a decline of transpiration in studies that measured low wood density species. This sentence has been reformulated to highlight what we mean by similar.

L. 356: why "hydrological"? Do the authors mean hydraulic?

Response: yes, we changed this to hydraulic.

L. 358: please replace "cancelled out" by "offset"

Response: this has been changed.

L. 392-394: I don't follow this sentence.... these effects can be consistently observed for weeks, and even months? Do the authors mean that leaf effects are typically observed on shorter time scales due to the "life expectancy" of a leaf compared to a tree, or to an ecosystem?

Response: yes, this is partly what is referred to here but not explicitly mentioned. Leaf shedding and flushing can be a mechanism that results in leaf scale responses being visible on shorter time scales but on even shorter timescales also the opening and closure of the stomates. The purpose of these two sentences is to highlight the presence of buffering in the system, in this case because of non-structural carbohydrates, that could result in the observed inconsistencies in drought responses going from the leaf to the ecosystem. This sentence has been reformulated.

L.405-406: the mention of these "ENSO swings" would be a better fit L. 400, right after the list of references. But what is an ENSO swing? This is never defined...

Response: this sentences has been moved and reformulated.

*L.* 398-407: I'm not entirely clear why the increase in the frequency of episodic droughts is not first mentioned in the results section?

Response: this is increase in episodic drought frequency is now elaborately mentioned in the results (3.2, first paragraph)

L. 538-540: this should come earlier, after L. 120-123

Response: these sentences have been removed following suggestion of referee #2

L. 546: but can also be explained by plant capacitance

Response: we would like to highlight the importance of deep roots to survive long droughts here. We discuss the importance of plant capacitance in other paragraphs of the Discussion.

L. 564: typo: missing "and" after "environments"

Response: "and" has been included.

**Response to Referee #2**

In this study, Janssen et al perform a meta-analysis to look to drought impacts on carbon and water exchange across scales ranging from leaf, to plant, to ecosystem in neotropical rainforests. In particular, the authors contrast physiological responses from seasonal water stress versus 'episodic' drought events. In doing this, the authors also look at wood density as a proxy for plant physiological responses. Its clear that the authors put in a significant amount of work through compiling 138 studies across 229 sites and this study will clearly contribute to the physiological literature on forest drought responses. I do have some major revision suggestions before publication which are summarized here and, in some cases, elaborated in the line specific comments below.

Response: thank you for your elaborate review of our manuscript and acknowledging the work that was put into creating the database behind the meta-analysis. We believe that your comments and suggestions have greatly enhanced the quality of the manuscript. Below we will respond to the major and minor comments raised by you in your report and describe where we made changes in the manuscript following your comments and suggestions.

1) I think that the manuscript would benefit from reframing of the dry season 'drought' as a routine period of decreased water availability. When I think of droughts, I think of a prolonged period of abnormally low rainfall. Given that dry seasons occur every year, I don't see them meeting this definition. This reframing would provide a nice platform to discuss physiological responses to routine (seasonal) stress, such as phenology, versus episodic stress and can help get at important physiological mechanisms. This would involve some substantial reworking of the text, but I think it would really help the story line.

Response: we agree that seasonal drought would not meet the criterium of being a prolonged period of abnormally low rainfall and we highlight this distinction with episodic drought and multi-year drought in the Introduction (L50-L54). However, the term "seasonal drought" is widely used in the literature (Esquivel-Muelbert et al., 2017; Rowland et al., 2013; Stahl et al., 2013) and leaf, tree and ecosystem processes similar to episodic drought are operating during seasonal drought. We discuss the differences between seasonal and episodic drought in the Discussion section 4.3 and mention how for example phenology is likely driving the observed seasonal responses in leaf flushing, shedding and stem growth.

2) Some of the methods are confusing and I think elaborating a bit more, providing and providing a table might help. See line specific comments below

Response: referee #1 also provided suggestions to improve the readability of the Methods. Changes in the text have now been made to elaborate on the Methods, including the calculation of VPD (section 2.1), leaf and tree transpiration (section 2.2) and dry season and drought definition (section 2.3). The requested table with database diagnostics has now been added to the supplement (Table S1).

3) The Figure legends need to be more descriptive of all the features in the figures

Response: the Figure captions and legends have been improved in the new versions of the Figures. See specific comments on the Figures about the exact changes in each Figure.

4) Overall the manuscript does an impressive job discussing a range of processes, but the reader might be more attentive if it were a little shorter. Where possible, I would suggest the authors

minimize extraneous discussion. In particular, I think talking about isohydricity requires significant motivation for a general audience (which is not provided), so I would cut this text.

Response: thank you. We agree that the readability of the paper would improve if it were shorter. While preparing the revised version we have critically examined every paragraph to reduce the length of the manuscript. In the revised version of the manuscript we do not longer use the concept of isohydric and non-isohydric behaviour because, as you mention, it is not suitable for the broad audience of Biogeoscieces (specific comment L121). Furthermore, we agree that focusing on the mechanisms is more interesting that focusing on the (debated) concept of isohydric vs non-isohydric behaviour. Omitting the isohydric vs non-isohydric text has significantly reduced the length of the introduction.

Line-specific comments:

L21-22 There is nothing to back up this statement on LSMs. I suggest the authors remove it

Response: See also specific comment of referee #1. The statement has been rephrased to: ""We present new insights into the functioning of tropical forest in response to drought and present novel relationships between wood density and drought responses that can help guide the parametrization of land surface models."

L35 Khanna et al 2017 Regional dry-season climate changes due to three decades of Amazonian deforestation

Response: thank you for the literature suggestion, Khanna et al. 2017 has been added in the revised version of the manuscript.

L50 after going through the MS, I am confused about where the multi-year drought is presented in the authors analysis

Response: no it is not presented, see for the justification of omitting multi-year droughts L70-71

L53-54 This is exactly why I would argue that 'seasonal drought' is a misnomer

Response: see response to major comment #1.

L121 The concept of isohydricity/anisohydricity will likely not be familiar to a broad audience at Biogeosciences. I would encourage the authors to eliminate the jargon and focus on the mechanisms of interest (see Martinez-Vilalta 2016 "Water potential regulation, stomatal behaviour and hydraulic transport under drought: deconstructing the iso/anisohydric concept") or else devote more space to describing isohydric behaviour

Response: this paragraph has been removed in the revised version of the manuscript, following your suggestion. See also major comment #4.

L131-132 But the authors actually try and demonstrate that wood density is usable as a proxy first, so isn't this more of a hypothesis?

Response: we agree this was a confusing sentence. We have restructured this paragraph.

L145 where does the multiyear drought aspect come in that the authors mentioned in the intro? Intro overall: The introduction is on the longer side (as is the manuscript) and the three separate sections do not have a smooth transition. I would try to combine 1.2 and 1.3 and streamline the text. Given that sections 1.2-1.3 are more about the mechanisms, I would put thes sections first and then say that different types of physiological stress also impact C and W fluxes including seasonal, routine decreases in water availability versus droughts

Response: thank you for your suggestions on restructuring the Introduction. We have now omitted explaining the concept of isohydric vs non-isohydric behaviour from the Introduction, making it significantly shorter. We would like to keep the structure of the three subsections in the Introduction as this structure mirrors that of the results and somewhat also the discussion: 1) what types of drought are there? 2) what are the average responses observed? and 3) can wood density/hydraulic behaviour explain differences between studies? We think this is a logical structure of the Introduction but we agree that many different structures are possible because of the many dimensions present in the meta-analysis (types of drought, different scales: leaf, tree, ecosystem) that have always presented a challenge on how to structure the text.

**L152 ERA5 citation?**

Response: a citation to the ECMWF website is now included.

L153-155 It would be good for the authors to say how many studies were associated with each of these diagnostics and how many studies had multiple observations, maybe a table would be useful?

Response: we have now added a table with a summary of the database in the Supplement of the revised version of the manuscript (Table S1).

L157-159 I am confused about how observations were calculated/recorded across scales? Could the authors elaborate? For example, were ecosystem-level measurements independent from leaf level, or were different levels calculated using observations from different scales by the authors? Both more details and a table detailing number of measurements for each diagnostic and number of studies that cover multiple diagnostics and scales would help

Response: the scales refer to the native "resolution" on which measurements were provided in the source papers. Many studies that measured leaf an tree scale responses provided data on the individual tree level, giving also a species and genus name. For these studies we could link these individual observations to a wood density value. However, many studies reporting ecosystem level responses provided data on the ecosystem level, e.g. 1 ha of forest. We have now included an extensive table in the Supplement (Table S2) which details which measurements were retrieved from each study and each site in the database, specified on the scale of the measurement (leaf, tree or ecosystem). We hope that this table will provide an overview for the reader who is interested on exactly what every study in the database measured and how we subdivided these measures into the three scales.

L161-162, this doesn't need to be included

Response: agree, this is omitted. See also specific comment by referee #1.

L166-167 This is a huge amount of work, I commend the authors

Response: thank you.

L170 it would be good to say the spatial resolution and include a citation for on ERA5 (30km?)

Response: yes indeed, 0.25 degrees is 27.75 km at the equator. This information is now added in the revised version.

L171 monthly average midday VPD?

Response: yes, this is the monthly averaged VPD at midday (12:00). We have now more elaborately described how we derived monthly averaged VPD at midday in the Methods.

**L187 how was this error estimated?**

Response: this was estimated as the RMSE (3.25 kg dm-2 d-1) divided by the mean daily transpiration rate (9.56 kg dm-2 d-1) = 0.34 = 34%. We have now omitted this confusing sentence and included an equation (new equation 3) to clarify how we estimate total daily transpiration.

L219 Figures should be renumbered so Fig 2 comes after Fig 1

Response: this reference to Figure 4 was not necessary and has been omitted in the revised version of the manuscript.

L208-209 It would be good to include a possible caveat about uncertainty associates with ERA5 soil moisture (which I presume is modeled)

Response: we have now compared ERA5 derived REW (based on soil moisture) with site measured soil water potential (Supplement Figure S2) and included a sentence that links to this Figure. In the new sentence we also refer to the possible uncertainties in the ERA5 soil moisture and explicitly mention that ERA5 data is a product of data assimilation and modelling.

L212-213 why the 65% and 10% quantiles? Did the authors test the sensitivity of their results to this assumption?

Response: the lack of a "sensitivity analysis" was also noted by referee #2 L212-213. In the revised version of the manuscript we have now included as sensitivity analysis (see Methods) and we included the results of this sensitivity analysis in the results (referring to new Figure S8 and S9), and discuss its implications in the Discussion.

L220 Could a study be both a dry season and episodic drought? I am a little confused about the partitioning. Also, where does multi-year drought come in?

Response: the partitioning is mentioned in the Methods section 2.3 "Dry season and drought definition", 10% of the driest dry season months were classified as episodic drought (and not anymore as dry season). We simplified the wet season / dry season definition in the revised version of the manuscript, using only the 10% quantile threshold and not the 65% threshold to delineate between wet, dry and episodic drought months. We hope this has helped to make the Methods in general more clear to the reader.

L225 did the authors check to see if it was necessary to log the response? Where did the ENSO data come from?

Response: the log-response ratio was calculated because this is the standard in the method used: the log transformed ratio of means (Lajeunesse, 2011). The ENSO data was retrieved from NOAA, a reference to this dataset has now been included in the revised version.

L254 how can predawn wp be positive 0.22?! Please check for a typo

Response: the minus sign unfortunately ended up on the previous page.

L281 Could the authors include Reco in some of the figures, they refer a change in Reco several times in the text but no visuals are provided

Response: Reco is the same as Ecosystem respiration in Figure 3 and 4. We have removed this abbreviation altogether.

L363 typo include 'us'

Response: yes, this is a typo. This has been corrected.

L370 denoted iWUE previously

Response: yes, this has been corrected.

L401 it is also true that there are more observations post 2000. The authors should discuss how this might impact their results

Response: to avoid a sampling bias, we have now also counted the number of episodic drought months in a regular grid across the entire study area (new Figure S3) and describe this in the Methods (2.3  $2^{nd}$  paragraph), Results (3.2  $1^{st}$  paragraph) and Discussion (4.2  $1^{st}$  paragraph).

L419 it would be nice to put these numbers in a physiological context

Response: it is not entirely clear what is meant by this comment. Could you clarify?

L451-458 I think there is a really nice opportunity to contrast tree physiological strategies when exposed to routine stress (the dry season) versus drought that the current narrative doesn't allow for when both are classified as drought

Response: We agree that this comparison is important and, although we prefer sticking to the term seasonal drought (see further), comparing seasonal drought (routine stress) to episodic drought is one of the main objectives of the meta-analysis. There are basically two reasons why we use the term seasonal drought: 1) the term "seasonal drought" is commonly used in the literature and 2) the difference in drought conditions between the wet season and dry season is comparable to the difference between a regular dry season and episodic drought (Figure 2).

L461 is it the short timescale, or that fact that the plants are used to this type of stress and use phenology to deal with it?

Response: it is not clear what is driving these tree scale responses to drought and on which time scale these responses operate (Doughty et al., 2015; Hofhansl et al., 2014). The purpose of this sentence was to give a potential explanation for the lack of leaf litterfall and leaf flushing responses during episodic drought. We argue that this could be the result of the phenology time-scales and the amplitude of leaf exchange overwriting the more subtle responses to drought, if there are any.

L464 Or maybe title: how do we scale from the leaf to the ecosystem? This is a universal problem in ecology that researchers across many subdisciplines will sympathize with

Response: thank you for the nice suggestion. We have now used the suggested title in the revised manuscript.

L502 see previous suggestion about isohydricity

Response: noted. All mentions of isohydric/non-isohydric behaviour have been omitted in the revised version.

L536 This isn't a great comparison. The authors of this manuscript analyze site-specific data whereas the spatial scale of the Konings study is o(100km). For me, this paragraph does not contribute much to the study and in general I think the isohydricity framework is not useful here (and otherwise)

Response: we agree this is not a great comparison, these two sentences are now omitted.

L575 LSMs are brought up in only in the abstract and conclusion. It doesn't add to the discussion and I would remove this

Response: noted, we do not mention LSMs anymore in the Conclusions in the revised version.

L582 How should they be used to benchmark LSMs? If the authors insist on including this, please the need to spell out the methodology rather than throwing it in as a concluding sentence Other relevant citation: Detto 2018 "Resource acquisition and reproductive strategies of tropical forest in response to the El Niño-Southern Oscillation"

Response: the reference to LSMs are removed.

**Figures combined comments**

**Figure 1**

Fig. 1 panel a make lat/lon bigger

Response: the latitude, longitude axes ticks are now larger.

*b) its really hard for me to wrap my head around what the authors mean by this metric, can they elaborate?*

Response: an explanation of the calculation of the number of episodic drought months per year is now provided in the figure caption.

*d)* What do the dots mean? Please describe this in the legend and also detail what dark and light gray correspond to legend

Response: the explanation of the dot colour and the dark and light grey lines are now explained in the figure caption.

(e) it would be good to remind the reader what the positive/negative ENSO index means

Response: the meaning of the positive and negative ENSO index is now included in the figure caption.

L1131 "terrestrial" isn't capitalized

Response: noted, this has now been changed.

Fig. 1a: the K34 site should be indicated on the map given Fig. 1d and 1e

Response: the location of K34 is now indicated in yellow on the map in the revised version.

Fig. 1b: this is averaged across sites, right? I wonder whether it would make more sense to actually average the episodic drought months across the whole area of neotropical forests shown on the map. This would potentially reduce sampling biases in concluding that episodic droughts have been

increasing in neotropical forests. Alternatively, the authors could consider weighting this by the number of monthly observations per year.

Response: yes, this is averaged across sites. The number of episodic drought months counted in Fig. 1b are independent of the monthly observations retrieved from the literature as all months classified as episodic drought in the time series (1979-2019) at that each site are included. We have now also calculated the number of episodic drought months across the neotropics in a regular grid of 1° (Figure S4) to compare this to the counted number of episodic droughts in Fig. 1b. The data shows a similar increase in episodic drought months per year over the 1979-2019 timespan. We discuss these findings and refer to Figure S4 in the results (Lxx) and discussion (Lxx).

*Fig. 1e: visually, it would be very nice if the ENSO index was coloured to match the wet and dry season and the episodic droughts*

Response: thank you for the suggestion. Wet and dry season colouring of the ENSO index is not possible as wet and dry season occur at different times across the study area. However, we now coloured the ENSO index based on the counted number of episodic droughts across the sites per month to show at which ENSO modes episodic droughts occurred in the study area.

**Figure 2**

Fig. 2a: where do the top soil  $\Psi$ s data come from? The caption says published data, but I didn't find it in the methods?

Response: the references for the soil matric potential are in the supplementary material (main database excel file). We now refer to the supplementary database in the figure caption and included a sentence about the soil matric potential in the methods section on data collection.

Fig. 2a and 2b: yes to the mention of capital letters in the legend, but what does it mean when letters are coupled (e.g. AB in the dry season in Fig. 2a) or when the letter A or B appear during episodic droughts?

Response: for Fig 2a this indicates that there is a significant difference in topsoil water potential between the wet season (A) and episodic drought (B) but not between the dry season and either the wet season or episodic drought (AB). We now explain the coupling of capital letters in the final sentence of the figure caption.

*Fig.2: I imagine the horizontal lines in the box plots show the median, the boxes themselved interquartile ranges, the vertical lines the 5th-95th percentiles and the points are outliers? This needs to be mentioned in the legend*

Response: yes exactly. The values depicted in the boxplots (median, interquartile range, min/max and outliers) are now provided in the figure caption.

Fig. 2 can the authors denote the sample size above each category in the figure? For example, does n=3 for Episodic drought soil-leaf hydraulic conductance in panel b? Make sure to describe the figure fully (detail quantile boxes, median line, error bars, and outliers) in the legend

Response: the sample size is now denoted above Figure 2 a and b. Other descriptions have also been included in the figure caption (see previous comments on this figure).

**Figure 3, 4**

Figs. 3 4: what do the horizontal lines represent? Ranges?

Response: the horizontal lines are the 95% confidence interval range, this description has now been included in the figure caption.

Additionally, it would be useful:

- 1. to also mention the number of data points (or average number of data points per study/site) in brackets;
- 2. to visually separate the variables that were directly retrieved from the literature from those that necessitated further calculations.

Response: we have combined Figure 3 and 4 after the suggestion by referee #2, making the visual comparison of seasonal and episodic drought responses easier. However, this makes the new Figure 3 also more crowded with symbols and numbers. We therefore would not like to add more descriptive statistics besides the number of individuals studies and sites for each measure during each drought comparison. The number of data points can be retrieved from the new Supplementary Table S1. Furthermore, separating the variables that are directly retrieved from the literature and those that required further calculation would add another dimension to this already crowded figure. Besides, some of the variables have data points directly from the literature as well as recalculated values (for example daily transpiration) and some variables are calculated but are a simple difference between two variables directly retrieved from the literature (for example the water potential gradient).

*Fig. 3-4 I would combine these two figures into 1 2-column 3-row figure. I generally really like this format and found it very effective in the Ainsworth review. Great job describing all aspects of the figure in the legend*

Response: thank you. We have now combined Figure 3 and 4 into a single figure (new Figure 3) following your suggestion. We hope this new Figure 3 will benefit the comparison of seasonal and episodic drought responses in the revised version of the manuscript.

**Figure 5, 6**

*Figs. 5 6: so bigger points mean smaller errors? Does that also play a role in the weighting of the solid and dashed lines?*

Response: yes, the size of the points is determined based on the inverse of the sampling variance of that particular study (i.e. precision) and yes, the model is also constructed using inverse-variance weights. These details have now been added to the Methods and are described in the caption of the revised figure.

Fig 5-6 I would combine these figures into a 2-c 3-r figure

Response: Figure 5 and 6 have now been merged following your suggestion, This is now the new Figure 4.

The point size is the inverse of the sample standard error of the effect size in the study.—> so studies with a smaller SE have a larger dot? Moderator= independent variable?

Response: this is now explained in the Methods and the caption of the new Figure 4. See also the response to the final comment of referee #1.

Fig 5 legend, make sure to walk the reader though each panel

Response: the caption has now been adapted to guide the reader through each panel of the Figure.

$$WUE = \frac{A}{E_{pot}} \tag{4}$$

[revised manuscript text omitted]
 measures averaged over wet season months (control) to measures averaged over dry season months (treatment), and in the second comparison the measures averaged over the dry season months (control) with measures averaged over the episodic drought months (treatment) (Figure 3). To be clear, we used only natural drought conditions in the meta-analysis and omitted all data that was acquired in artificial
- 305 drought experiments. Measurements were often always available in pairs or as repeated measurements (wet season-dry season, dry season-episodic drought), so that the variance of the calculated response ratio has to be adjusted for by the Pearson product correlation coefficient between the measurement pairs (Lajeunesse, 2011). For individual tree measurements, which were available for stomatal conductance, photosynthesis, leaf water potential, tree transpiration and sometimes leaf flushing, the average, standard deviation and correlation coefficient were calculated from the pool of measured trees in each study. When
- 310 site averages were used, which was the case for all the other measures, the average and standard deviation calculated from the different measurement years were used. The log response ratio and sample variance of the measures in individual studies and sites were calculated using the escalc routine and the mean effect sizes and 95% confidence intervals in the rma routine, both available in the R package metafor (Viechtbauer, 2017).
- 315 The variability in magnitude and direction of leaf and tree scale responses to drought were related to the average wood density of the species measured in the different studies. To calculate the average wood density for each study, we created a separate dataset including for each study the genus and species names of the individual trees measured in the study. Preferably, the species-specific wood density was retrieved from the original source. However, if this was not possible, we retrieved wood density from a database of wood properties in neotropical tree taxa collated previously by us (Janssen et al., 2020) or from the global wood density database (Chave et al., 2009b; Zanne et al., 2009). If sSpecies-specific wood density was not available in

[revised manuscript text omitted]

---

## Author Response (AR2)

**Response to referees**

*Drought resistance increases from the individual to the ecosystem level in highly diverse neotropical rain forest: a meta-analysis of leaf, tree and ecosystem responses to drought, by TAJ Janssen et al.*

**Referee #1**

The authors have done a thoughtful job considering all of my concerns about the present manuscript in their response, thank you. If the authors make the corrections suggested in their Response, I am satisfied and recommend this manuscript for publications.

A minor comment: I can't figure out what I was talking about with my comment on line 419 either, so please disregard it. I may have been referencing another line, but it wasn't a major concern in any case.

Response: thank you for reviewing our manuscript for the second time. It is nice to hear that you did not encounter any other issues or problems in the revised version of our manuscript.

**Referee #2**

In revision, Janssen et al. much clarified the methods section, updated their result section to reflect methodological changes, and expanded the discussion section. I also really appreciate the authors taking comments and suggestions about their (greatly improved) figures on board, and providing new supplementary material which will be very useful to the readers. Whilst acknowledging all of these positive changes, as well as being convinced by the authors responses to the sampling bias, drought definition, and VPD issues raised before, I still have major methodological concerns about the additional transpiration responses reported in this study.

In my earlier review of the manuscript, I commented that opposite hydraulic strategies were likely being compensated / cancelled at both the leaf- and tree- levels, as the authors chose to analyse all species' leaf-level transpiration responses at once. In their response to this comment, the authors essentially agreed that high wood density and low wood density species were generating opposite responses when going from the wet to the dry season, therefore resulting in no change in leaf-level transpiration (now called potential transpiration). But in the manuscript, lines 297 - 305, the authors invoke gs and VPD compensating effects as a mechanism to explain sustained potential transpiration. This is very confusing, given that besides displaying varied gs responses to soil moisture decline, different species also display different gs sensitivities to changes in VPD. The reader has to wait for Section 3.3 to see mentions of varied hydraulic strategies. Essentially, lines 297 - 305 lack a clear mention of the fact that the lack of change in potential transpiration is also the product of compensatory gs responses to changes in soil moisture and VPD among the different species!

A further issue with the potential transpiration estimates is that, although the authors now acknowledge that atmospheric VPD is not akin to leaf-to-air VPD, they wrongly assume that because the "meta-analysis deals with relative changes", then the "overestimation should not have a major impact on the drought-induced percentage changes in potential leaf transpiration". This in itself assumes that feedbacks from the leaf to the atmosphere, as well as the strength of the coupling relationship between the leaves and the atmosphere above, remains the same from the wet to the dry season; but this is not supported by previous studies, and at the very least warrants introducing more caveats in the methods section.

As a result, the authors also have to be more careful in reporting "WUE" (or potential WUE?) changes between the wet and dry seasons.

The revised methods section now explicits how tree-level transpiration was calculated. I previously thought that the relationship between Jmax and tree-level transpiration shown in Figure S2 had come from another study, but in fact, it seems that it comes from this study? If I understand this properly, then the authors are using a relationship based on Jmax to generate tree-level transpiration data for 17 studies, based on the data from another 17 studies that they themselves have collected. Is this actually adding information, or is this propagating errors / divergence from the fitted line? It would be useful to know whether the 17 data the authors are deriving are for high Jmax (where the data in Figure S2 diverges from the fitted line) or whether they are for low Jmax. Either way, providing uncertainty bounds on these tree-level transpiration estimates would be useful (I think the previous manuscript mentioned a 34% uncertainty, which is non trivial).

Additionally, the editor asked about compounding LAI effects, but I don't find the authors' response very convincing on that point. Given that some species flush more leaves than others, and that leaf flushing is not happening at a constant rate between the wet and the dry season, and between the dry season and an episodic drought, then not normalising tree-level transpiration (in Figure S2) by LAI cannot lead to a sensible comparison (put simply, isn't this like looking at apples in July vs oranges in December?).

In view of the range of uncertainties associated with the leaf- and tree- level transpiration estimates (and given that the authors have now added a measure of crown conductance which can loosely be compared to stomatal conductance when discussing responses across scales), I would suggest:

(1) to completely remove the additional transpiration measures from the meta-analysis; (2) to remove them from the "bulk analysis" and to only incorporate the leaf-level responses in Figure 4 / the analysis clustered by wood density. In that case, Section 2.2 would also benefit from being reworked to clearly separate Epot and WUE from the other additional measures which seem less problematic.

So in my opinion, the additional derived measures of transpiration are taking away from the study's scientific soundness, by relying on too many uncheckable assumptions and by largely confounding information. Otherwise, this study represents a timely and very nice piece of work.

Response: thank you for taking the time to rigorously review the revised version of our manuscript again. We agree that
presenting the derived leaf scale transpiration and associated WUE results alongside results that are directly coming from the original sources is not contributing to the scientific soundness of our analysis. Therefore we followed your suggested option 2 and omitted leaf-scale transpiration from the general analysis and only include this variable in the analysis of wood density effects on drought responses. We replaced al mentions of leaf transpiration that were still present in the text by the more nuanced "potential leaf transpiration". We also restructured the methods section 2.2 to present the estimation of potential leaf-
scale transpiration at the end of this section together with the estimation of daily tree transpiration for studies that only reported maximum daily transpiration.

For tree daily transpiration we have expanded the database where possible and included daily transpiration rates for an additional 3 studies for which we initially did not retrieve daily transpiration, after diving into the original source papers (Figure
S2). We have now also averaged the transpiration rates per study and measured month as this is the level at which the meta-analysis is performed. Furthermore we have now added a 95% confidence interval shading to the linear fit in Figure S2 to show the uncertainty in this relationship. The RMSE of this relationship is 2.89 kg dm$^{-2}$ day$^{-1}$ which is 22% of the mean daily transpiration rate of 13.0 kg dm$^{-2}$ day$^{-1}$. However, this is the absolute error of the transpiration estimate but not of the effect size or percentage change in transpiration that is calculated in the meta-analysis which is expected to be (much) smaller because
this analysis deals with relative changes and we found no change in the relationship between daily transpiration and daily maximum transpiration with drought (ANCOVA, caption Figure S2). We have also included a histogram in Figure S2 to show the frequency distribution of the measurements for which we estimated daily transpiration. The histogram shows that the bulk of these observations fall below Jmax $< 2$ kg dm$^{-2}$ hr$^{-1}$, in the region where the errors in the relationship are relatively small. We chose to keep this gap filling approach in our present analysis because daily maximum transpiration and daily transpiration
are derived from the same measurement technique (heat dissipation method). Furthermore, due to the small sample size of transpiration measurements we prioritize increasing the sample size and geographic coverage of the transpiration measurements over added measurement error. We hope that by expanding the methods about our gap filling approach and updating Figure S2, we have removed your remaining objections to this method.

Considering the LAI effects, we have deliberately stayed out of pretending to calculate absolute values of the transpiration fluxes and only describe the relative changes in these variables reported in the studies. Because daily tree transpiration was always provided on a sapwood-area basis and not a leaf-area basis, changes in sapwood-area from wet to dry season and during episodic drought are driving the relationship between ecosystem (evapo)transpiration and tree daily transpiration. We agree that differences in sapwood-area (not LAI) between the measured individual trees can result in a discrepancy between tree
daily transpiration rates and evapotranspiration. However, the contribution of individual trees to the total sapwood area for
entire forest ecosystems are almost never provided in the original source papers (but see Kunert et al. 2017 in Agricultural and
Forest Meteorology) and we can therefore not correct for differences in sapwood-area and certainly not for temporal dynamics
in sapwood-area.

We thank you again for your elaborate comments and suggestions that have again greatly improved the quality of our
manuscript!

Minor comments
* * *
The flow of text would greatly benefit from shortening any sentence that spans more than 3 lines.
There is a mix of referring to the "dry season" or to a "seasonal drought" in the manuscript. This is somewhat confusing, can
the authors make sure the term used is "dry season" throughout?
Response: we agree that this can be confusing but we would like to maintain the term "seasonal drought" in the text to avoid
cumbersome sentences, for example in the context of: "the seasonal and episodic drought response…"
It occurred to me that using the term "measures" when referring to variables is misleading as it somewhat implies measures of
traits or so. Why not simply refer to variables?
Response: all mentions of "measures" have now been replaced by "variables" in the revised version.
L. 23: whilst I appreciate the reformulation, there needs to be a clear explanation of how this could be done in the main body
of text, perhaps in the conclusions?
Response: after removing the reference to Land surface models in the Conclusions in the first revision, we have now also
removed the reference to LSMs in the abstract.
L. 35: typo, replace "forest" by "forests"
Response: this has been changed
L. 46: is it totally fair to say there is no understanding?
Response: agree, this is not fair, the sentence has been changed to only include "no quantitative overview"
L. 60: please move "in the neotropics" to the start of the sentence
Response: this has been moved
L. 61: "plant functioning" should be "plant function"
Response: this has been changed
L. 63: "in atmospheric demand" instead of "of atmospheric demand"
Response: this has been changed
L. 67: typo, missing the ending "e" in severe. Also, please replace "climate warming" by "climate change"

Response: these adaptations have now been made

L. 88: the text following this equation would be clearer if, instead, the equation were written as a system of two equations

Response: Agree, we have now separated leaf and sapwood area specific transpiration into two equations. This is now also more clearly related to Equation 7 and 8 in the Methods (previously Equations 5 and 6).

L. 89: E is still the leaf transpiration rate here

Response: this is correct here, the VPDs is the VPD at the leaf surface. This was not clear in the previous version but we have included this now (see next response).

L. 90: VPD appears in the equation vs VPDs on this line

Response: Thank you, this was accidentally omitted and has now been included.

L. 90 - 92: these sentences would be clearer if referring to two different equations, which can be done by rewriting Eqn 1 as a system

Response: this has been changed (see response to L88)

L. 93: typo, should be "a drier soil"

Response: this has been corrected

L. 93: rephrase "reduced hydraulic conductance of the xylem as a result of xylem embolism" as "of a reduced xylem hydraulic conductance from embolism"?

Response: this has been changed

L. 94: "all things being equal" is unnecessary text

Response: agree, this has been changed

L. 94: "decline in" instead of "decline of"

Response: this has been changed

L. 96: please replace "if" by "whether"

Response: this has been changed

L. 99 - 100: this is nicely phrased!

Response: thank you!

L. 101: please replace "compared to" by "than"

Response: this has been changed

L. 101: "increase in" instead of "increase of"

Response: this has been changed

L. 116 - 118: this sentences doesn't logically follow

Response: this sentence has been adapted

L. 124: "drought avoidance" instead of "drought avoiding"

Response: this has been changed

L. 133: "cavitation" instead of "dehydratation"?

Response: dehydration is broader than cavitation, it also includes depletion of tissue water that's is not in the xylem vessels. They still avoid cavitation because of high resistance against embolism and can therefore allow some extent of dehydration.

L. 135: it's not just midday! "midday" could simply be removed

Response: agree, this has been changed

L. 150 - 151: would make more sense as "We searched Web of Science for literature published between 1979 and 2019, which matches the span of the ERA5…"

Response: agree, this has been changed

L. 152 - 155: and given Figure S2, tree level daily transpiration too?

Response: Yes, tree daily transpiration has been included.

L. 158: soil matric potential is not the same as soil water potential, but the authors later refer to soil water potential. So which is it?

Response: This is soil matric potential, all references to soil water potential have been adapted.

L. 173 - 175: so it's not actually midday data? This is really unclear from the sentence.

Response: we retrieved temperature from four hourly averages (15:00 – 18:00 UTC), to get local time 12:00 values at the study sites that are located in four time zones (UTC -3, UTC -4, UTC -5 and UTC -6).

L. 179 - 181: therefore I'd expect the authors to comment on the fact that the REW metric is not an exact measure of how much drought an ecosystem is really experiencing

Response: These three sentences have been moved to section 2.3 where we discuss the calculation of REW from ERA5 soil moisture. The limitations of our approach are also discusses in this paragraph. Furthermore, we do not exactly understand what is meant by an "exact measure of how much drought an ecosystem us experiencing", does this not depend on the definition of drought?

L. 188 - 189: it's not using atmospheric VPD instead of leaf-to-air VPD which disregards the fact that leaves are often decoupled from the atmosphere above in dense tropical canopies, it's using gs instead of gc that does. Rather, the atmospheric

VPD instead of leaf-to-air VPD ignores feedbacks. Those two points are different.

Response: this paragraph has been changed.

L. 196: missing "rate" after photosynthesis

Response: this has been changed

L. 201: "summed" is unnecessary

Response: this has been changed

L. 203: "Jdaily" is a weird choice, why not use "Edaily"?

Response: E is generally used to describe leaf-area specific transpiration, J is used to describe sapwood-area specific transpiration (see also equation 2). We have now corrected some of the equations and added units to the variables in the Introduction and Methods to make this more clear.

L. 204: please specify what "a" equates to

Response: a sentence has now been added here explaining the meaning of parameter a, see also more elaborate caption in Figure S2

L. 214: so given Eqn 1 we know have E = Jmax * P? So following that logic, Epot could be Jmax * P instead of gs * VPD? Have the authors compared those values at all?

Response: Yes, we have compared these values but did not include these as they are interdependent. Not many studies have measured both stomatal conductance at the leaf level as well as sap flux density at the tree level. This is done in for example Meinzer et al. 1997, which is included as a reference, where the authors could estimate the boundary layer conductance and the VPD at the leaf surface using sap flux density, stomatal conductance and atmospheric VPD.

Also, where do the P data come from? I don't think that information is in the text.

Response: this has now been included, we chose one standard atmosphere for all measurements (101.325 kPa).

L. 241: "per year" should be moved after "at each site"

Response: this has been adapted

L. 242: typo, "has changed" instead of "have changed"

Response: this has been adapted

L. 245 - 248: please rephrase this sentence

Response: this has been changed

L. 248 - 249: this sentence is completely unrelated?

Response: this has been changed into a new paragraph

L. 251 - 253: this doesn't logically follow

Response: this has been changed

L. 260 - 262: given that you never get back to this "control" and "treatment" linguo, it's just confusing, why keep it?

Response: agree, this has been removed.

L. 272 - 273: isn't that a result rather than methods?

Response: agree, this sentence has been removed.

L. 279 - 280: why not simply drop these individuals? This is yet a further assumption which introduces uncertainty that's hard to quantify, and it is unnecessary given the authors would still have data from 738 individuals to analyse.

Response: this is done because omitting these wood density values would likely result in more uncertainty because we average wood density on the study level. We have now included a sentence about our motivation in the text: "Furthermore, gap filling using genus averaged wood density prevents that missing values cause a large bias in the study averaged wood density in studies that measured relatively few species." For example, one study measures species A and B, that have species averaged wood density values of 0.4 and 0.8 g cm$^3$, respectively. The actual wood density average for that study would then be 0.6 g cm$^3$. If we don't have a species averaged wood density for one of the two species, the error without gap filling is 0.2 g cm$^3$ in both cases. Let's say that the genus average wood density of species B is 1.0 (also an error of 0.2 g cm$^3$), then including this value results in a study average of 0.7 g cm$^3$ (error of 0.1 g cm$^3$).

235 L. 329: typo, "change" instead of "changed"

Response: thank you, this has been changed.

L. 340: typo, "correlated" is missing an "r"

Response: this has been changed.

L. 342 - 344: except only the years after 2000 are visible on the figure…

240 Response: this is shown in Figure S4, which is referenced.

L. 372: "even" is unnecessary

Response: this has been removed.

L. 388 - 390: please rephrase the sentence

Response: this sentence has been rephrased.

245 L. 394 - 396: have the authors looked at any literature on tissue porosity? This could be an explanation relating to low wood density + high stomatal control.

Response: Yes, this is discussed in the Discussion section 4.5, 2nd paragraph: "Sapwood capacitance, the amount of water released from the xylem under a certain pressure, is arguably the only hydraulic property that is functionally related to wood density, as the amount of space available for water storage in the wood scales inversely with wood density (Janssen et al.,

250 2020; Meinzer et al., 2008b; Poorter, 2008; Pratt and Jacobsen, 2017; Ziemińska et al., 2019)."

L. 398 - 399: which falls from the previous finding

Response: Yes, we have included a new introduction to this paragraph that links it with the previous paragraph.

L. 425: and also hydraulic architechture, etc. this statement would benefit from being more nuanced

Response: this sentence has been adapted.

255 L. 437: this is not always true, there is also observational evidence of unchanged soil to leaf hydraulic conductance, for a decreased stomatal conductance during drought

Response: Could you please provide references to this observational evidence? We would like to read it and possible include it to nuance the statements we make here.

L. 447 - 448: why despite? There is no obvious incompatibility between these two things

260 Response: Agree, this sentence has been changed.

L. 473 - 475: the authors have to be careful here, this is in ERA5 which is a reanalysis product, so might contain such a drying trend due to intrinsic biases...

Response: Yes, we have included the possibility that a bias in ERA5 is driving this trend in the text.

L. 492 - 500: this feels like a repeat of one of the results paragraphs and doesn't actually discuss anything per se

265 Response: we agree, this has been rewritten and moved to the results section.

L. 516 - 518: very nice discussion!

Response: thank you!

L. 527 - 528: which begs the question of why the authors chose to include the first couple months after an episodic drought in the wet season, rather than to exclude them from the analysis altogether...

Response: yes we thought about omitting "post drought" months from the analysis but this would result in another arbitrary threshold, which is problematic. Some processes can take many years to stabilize after drought (e.g. Hofhansl et al. 2014) while others return to the "wet season state" within a few days/weeks. So choosing a boundary for how many months after the episodic drought should be omitted is not so easy and we argue also not desirable.

L. 593: hydraulic properties and architecture

Response: hydraulic architecture has been included in this sentence.

L. 619: should the ref to tropical montane cloud forest be mentioned here considering how montane forests have specifically been excluded from the meta-analysis?

Response: no this is not needed and has been omitted.

Figure 1: the legend reads "number of episodic drought months recorded per month at each site in the database", shouldn't it
be "across sites in the database"?

Response: yes it should, thank you. This has been changed.

Also, subplots d and e are wrongly labelled b and c on the figure

Response: thank you for noticing! This has been changed.

Figure 2: whilst the legend now explains what the capital letters are, it omits the fact that the numbers below indicate the
number of? Consulted studies? Sites?

Response: we have included a sentence about what the numbers indicate in the caption. It indicates the number of unique source site combinations. For example, at some sites a variable has been measured and is reported in multiple studies (independent measurements only) while some studies report on multiple sites.

Figure 3: what is blue and what is red? Same applies to some supplementary figures…

Response: this has now been included in the caption.

Figure 4: why not match the colors with those used in Figure 3? Also, please consider separating the r2 values and p values reported on the plot by introducing a semi-colon or increasing the space between them (it's currently a bit hard to read)

Response: the colours have been matched with the previous figures, thank you for the nice suggestion. We have now also added a comma in between the $R^2$ value and the p value.

[revised manuscript text omitted]

---

## Author Response (AR3)

**Response to corrections from the editor**

a reviewer has now seen your revised manuscript, they've raised a few technical queries that once corrected I'm happy to recommend publication. Looking through your manuscript again I also have a few technical points that could be further clarified in the manuscript/fig legend.

- Stomatal conductance and leaf photosynthesis, what are these measurements? Maximum values, in which cases please indicate (e.g. Amax, Gsmax)? Specific times of the day (morning, afternoon)? It is possible that I've raised this point previously but as I look again I'm concerned that a reader will likely want to know that you're meaningful comparing relative changes at appropriate (similar) times of the day.

Response: we have now included "midday" in front of all variable names in Figure 2, 3 and 4 in the cases that midday values were used. That we use midday values for stomatal conductance and leaf photosynthesis is also described in the Methods (2.1, first paragraph).

- As the reviewer points out, what does the R2 of 1 in fig 4b mean - is this a mistake? Can we please remove non-significant R2 information from the plots, after all, you don't plot the line, so I don't follow why you'd report an R2=0 (e.g. Fig 4c,d,f)

Response: this has been changed, please see response to referee corrections.

- Do you explain how the R2 could have increased between Fig 4a and b in the manuscript?

Response: no we did not, this is an interesting point. We have added three sentences explaining this change in R2 between Fig 4a and b to the first paragraph of section 3.3 in the Results.

**Response to corrections from the referee**

I'd like to thank the authors for thoroughly addressing my comments. This is a relevant and timely piece of work, that will make a very nice addition to the literature.

A couple things on Figure 4:

- the caption should now say '(blue)' not '(black)'

Response: thank you for noticing, this has now been changed.

- is R2 = 1 possible in Fig 4b or is this a result of rounding? Maybe consider rounding at three or four decimals as this looks weird

Response: we agree this looks weird but this is not a rounding error. The mixed effect model provides an R2 value, which is 1.0 in this case. This suggests that 100% of the heterogeneity in the model is accounted for. This is different than the $R^2$ in a linear model. We have now included a short sentence in the caption explaining what the $R^2$ indicates.

- why including the n.s. R2s when the relationships are not drawn? They weren't included before and that was more coherent

Response: yes this was erroneously left in the revised version of this figure, these have now been removed.

- 'p is n.s.' would make more sense than 'p < n.s.'

Response: this has been changed.

[revised manuscript text omitted]